# Robust Contextual Optimization with Missing Covariates

**Qingyuan Xu** [1]   **Ruiwei Jiang** [1]

## Abstract

Modern decision-making increasingly relies on contextual features (covariates) to improve optimization under uncertainty. In practice, however, such covariates are often only partially observed due to, e.g., data source heterogeneity or costly data collection. Nonetheless, most existing methods assume fully observed historical data and can become unreliable when this assumption is violated. We address this gap by proposing a distributionally robust optimization approach that exploits incomplete covariates to produce robust decisions *without* imputing a complete dataset. Our method builds ambiguity sets from the observed partial data and incorporates the general structure of the missingness mechanism, ensuring candidate distributions remain consistent with what is observed. Across settings with discrete or continuous covariates and outcomes, we derive tractable reformulations and establish finite-sample out-of-sample performance guarantees. Empirical results across a range of contextual decision-making tasks demonstrate that the proposed integrated approach consistently outperforms state-of-the-art baselines, including various impute-then-optimize pipelines, in both out-of-sample performance and reliability.

## 1. Introduction

Modern decision-making problems under uncertainty rely on *covariates* (features or contexts) that inform future random outcomes. Such problems can be modeled as contextual stochastic optimization (CSO), where decisions adapt to covariates available before the uncertain outcome is revealed. Examples include inventory control, where product or market attributes help predict demand (Ban & Rudin, 2019), and battery arbitrage scheduling, where temperature and electricity load help predict price (Donti et al., 2017); see Sadana et al. (2025) for a recent survey.

Decision makers can learn the dependence between the covariates and outcomes in CSO through historical data. **In practice, however, covariate data is oftentimes partially missing**. Systematic missingness is increasingly common in modern machine learning datasets (Chen & Cummings, 2025), particularly in settings where covariates are drawn from multiple heterogeneous sources or span high-dimensional feature spaces (Van Loon et al., 2024), or where consistent data collection is costly or difficult (Emmanuel et al., 2021). Missing covariates compromise the reliability of CSO by diminishing the informational content of the observed data. Inadequate treatment of missingness can lead to a reduced effective sample size (Little & Rubin, 2019) and, when the missingness mechanism depends on the underlying data variables, further induces distributional shift that biases the ensuing decision-making (Little et al., 2024).

A common response is to pre-process the data before solving the downstream CSO problem. Typical strategies include deletion such as complete-case analysis (CCA), which removes any sample containing at least one missing value, and complete-feature analysis (CFA), which restricts the dataset to fully observed features, as well as a variety of imputation and estimation methods (see Appendix B.1 for a detailed review). From a decision-making standpoint, however, these two-stage pipelines often fall short of decision performance guarantee because the pre-processing step is not calibrated to the downstream optimization objective. As we demonstrate empirically in Section 4, impute-then-optimize approaches, even equipped with asymptotically consistent imputation and robust optimization, still produce suboptimal out-of-sample performance.

Despite these limitations, the CSO literature has to date lacked a systematic treatment of missing covariates and largely assumed access to fully observed covariates. Existing CSO methods include approaches that aim to mitigate (local) sampling error by producing solutions robust to finite-sample variability, such as distributionally robust formulations (Nguyen et al., 2020; Bertsimas & Van Parys, 2022; Esteban-Pérez & Morales, 2022; Kannan et al., 2024) and regularized variants (Wang et al., 2026; Lin et al., 2022).

---

[1]Department of Industrial and Operations Engineering, University of Michigan, Ann Arbor, Michigan, United States. Correspondence to: Qingyuan Xu <qyxu@umich.edu>, Ruiwei Jiang <ruiwei@umich.edu>.

*Proceedings of the 43rd International Conference on Machine Learning*, Seoul, South Korea. PMLR 306, 2026. Copyright 2026 by the author(s).

However, they do not apply to our setting, where missingness induces distributional shift that cannot be captured by local sampling error. There are also works on covariate shift, global contamination, or adversarial perturbations across datasets (Bennouna & Van Parys, 2022; Wang et al., 2024). While these approaches offer protection against worst-case deviations, they presume full access to covariates and do not characterize the missing mechanism. As a result, they are not applicable to our setting of missing covariates. These literature reviews reveal a clear need for CSO methods that can accommodate missing covariates.

### 1.1. Contributions

This paper develops a principled and tractable framework for CSO under partially missing covariates. Our main contributions follow.

**(i) A unified framework for estimation and contextual optimization.** We propose a distributionally robust model that explicitly encodes the missingness mechanism and ensures compatibility with the data. This waives the need for imputation and integrates estimation and optimization.

**(ii) Finite-sample guarantees and asymptotic optimality.** We establish non-asymptotic, out-of-sample performance bounds for our robust decision and show that, with a properly calibrated radius, this decision almost surely recovers the oracle decision as the data size increases.

**(iii) Tractable reformulations for various data geometries.** We derive tractable reformulations for our model under various data geometries with discrete or continuous covariates and outcomes.

**(iv) Superior empirical performance to baseline approaches.** Extensive experiments demonstrate that our unified framework achieves superior out-of-sample performance and reliability to CCA, CFA, and various impute-then-optimize pipelines across a wide range of missingness levels and data sizes.

The remainder of the paper is organized as follows. Section 2 introduces our unified model, Section 3 derives its tractable reformulations and statistical guarantees, and Section 4 reports numerical results. We summarize the notations in this paper in Appendix A. All technical proofs are deferred to Appendix C.

## 2. A Unified Model

### 2.1. Problem Setup

In contextual stochastic optimization, a decision maker observes a covariate $\bar{x} \in \mathcal{X}$, where $\mathcal{X}$ is the covariate space associated with dimension $d_x$, which can help predict an uncertain outcome $Y$, and seeks to make a decision $z$ accord-

ingly. An optimal decision $z^*(\bar{x})$ minimizes a conditional expected cost,

$$z^*(\bar{x}) \in \arg\min_{z \in \mathcal{Z}} \mathbb{E}_{\mathbb{P}^*}\left[c(z, Y) \mid X = \bar{x}\right], \quad \text{(CSO)}$$

where $\mathcal{Z}$ denotes a compact feasible region for $z$, $\mathbb{P}^*(Y|X = \bar{x})$ denotes the true conditional law of the outcome $Y \in \mathcal{Y} \subseteq \mathbb{R}^{d_y}$ given covariates $X = \bar{x}$ and $c(z, y)$ denotes the cost incurred when a decision $z$ is taken and an outcome $Y = y$ is realized. We specify regularity conditions ensuring that (CSO) is well-posed, i.e., the conditional expected cost is finite, and a minimizer exists.

**Assumption 2.1** (Cost Regularity). There exists a $C_{\max} < \infty$ such that $|c(z, y)| \leq C_{\max}$ for all $(z, y) \in \mathcal{Z} \times \mathcal{Y}$. In addition, the map $z \mapsto c(z, y)$ is lower semi-continuous on $\mathcal{Z}$ for all $y \in \mathcal{Y}$.

**Assumption 2.2** (Conditional Expectation Regularity). Let $(\mathcal{X}, \mathcal{B}(\mathcal{X}))$ and $(\mathcal{Y}, \mathcal{B}(\mathcal{Y}))$ be standard Borel spaces equipped with $\sigma$-finite reference measures $\lambda_X$ and $\lambda_Y$. The joint (true) law $\mathbb{P}^*_{XY}$ satisfies $\mathbb{P}^*_{XY} \ll \lambda_X \otimes \lambda_Y$ and $\mathbb{P}^*_X \ll \lambda_X$ with densities

$$p_{XY} := \frac{d\mathbb{P}^*_{XY}}{d(\lambda_X \otimes \lambda_Y)}, \qquad p_X := \frac{d\mathbb{P}^*_X}{d\lambda_X}.$$

For each $z \in \mathcal{Z}$, the map $y \mapsto c(z, y)$ is $\mathcal{B}(\mathcal{Y})$-measurable. In addition, $p_X(x) > 0$ for all $x \in \mathcal{X}$. In particular, the decision-time context $\bar{x}$ satisfies $p_X(\bar{x}) \geq \epsilon_0 > 0$.

*Remark* 2.3. These assumptions cover discrete, continuous, and mixed covariates/outcome distributions. We discuss some of these data geometries in Section 3.

To formalize the missingness mechanism, we partition the entries of $X$ into always-observed coordinates $\mathcal{I}_O$ and potentially-missing coordinates $\mathcal{I}_M$. Accordingly, we write $X = (X_o, X_m)$ for the observed and potentially missing subvectors. We encode missingness through a binary mask $M \in \{0, 1\}^{|\mathcal{I}_M|}$, where $M_j = 1$ indicates that the $j$th entry of $X_m$ is unobserved (masked). Given $M$, the observed covariate becomes

$$\tilde{X}_m = X_m \odot (1 - M) + \mathsf{NA} \odot M, \quad (1)$$

where $\odot$ denotes entry-wise multiplication and $\mathsf{NA}$ denotes a missing entry.

In this work, we restrict attention to settings in which the true data-generating distribution is, at least in principle, *asymptotically identifiable*, without which it would be impossible to develop any reliable methods with finite data. To ensure identifiability, we impose two sufficient conditions on the missing mechanism. First, we require all potentially missing covariates to be observed with strictly positive probability; otherwise, certain components would remain permanently unobserved and therefore unlearnable.

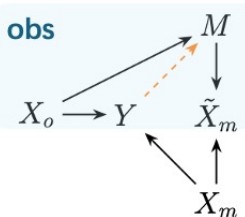

*Figure 1.* Graphical illustration of MAR. The missingness pattern $M$ may depend on both fully-observable covariates $X_o$ and outcome $Y$, while it is conditionally independent of the potentially missing covariates $X_m$ given $(X_o, Y)$. A shaded box indicates the observed quantities.

**Assumption 2.4** (Complete-Case Positivity). It holds that $\lambda_{\min} > 0$, where

$$\lambda_{\min} := \min_{\substack{(x_o, y) \in \mathcal{X}_o \times \mathcal{Y}: \\ p(X_o = x_o, Y = y) > 0}} \mathbb{P}\big(M = \mathbf{0} \mid X_o = x_o, Y = y\big)$$

and $\mathbf{0}$ denotes the all-zero missingness mask.

Second, we restrict attention to a commonly adopted missing mechanism called Missing at Random (MAR). Missingness is commonly categorized into three regimes (Rubin, 1976; Little & Rubin, 2019): (i) Missing Completely at Random (MCAR), where missingness is independent of all variables; (ii) Missing at Random (MAR), where missingness depends on fully observable covariates and outcomes; and (iii) Missing Not at Random (MNAR), where missingness may depend on the unobserved values. MAR is substantially more general and realistic than MCAR and widely adopted in practice, while still avoiding the severe non-identifiability issues inherent in MNAR. Addressing MNAR would require additional structural assumptions or external information (see, e.g., Miao & Tchetgen, 2018). In their absence, the model can be observationally equivalent to MAR (Molenberghs et al., 2008) and is therefore beyond the scope of this work.

**Assumption 2.5** (Missing at Random (MAR)). The missingness indicator $M$ is conditionally independent of $X_m$ given $(X_o, Y)$, i.e., $M \perp X_m \mid (X_o, Y)$.

MAR represents the missingness mechanism by a Markov kernel $q(\cdot \mid x, y) : (\mathcal{X} \times \mathcal{Y}) \to \mathcal{P}(\mathcal{M})$, defined through $q(\cdot \mid x, y) = \mathbb{P}(M \in \cdot \mid X = x, Y = y)$. Accordingly, the class of MAR kernel is defined as

$$\mathcal{Q}_{\mathrm{MAR}} := \big\{ q : q(\cdot \mid x, y) = q(\cdot \mid x', y) \text{ whenever } x_o = x'_o \big\}. \tag{2}$$

Figure 1 illustrates MAR. For example, customer occupation ($X_o$) and income level ($X_m$) can help predict the sale amount $Y$ in e-commerce. While most customers provide their occupations when surveyed, many choose to hide (mask) their income levels. Nevertheless, the masking pattern differs significantly among occupations (e.g., students vs. executives). In this case, conditional on $(X_o, Y)$, $X_m$ carries no additional information about the missingness $M$.

### 2.2. Proposed Framework

To solve (CSO) for an arbitrary covariate, we need access to the (true) joint distribution $\mathbb{P}^*_{XY}$. Despite the missingness, given $\mathbb{P}^*_{XY}$ and a missingness mechanism $q^* \in \mathcal{Q}_{\mathrm{MAR}}$, we can induce the distribution of the observables $(\widetilde{X}, Y, M) \equiv (\widetilde{X}_o, \widetilde{X}_m, Y, M)$ by marginalizing over the unobserved covariates: for any observed $(\widetilde{x}, y, m)$,

$$\mathcal{H}(\mathbb{P}^*_{XY}, q^*)(\widetilde{x}, y, m)$$
$$:= q^*(m \mid \tilde{x}_o, y) \int_{x_m \in \mathcal{A}(\tilde{x}_m, m)} p_{XY}((\tilde{x}_o, x_m), y) \, dx_m$$

where $\mathcal{A}(\tilde{x}_m, m) := \{x_m \mid \tilde{x}_m = x_m \odot (1-m) + \mathsf{NA} \odot m\}$ denotes the "admissible" covariates $x_m$ compatible with the observation $\tilde{x}_m$.

Unfortunately, access to $\mathbb{P}^*_{XY}$ or $q^*$ is rarely available in reality and we are usually endowed with a set of $n$ historical data $\{(\tilde{x}_i, y_i, m_i) : i \in [n]\}$, inducing an empirical distribution $\widehat{\mathbb{P}}^{\mathrm{obs}}_n = (1/n) \sum_{i=1}^n \delta_{(\tilde{x}_i, y_i, m_i)}$. For a reliable estimate and decision, we consider all (latent) joint distributions $\mathbb{P}_{XY}$ for which there exists an admissible missingness mechanism such that their induced distribution of $(\tilde{X}, Y, M)$ lies within a neighborhood of $\widehat{\mathbb{P}}^{\mathrm{obs}}_n$. We formalize this next.

**Definition 2.6** (Latent Ambiguity Set). Let $D(\cdot \| \cdot)$ be a discrepancy measure for probabilities over the observed space $\widetilde{\mathcal{X}} \times \mathcal{Y} \times \mathcal{M}$ (e.g., a divergence or metric; concrete choices to be specified later). For $\varepsilon \geq 0$, we define

$$\mathcal{P}_n(\varepsilon) := \Big\{ \mathbb{P}_{XY} : \exists q \in \mathcal{Q}_{\mathrm{MAR}}, \text{ s.t. } D\big(\widehat{\mathbb{P}}^{\mathrm{obs}}_n \| \mathcal{H}(\mathbb{P}_{XY}, q)\big) \leq \varepsilon \Big\}. \tag{3}$$

By construction, $\mathcal{P}_n(\varepsilon)$ collects those latent laws that produce an observed distribution close to the empirical law under some admissible MAR mechanism. The notion of closeness is governed by the choice of $D$, which depends on the data geometry and are crucial for achieving both statistical efficiency and meaningful out-of-sample guarantees. Different discrepancies also lead to different tractable reformulations and analytical properties, which we develop in Section 3. Then, we define the robust analogue of (CSO):

$$\min_{z \in \mathcal{Z}} \sup_{\mathbb{P}_{XY} \in \mathcal{P}_n(\varepsilon)} \mathbb{E}_{\mathbb{P}_{XY}}[c(z, Y) \mid X = \bar{\mathbf{x}}], \quad \text{(Robust-CSO)}$$

whose well-posedness is addressed in Section 3. We denote by $\widehat{\Phi}_n(\bar{\mathbf{x}})$ and $\hat{z}_n(\bar{\mathbf{x}})$ the optimal value and an optimal solution to (Robust-CSO), respectively. Analogously, we denote by $\Phi^*(\bar{\mathbf{x}})$ the optimal value of (CSO) pertaining to the true conditional law $\mathbb{P}^*(Y | X = \bar{\mathbf{x}})$.

We intent to achieve three goals simultaneously with (Robust-CSO), an appropriate choice of $D(\cdot\|\cdot)$, and a data-driven radius $\varepsilon(n)$: (i) reliability, ensuring that $\mathcal{P}_n(\varepsilon)$ contains the true data-generating distribution with high probability; (ii) statistical efficiency, ensuring that $\widehat{\Phi}_n(\bar{x})$ and $\hat{z}_n(\bar{\mathbf{x}})$ recover their counterparts $\Phi^*(\bar{\mathbf{x}})$ and $z^*(\bar{\mathbf{x}})$ as the data size $n \to \infty$; and (iii) tractability, so that (Robust-CSO) admits a computationally efficient reformulation. The remainder of the paper details how these can be achieved under different data geometries.

# 3. Tractable Reformulations and Statistical Guarantees

For ease of presentation, the main text focuses on results where covariates $X$ are discrete. Section 3.1 studies the case of discrete outcomes $Y$ and Section 3.2 extends to continuous outcomes. We relegate and discuss the case of continuous covariates, wherein both covariates and outcomes can be missing, to Appendix D.

## 3.1. Discrete Outcomes and Covariates

We begin with a discrete setting where both covariates and outcomes take values in a finite alphabet, and take the reference measures in assumption 2.2 to be the counting measures. In this setting, we adopt the Kullback-Leibler (KL) divergence as the discrepancy $D(\cdot\|\cdot)$ defining $\mathcal{P}_n(\varepsilon)$ in (3).

**Definition 3.1** (KL Divergence). Let $P = (p_1, \ldots, p_N)$ and $Q = (q_1, \ldots, q_N)$ be two probability distributions supported on a finite set $\Xi := \{\xi_1, \ldots, \xi_N\}$. The KL divergence from $P$ to $Q$ is defined as

$$D_{\mathrm{KL}}(P \| Q) := \sum_{i=1}^{N} p_i \log\left(\frac{p_i}{q_i}\right),$$

with the conventions that $0 \log(0/q_i) := 0$ and $p_i \log(p_i/0) := \infty$ for $p_i > 0$.

With $D(\cdot\|\cdot) = D_{\mathrm{KL}}(\cdot\|\cdot)$, the ambiguity set becomes

$$\mathcal{P}_{\mathrm{KL}}(\varepsilon) := \left\{\mathbb{P}_{XY} : \exists q \in \mathcal{Q}_{\mathrm{MAR}}, \text{ s.t. } D_{\mathrm{KL}}(\widehat{\mathbb{P}}_n^{\mathrm{obs}} \| \mathcal{H}(\mathbb{P}_{XY}, q)) \leq \varepsilon\right\}.$$

### 3.1.1. Tractable convex reformulation

Although conceptually benign, the robust formulation (Robust-CSO) is not suitable for analysis and efficient computation. The main difficulties arise from (i) the abstract latent-space ambiguity set, and (ii) the non-convex fractional and inf-sup structure of the objective. To overcome these obstacles, we first introduce an equivalent representation based on a transformed ambiguity set. This alternative formulation exposes the problem's structural properties and, through variable transformations and primal-dual arguments, leads to a tractable convex reformulation.

We first represent a latent distribution by a vector $p \in \Delta(\mathcal{X} \times \mathcal{Y})$ and introduce an indicator matrix $T \in \{0,1\}^{n \times |\mathcal{X} \times \mathcal{Y}|}$ with $T_{i,(x,y)} = 1$ whenever $y = y_i$, $x_o = \tilde{x}_{o,i}$ and $x_m \in \mathcal{A}(\tilde{x}_{m,i}, m_i)$, i.e., $(x,y)$ is compatible with the $i$-th observation.

**Proposition 3.2** (Equivalence of KL-Robust CSO). *Define an ambiguity set*

$$\mathcal{D}(\gamma) := \left\{p \in \Delta(\mathcal{X} \times \mathcal{Y}) : \frac{1}{n}\sum_{i=1}^{n} \log\left((Tp)_i\right) \geq \gamma\right\},$$
(4)

*and, for any $\varepsilon \geq 0$, define*

$$\gamma := \sum_{\xi \in \widehat{\Sigma}} \frac{n_\xi}{n} \log\left(\frac{n_\xi}{n}\right) - \frac{1}{n}\sum_{i=1}^{n} \log \widehat{\mathbb{P}}(m_i \mid x_{o,i}, y_i) - \varepsilon,$$

*where $n_\xi$ is the number of samples such that $(\tilde{x}_i, y_i, m_i) = \xi$, and $\widehat{\Sigma} \subseteq \tilde{\mathcal{X}} \times \mathcal{Y} \times \mathcal{M}$ is the empirical support of the data. Then, (Robust-CSO) is equivalent, in terms of optimal value and solutions, to:*

$$\inf_{z \in \mathcal{Z}} \sup_{p \in \mathcal{D}(\gamma)} \mathbb{E}_p[c(z, Y) \mid X = \bar{\mathbf{x}}].$$
(5)

The reformulation (5), and equivalently (Robust-CSO), is well-posed under mild conditions.

**Proposition 3.3** (Attainability of (Robust-CSO)). *Under assumptions 2.1–2.2, (Robust-CSO) attains its worst-case distribution and optimal solution(s) for any feasible $\gamma$ if there exists an $\epsilon_1 > 0$ such that, for any $p \in \mathcal{D}(\gamma)$, $p(\bar{\mathbf{x}}) := \sum_{y \in \mathcal{Y}} p(\bar{\mathbf{x}}, y) \geq \epsilon_1$.*

*Remark* 3.4 (Sufficient Conditions for Attainability). Define

$$L_0^\star := \sup\left\{\frac{1}{n}\sum_{i=1}^{n} \log\left((Tp)_i\right) : p \in \Delta(\mathcal{X} \times \mathcal{Y}), \ p(\bar{\mathbf{x}}) = 0\right\},$$

$$L_1^\star := \sup\left\{\frac{1}{n}\sum_{i=1}^{n} \log\left((Tp)_i\right) : \ p \in \Delta(\mathcal{X} \times \mathcal{Y})\right\}.$$

Both quantities can be efficiently solved via a convex program and $L_0^\star \leq L_1^\star$. Setting $\gamma \leq \gamma_{\max} := L_1^\star$ guarantees feasibility, while choosing $\gamma > L_0^\star$ further ensures non-negligible probability mass at $\bar{\mathbf{x}}$ for any $p \in \mathcal{D}(\gamma)$. Moreover, if there is a complete observation at $\bar{\mathbf{x}}$ in the data, then $L_0^\star = -\infty$ and the condition is trivially satisfied .

We next develop reformulation under the attainability condition. The inner problem of (5) is nonconvex, but exhibits a hidden convex structure that enables tractable reformulations. First, the ambiguity constraint involves a sum of logarithms applied to linear mappings $Tp$. This structure suggests that a suitable lifting, combined with an appropriate variable substitution, can expose a perspective form. Second, the fractional structure of the conditional expectation

in the objective can be homogenized through a change-of-variables, recasting the objective as a linear function over an augmented variable set. Motivated by these observations, we introduce the following notations: for a fixed $z$, we define for each $j = 1, \ldots, |\mathcal{X} \times \mathcal{Y}|$

$$c(z)_j = c(z, y_j)\, 1\{x_j = \bar{\mathbf{x}}\}, \quad d_j = 1\{x_j = \bar{\mathbf{x}}\},$$

so that only atoms with covariate component $\bar{\mathbf{x}}$ contribute to the conditional expectation. We then introduce the change-of-variables $v := \frac{p}{d^\top p}$, $t := \frac{1}{d^\top p}$, $u := \frac{Tp}{d^\top p}$ in the inner problem of (5). After lifting, we obtain the following equivalent convex problem.

**Proposition 3.5** (Convex Reformulation for Inner Problem). *The inner problem of* (5) *is equivalent to*

$$\sup_{v \succeq 0, u \succeq 0, t \geq 0} \quad c^\top v$$
$$s.t. \quad n\gamma t + \sum_{i=1}^{n} t \log\left(\frac{t}{u_i}\right) \leq 0, \qquad (6)$$
$$u = Tv, \mathrm{e}^\top v = t, d^\top v = 1.$$

The dual of (6) gives rise to a dual representation for (5).

**Theorem 3.6** (Convex Representation for (Robust-CSO)). *For any $\epsilon > 0$,* (5) *admits the following reformulation:*

$$\inf_{\substack{\lambda \geq 0,\, \mu \leq 0,\, \nu \in \mathbb{R}, \\ \alpha \in \mathbb{R}, z \in \mathcal{Z}}} \quad -\alpha$$
$$s.t. \quad c(z) - T^\top \mu + \nu 1 + \alpha d \leq 0,$$
$$\lambda\left(n(1 + \gamma) + \sum_{i=1}^{n} \log\left(-\mu_i/\lambda\right)\right) + \nu \geq 0.$$

*If $c(\cdot, y)$ is convex in $z$ and $\mathcal{Z}$ is convex, then this representation is a convex optimization problem.*

The representation in Theorem 3.6 can be implemented using off-the-shelf optimization solvers. Note that its scalability is not governed by the data size $n$, because data sharing the same observed triple $(\tilde{x}, y, m)$ can be aggregated (see Appendix D.1). The computational bottleneck instead arises from the discrete latent support, which scales with $|\mathcal{X}| \times |\mathcal{Y}|$ and determines the number of first constraints. To address this, separation methods can be used to introduce violating atoms on-the-fly, allowing the number of active constraints to reflect the effective, as opposed to full, support.

The proposed framework can also be extended in a straightforward manner to the setting where both covariates and outcomes may be missing, by augmenting the missingness mask to include the outcome component. We present this extension in Appendix D.2.

### 3.1.2. STATISTICAL GUARANTEE

We now establish statistical guarantees for (Robust-CSO). We first show that, with a properly calibrated radius $\gamma_n(\eta)$

in its representation (5) with respect to data size $n$ and confidence level $1 - \eta$, the worst-case conditional cost acts as an *upper confidence bound* on the true conditional cost, therefore producing a reliable decision rule. Specifically, we calibrate the radius by setting $\gamma_n(\eta)$ as the minimum of $\gamma$ defined in Proposition 3.2 and $\gamma_{\max} \equiv L_1^\star$ defined in Remark 3.4, and

$$\varepsilon_n(\eta) := \frac{k \log(n + 1) + \log(1/\eta)}{n}, \qquad (7)$$

where $k$ is the support cardinality of the true latent law.

**Theorem 3.7** (Out-of-Sample Disappointment). *Under Assumptions 2.1–2.5, it holds with probability at least $1 - \eta$ and for all $z \in \mathcal{Z}$ that*

$$\sup_{p \in \mathcal{D}\left(\gamma_n(\eta)\right)} \mathbb{E}_p\big[c(z, Y) \mid X = \bar{\mathbf{x}}\big] \geq \mathbb{E}_{\mathbb{P}^*}\big[c(z, Y) \mid X = \bar{\mathbf{x}}\big].$$

*In particular, $\widehat{\Phi}_n(\bar{\mathbf{x}}) \geq \mathbb{E}_{\mathbb{P}^*}\big[c(\hat{z}_n, Y) \mid X = \bar{x}\big] \geq \Phi^*(\bar{\mathbf{x}})$.*

A proof is provided in Appendix C.4, which mainly uses KL concentration inequalities and the event containment $\{\omega : D_{\mathrm{KL}}(\widehat{\mathbb{P}}_{XYM}, \mathbb{P}^*_{XYM}) \leq \epsilon_n(\eta)\} \subseteq \{\omega : \mathbb{P}^*_{XY} \in \mathcal{D}(\gamma_n(\eta))\}$, which implies $\Pr\left[\mathbb{P}^*_{XY} \in \mathcal{D}(\gamma_n(\eta))\right] \geq 1 - \eta$. Hence, with confidence level $1 - \eta$, the true latent joint distribution is guaranteed to lie inside the ambiguity set for *every* sample size $n$, which can be translated to performance bounds for (Robust-CSO). As a remark, the probability of the out-of-sample disappointment decays *exponentially fast*. Furthermore, (Robust-CSO) admits asymptotic consistency.

**Theorem 3.8** (Asymptotic Consistency). *Under assumptions 2.1–2.5, choose a confidence sequence $\{\eta_n\}_{n \geq 1} \subset (0, 1]$ with $\sum_{n=1}^{\infty} \eta_n < \infty$, $\lim_{n \to \infty} \gamma_n(\eta_n) = 0$ and set $\gamma_n := \gamma_n(\eta_n)$ as in Theorem 3.7. Then, the following hold if an optimal solution $\hat{z}_n$ to (Robust-CSO) exists: (a) Value consistency. $\Phi_n \downarrow \Phi^*$ almost surely as $n \to \infty$; (b) Optimizer consistency. With probability one, every accumulation point of the sequence $\{\hat{z}_n\}$ is an optimal solution to (CSO).*

### 3.2. Continuous Outcomes

We now extend to continuous outcome distribution, where the outcome variable $Y$ takes values in a compact space, and the reference measure $\lambda_Y$ in assumption 2.2 is the Lebesgue measure. In this case, however, a direct extension of the KL-based formulation is not available. A key insight is that the KL divergence $D_{\mathrm{KL}}(\hat{P}_n^{\mathrm{obs}} \parallel Q)$ is not continuous in $\hat{P}_n^{\mathrm{obs}}$ when observations are continuous. Consequently, weak convergence of the empirical measures does not imply convergence in the KL divergence on general measurable spaces (Yang & Chen, 2018). As a result, no meaningful KL concentration inequality is available without posing additional structure without imposing additional structure, and

the out-of-sample performance guarantees as in Theorem 3.7 can no longer be expected.

This observation motivates the use of discrepancy measures that metrize weak convergence while respecting the geometric structure of continuous outcome spaces. Natural candidates include the Lévy-Prokhorov (LP) metric. To this end, we introduce an *LP-projected KL divergence*, which combines KL divergence with LP-metric neighborhoods.

**Definition 3.9** (PKL Divergence). Let $(\Xi, d)$ be a metric space. The LP distance between probability measures $\mathbb{P}$ and $\mathbb{Q}$ on $(\Xi, \mathcal{B}(\Xi))$ is defined as

$$d_{\mathrm{LP}}(\mathbb{P}, \mathbb{Q}) := \inf \big\{ \varepsilon > 0 : \mathbb{P}(A) \le \mathbb{Q}(A^{\varepsilon}) + \varepsilon$$
$$\text{for all measurable } A \subseteq \Xi \big\},$$

where $A^{\varepsilon} := \{ \xi \in \Xi : \ \inf_{z \in A} d(\xi, z) < \varepsilon \}$ denotes the $\varepsilon$-enlargement of $A$. Given a reference distribution $\mathbb{P}_0$ and radius $\delta_0 > 0$, the associated LP ball is $B_{\mathrm{LP}}(\mathbb{P}_0, \delta_0) := \{ \mathbb{P} : \ d_{\mathrm{LP}}(\mathbb{P}, \mathbb{P}_0) \le \delta_0 \}$. Then, we define a *LP-Projected KL (PKL) Divergence* from a distribution $\mu$ to this LP neighborhood by

$$\mathrm{D}_{\mathrm{PKL}}(\mu; \mathcal{H}(\mathbb{P}_{XY}, q), \delta) := \inf_{\mathbb{Q} \in B_{\mathrm{LP}}(\mathbb{P}, \delta)} D_{\mathrm{KL}}(\mu \| \mathcal{H}(\mathbb{Q}_{XY}, q)).$$

$D_{\mathrm{PKL}}$ shares a similar embedding structure with the robust KL divergence studied in the universal hypothesis testing for nonparametric continuous models (Yang & Chen, 2018), as well as with the robust predictors proposed for oblivious adversaries (Bennouna & Van Parys, 2022). Our setting differs in that the KL divergence and the LP metric are constructed over different spaces (namely, the observable space and the latent space). This distinction is essential in our context and is directly motivated by the presence of missing covariates.

To establish asymptotic convergence in the continuous-outcome setting while keeping the inner minimization over $D_{\mathrm{PKL}}$ tractable, we restrict the projected law $\mathbb{Q}_{XY}$ to be in a sequence of progressively richer model classes. This follows the idea of sieve methods in statistics (Grenander, 1981): as the sample size increases, these classes become more flexible, so that the effect of this restriction becomes asymptotically negligible. In particular, we adopt a bin-based model: the outcome space is partitioned, the projected law is piecewise constant over the same outcome bin for each covariate value, and the partition is refined as the sample size grows. Incorporating this idea into the LP-projected KL construction yields a binned PKL.

**Definition 3.10** (Binned PKL ambiguity set). Let $\Pi_J = \{Y_1, \ldots, Y_J\}$ be a measurable partition of the outcome space $\mathcal{Y}$ and let $\mathcal{B}_J$ denote the class of laws whose conditional density or mass function of $Y$ given $X = x$ is constant on its support within each bin $Y_j$, for every $x \in \mathcal{X}$

and $j \in [J]$. Denote $h_J := \max_{1 \le j \le J} \mathrm{diam}(Y_j)$ and $D_{\mathrm{BinPKL}}(\mu; \mathcal{H}(\mathbb{P}_{XY}, q), \delta; J)$ as

$$\inf_{\mathbb{Q}_{XY} \in B_{\mathrm{LP}}(\mathbb{P}_{XY}, \delta) \cap \mathcal{B}_J} D_{\mathrm{KL}}(\mu \| \mathcal{H}(\mathbb{Q}_{XY}, q)).$$

For $\epsilon, \delta \ge 0$, the binned PKL ambiguity set is

$$\mathcal{P}_{\mathrm{BinPKL}}(\epsilon, \delta; J) := \Big\{ \mathbb{P}_{XY} | \ \exists q \in \mathcal{Q}_{\mathrm{MAR}}, \ \text{s.t.}$$
$$D_{\mathrm{BinPKL}}\Big(\widehat{\mathbb{P}}_n^{\mathrm{obs}}; \mathcal{H}(\mathbb{P}_{XY}, q), \delta; J\Big) \le \epsilon \Big\}.$$

#### 3.2.1. TRACTABLE REFORMULATION

We begin by deriving an equivalent representation in the sense that the BinPKL ambiguity set admits a pair of tight sandwiching bounds in terms of a simplified ambiguity set.

**Proposition 3.11** (Sandwiching bounds). *For any $\epsilon, \delta \ge 0$,*

$$\gamma := \sum_{\xi \in \widehat{\Sigma}} \frac{n_\xi}{n} \log\left(\frac{n_\xi}{n}\right) - \epsilon - \frac{1}{n} \sum_{j=1} \log \widehat{\mathbb{P}}(m_i \mid x_{o,i}, y_i),$$

*any $\mathbb{P}(y \mid \bar{\mathbf{x}})$ induced by $\mathbb{P}(x, y) \in \mathcal{P}_{\mathrm{BinPKL}}(\epsilon, \delta; J)$ is contained in*

$$\mathcal{K}(\gamma, \delta; J) := \Big\{ \mathbb{P}(y \mid \bar{\mathbf{x}}) \Big| \mathbb{P}(y \mid \bar{\mathbf{x}}) \in B_{\mathrm{LP}}(\mathbb{Q}(y \mid \bar{\mathbf{x}}), \delta), \mathbb{Q} \in \mathcal{B}_J$$
$$\frac{1}{n} \sum_{i=1}^n \log\left( \sum_{x_{m,i} \in \mathcal{A}(\tilde{x}_{m,i})} \mathbb{Q}(x_{o,i}, x_{m,i}, y_i) \right) \ge \gamma \Big\}.$$

*In addition, the set $\mathcal{K}(\gamma, 2\delta; J)$ is* sandwiched *between the conditional laws generated by the PKL ambiguity sets:*

$$\big\{ \mathbb{P}(y \mid \bar{x}) : \mathbb{P}_{XY} \in \mathcal{P}_{\mathrm{BinPKL}}(\epsilon, \delta\epsilon_1; J) \big\} \subseteq \mathcal{K}(\gamma, 2\delta; J)$$
$$\subseteq \big\{ \mathbb{P}(y \mid \bar{x}) : \ \mathbb{P}_{XY} \in \mathcal{P}_{\mathrm{BinPKL}}(\epsilon, 3\delta; J) \big\}.$$

*Here, $\epsilon_1 > 0$ is the constant introduced in Proposition 3.3 and Remark 3.4.*

The sandwich bounds imply that $\mathcal{K}(\gamma, \delta; J)$ is essentially equivalent to the BinPKL ambiguity set up to the calibration of their radii, while being substantially more tractable for computation. In particular, the next proposition shows that the distribution $\mathbb{Q}^*(y \mid \bar{\mathbf{x}})$ in $\mathcal{K}(\gamma, \delta; J)$ that maximizes $\mathbb{E}_{\mathbb{Q}}[c(z, Y)|X = \bar{\mathbf{x}}]$ is supported on at most $K + 1$ atoms, where $K$ is the number of distinct outcomes of $Y$ observed in the data.

**Proposition 3.12** (Structure of Worst-Case Distributions). *Let $\{y_1, \ldots, y_K\}$ denote the distinct outcomes observed in the data for which $\mathcal{A}(\tilde{x}_i) = \bar{\mathbf{x}}$. For $z \in \mathcal{Z}$, define*

$$y_\infty^{\mathcal{N}}(z) := \arg\max_{y \in \mathcal{Y}} c(z, y).$$

*Then, it holds that if $\delta \ge h_J$,*

$$\mathrm{supp}\left(\mathbb{Q}^*(\cdot \mid \bar{\mathbf{x}})\right) \subseteq \{y_1, \ldots, y_K\} \cup \{y_\infty^{\mathcal{N}}(z)\},$$

*where $\mathbb{Q}^*(\cdot \mid \bar{\mathbf{x}}) \in \arg\max_{\mathbb{Q} \in \mathcal{K}(\gamma, \delta)} \mathbb{E}_{\mathbb{Q}}[c(z, Y)|X = \bar{\mathbf{x}}]$.*

We note that the regime $\delta \geq h_J$ is precisely what is used in practice. As we will show in Section 3.2.2, a finite-sample performance guarantee can be established only by choosing the radius $\delta_n$ large enough to dominate the bin mesh $h_J$. Thanks to Proposition 3.12, we can apply change-of-variables analogous to the discrete case and derive a dual representation for the extension of (Robust-CSO).

**Theorem 3.13** (Convex Representation). *It holds that*

$$\inf_{z \in \mathcal{Z}} \left\{ \sup_{\mathbb{Q} \in \mathcal{K}(\gamma, \delta)} \mathbb{E}_{\mathbb{Q}}[c(z, Y)|X = \bar{\mathbf{x}}] \right\} \quad \text{(Robust-CSO-CO)}$$

*admits the following reformulation:*

$$
\begin{aligned}
\min \quad & \delta M - (1 - \delta)\rho - \eta \\
s.t. \quad & M \geq \max_{y \in \mathcal{Y}} c(z, y), \\
& -(T^\top \boldsymbol{\varphi})_j - \mu e_j + (\eta + \sum_{k: y_k \in Y_{(j)}} \pi_k) d_j - \zeta f_j \leq 0, \ \forall j \\
& -\mu e_j + \eta + M + \rho + \zeta \leq 0, \\
& c^{\mathcal{N}}(z, y_k) - \pi_k + \rho \leq 0, \quad \forall k \in [K] \\
& -\lambda \left( n(1 + \gamma) + \sum_{i=1}^n \log\left(\frac{\psi_i}{\lambda}\right) \right) + \mu \leq 0.
\end{aligned}
$$

*where $c^{\mathcal{N}}(z, Y) := \sup\{c(z, Y - \Delta) : \Delta \in \mathbb{B}_1(0, \delta), \ Y - \Delta \in \mathcal{Y}\}$, $Y_{(j)}$ denotes the outcome region associated with the $j$-th cell $(x, Y)_j$, and $f_j = 1$ if $y_\infty^{\mathcal{N}}(z) \in Y_{(j)}$ otherwise 0. If $\mathcal{Y}$ contains the $h_J$-enlargement of the data support as a subset, $c(z, y)$ is separately convex, and $\mathcal{Z}$ is convex, then this representation is a convex optimization problem.*

In Appendix D.3, we present a sequence of implementation refinements that help solve the representation even faster.

### 3.2.2. STATISTICAL GUARANTEES

We first establish that, if the data is without missingness, the robust KL divergence proposed by (Yang & Chen, 2018)

$$D_{\text{RKL}}(\mu, B_{\text{LP}}(\mathbb{P}_0, \delta_0)) := \inf_{\mathbb{P} \in B_{\text{LP}}(\mathbb{P}_0, \delta_0)} D_{\text{KL}}(\mu \,\|\, \mathbb{P})$$

can characterize the data-generating distribution with high confidence. To this end, we derive a concentration inequality as follows.

**Proposition 3.14** (RKL Concentration Inequality). *Let $\hat{\mathbb{P}}_n$ be the empirical distribution based on $n$ i.i.d. observations $(x_i, y_i) \in \mathcal{X} \times \mathcal{Y}$. Then, for every given $\epsilon, \delta > 0$,*

$$\Pr\left[ D_{\text{RKL}}\left(\hat{\mathbb{P}}_n, B_{\text{LP}}(\mathbb{P}^*_{XY}, \delta)\right) > \epsilon \right] \leq$$

$$\left(\frac{8}{\delta}\right)^{|\mathcal{X}|\left(1 + \frac{\text{diam}(\mathcal{Y})}{\delta}\right)^{d_y}} \exp\left( -\frac{n\delta^2}{2}\left(e^{-\epsilon} + e^{-2\delta^2}\right)^{-1} \right),$$

*where $\text{diam}(\mathcal{Y})$ denotes the diameter of $\mathcal{Y}$ and $d_y$ denotes the dimension of $\mathcal{Y}$.*

Proposition 3.14 provides a coverage guarantee at the complete-data level. We next propagate this guarantee first to the PKL ambiguity set and then to the induced conditional-law ambiguity set. The key step is a set-containment argument linking the RKL ambiguity set to the PKL ambiguity set; Proposition 3.11 subsequently transfers the resulting inclusion to $\mathcal{K}(\gamma, \delta)$. This yields the following out-of-sample disappointment bound. We can then invoke Proposition 3.11 to transfer this inclusion from the PKL ambiguity set to $\mathcal{K}(\gamma, \delta)$. This gives rise to the following out-of-sample disappointment bound.

**Theorem 3.15** (Out-of-Sample Disappointment Bound). *Define $\kappa = 1 + \text{diam}(\mathcal{Y})$, $L = \left(\frac{4|\mathcal{X}|\kappa^{d_y}}{d_y + 2}\right)^{\frac{1}{d_y + 2}}$, confidence levels $\eta_n := \exp\left\{ -4L^2\left(\frac{n}{\log n}\right)^{\frac{d_y}{d_y + 2}} \cdot \log \log n \right\}$, and radii $\delta_n(\eta_n) := \frac{2L}{\epsilon_1}\left(\frac{\log n}{n}\right)^{\frac{1}{d_y + 2}} + \frac{2}{\epsilon_1}h_J$. Set $\gamma_n(\eta_n) = \gamma_{\max} - (7)$. Then, under Assumptions 2.1–2.5 and for all $n \geq 2$, it holds with probability at least $1 - \eta$ that*

$$\sup_{p \in \mathcal{K}(\gamma_n, \delta_n)} \mathbb{E}_p\left[c(z, Y) \mid X = \bar{\mathbf{x}}\right] \geq \mathbb{E}_{\mathbb{P}^*}\left[c(z, Y) \mid X = \bar{\mathbf{x}}\right]$$

*for all $z \in \mathcal{Z}$. In particular, $\widehat{\Phi}_n(\bar{\mathbf{x}}) \geq \mathbb{E}_{\mathbb{P}^*}\left[c(\hat{z}_n, Y) \mid X = \bar{\mathbf{x}}\right] \geq \Phi^*(\bar{\mathbf{x}})$.*

The choice of confidence levels $\eta_n$ ensures that they decay to zero at an exponential rate as $n \to \infty$. Indeed, the dominant term $n^{d_y/(d_y + 2)}$ in the exponent diverges, while the correction terms $(\log n)^{-d_y/(d_y + 2)}$ and $\log \log n$ grow slowly, resulting in $\eta_n \to 0$ exponentially fast. Moreover, the series $\sum_{n=2}^\infty \eta_n$ is convergent. This summability ensures that the ensuring probabilistic statements can be upgraded to hold with probability one via the Borel-Cantelli lemma, a property that will be used in our following asymptotic consistency analysis.

**Theorem 3.16** (Asymptotic Consistency). *Under Assumptions 2.1–2.5, assume bounded density and lipschitz continuous score function, and $y \mapsto c(z, y)$ is Hölder continuous. Let $\eta_n$, $\gamma_n(\eta_n)$, and $\delta_n(\eta_n)$ be calibrated as in Theorem 3.15. Then, the following hold if an optimal solution $\hat{z}_n$ to (Robust-CSO-CO) exists and there exits $\alpha > 0$ s.t. $h_J n^\alpha \to 0$, $\frac{J(n) \log n}{n} \to 0$ as $n \to \infty$: (a) Value consistency. $\Phi_n \downarrow \Phi^*$ almost surely as $n \to \infty$; (b) Optimizer consistency. With probability one, every accumulation point of the sequence $\{\hat{z}_n\}$ is an optimal solution to (CSO).*

## 4. Numerical Experiments

We now present numerical experiments to demonstrate the proposed robust framework. To maintain focus, we report the key findings in the main text, and the data-generation and implementation details are deferred to Appendix E.

We consider a contextual newsvendor problem with a cost function $c(z, Y) := c_u (Y - z)_+ + c_o (z - Y)_+$, with $c_u, c_o > 0$. Here, $z$ denotes an order quantity (decision variable), $Y$ denotes the random demand (outcome) to be fulfilled, and the cost function penalizes the supply-demand mismatch. To better predict $Y$, we resort to discrete co-variates $X$ and historical data of $(X, Y)$, which are subject to random missingness. We compare our proposed approach (called MAR-KL DRO) with several standard baselines, including CCA, CFA, and three impute-then-optimize pipelines that differ in the downstream optimization model (imputation+SAA, imputation+KL-DRO, and imputation+Wasserstein-1 DRO) (see Appendix E.3 for further baseline details). Under each configuration and with 100 independent replications, we expose the ensuing optimal ordering quantity to a batch of $10^5$ new data to evaluate their *out-of-sample cost* as well as *reliability*, which is defined as the empirical frequency with which the realized cost exceeds its in-sample counterpart (see Appendix E.5).

**Computational Scalability**  We first report computational results to demonstrate the scalability of the proposed approach. Specifically, the reformulation in Appendix D.1 suggests that the dominant computational burden is the latent support size $|\mathcal{X}||\mathcal{Y}|$, rather than the raw sample size $n$. To isolate the effect of $n$, we first fix the latent support size at 1250 and vary $n$ from $10^2$ to $10^4$ on a logarithmic scale. As shown in Table 1, the mean solution time of the proposed MAR-KL DRO remains nearly stable, increasing only from 2.67 seconds to 3.09 seconds. By contrast, the impute-then-optimize baselines scale directly with the number of completed samples. In particular, Impute+$W_1$-DRO increases from 0.06 seconds to 7.19 seconds over the same range.

*Table 1.* Mean solution time (in seconds) as the sample size varies. The latent support size is fixed at 1250.

| $n$ | 100 | 231 | 533 | 1232 | 2848 | 4328 | 10000 |
|---|---|---|---|---|---|---|---|
| MAR-KL DRO | 2.67 | 2.59 | 2.65 | 2.73 | 2.81 | 2.82 | 3.09 |
| Impute+SAA | 0.08 | 0.12 | 0.16 | 0.28 | 0.60 | 1.08 | 2.37 |
| Impute+KL-DRO | 0.05 | 0.10 | 0.16 | 0.33 | 0.65 | 1.14 | 2.70 |
| Impute+$W_1$-DRO | 0.06 | 0.11 | 0.17 | 0.41 | 0.95 | 2.16 | 7.19 |

We next fix $n = 1000$ and vary the latent support size $|\mathcal{X}||\mathcal{Y}|$ (see Table 2). Without separation, the runtime grows essentially linearly with the support size. To mitigate this burden, we use a restricted-master separation scheme. The constraints associated with $X = \bar{\mathbf{x}}$ are kept explicitly, while the remaining constraints with $X \neq \bar{\mathbf{x}}$ are generated on demand. Given a restricted-master solution $(\lambda^k, \mu^k, \nu^k, \alpha^k, z^k)$, the separation oracle solves, for each $y \in \mathcal{Y}$,

$$\max_{x \in \mathcal{X}^{d_x}, x \neq \bar{\mathbf{x}}} \left\{ \nu^k - (T'_{:,(x,y)})^\top \mu^k \right\}.$$

If the optimal value is positive, the maximizing atom $(x, y)$

yields a violated constraint, which is appended to the master problem. This avoids enumerating the full latent support. From Table 2, we observe that this separation scheme yields a substantial speedup and enables us to solve instances with support size up to $10^6$. For example, this approach solves the instance with $|\mathcal{X}||\mathcal{Y}| = 10^5$ in 17.27 seconds on average, which is faster than solving the full formulation at support size $10^4$.

*Table 2.* Mean solution time (in seconds) as the latent support size varies. Entries marked $> 300$ indicate that the solver did not terminate within 5 minutes.

| Support size | $10^2$ | $10^3$ | $10^4$ | $10^5$ | $10^6$ |
|---|---|---|---|---|---|
| Without separation (s) | 0.28 | 2.95 | 29.79 | $> 300$ | $> 300$ |
| With separation (s) | 0.21 | 0.45 | 3.67 | 17.27 | 151.23 |

**Experiment I: Effect of the Ambiguity Radius**  To calibrate the ambiguity set, we conduct an experiment to observe how the out-of-sample optimal value of (Robust-CSO) evolves under various data size and missingness strength. We study the role of the ambiguity radius using two complementary designs: (i) fixed missingness strength ($\alpha$; see definition in appendix E) and varying data size ($n$), and (ii) fixed data size and varying missingness strength. We report the mean cost, 20%-80% quantile bands, and reliability average over 100 replications

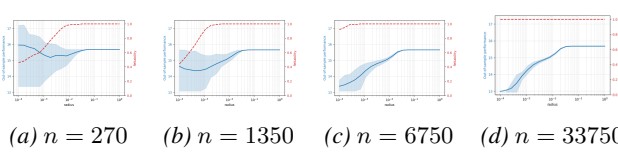

*(a)* $n = 270$  *(b)* $n = 1350$  *(c)* $n = 6750$  *(d)* $n = 33750$

*Figure 2.* Effect of the ambiguity radius on MAR-KL DRO performance under fixed missingness strength $\alpha = 0.8$ across training sample sizes.

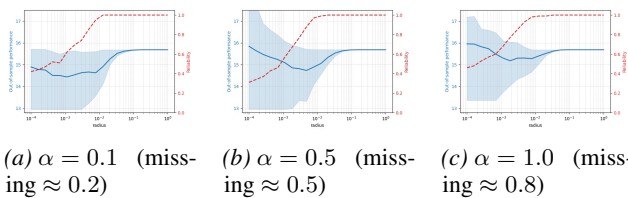

*(a)* $\alpha = 0.1$  (missing $\approx 0.2$)  *(b)* $\alpha = 0.5$  (missing $\approx 0.5$)  *(c)* $\alpha = 1.0$  (missing $\approx 0.8$)

*Figure 3.* Effect of the ambiguity radius with fixed sample size $n = 270$ under varying missingness strength $\alpha$.

Under fixed missingness, larger data sizes allow smaller radii and improved performance, consistent with the theoretical convergence guarantees. Under fixed data size, increasing missingness requires larger radii to maintain reliability, reflecting reduction in effective data size. In both designs, the cost exhibits a U-shaped profile and reliability increases monotonically in the radius, confirming the expected robustness–conservatism trade-off.

**Experiment II: Overall Comparison Across Covariates**
We next aggregate results across all covariates $\bar{x}$, varying the data size between $10^2$ and $10^4$ under fixed missingness strength $\alpha = 0.8$. For methods requiring tuning, radii are selected via a 25%/75% holdout validation split. Relative cost is reported with MAR-KL DRO as the reference.

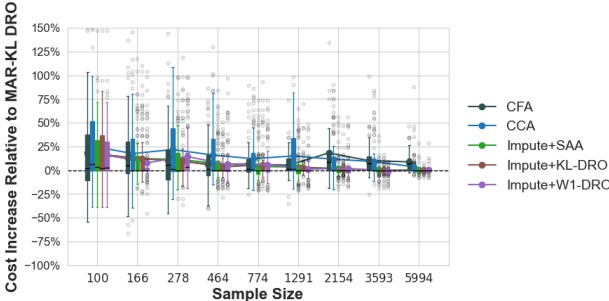

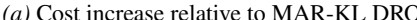

*(a)* Cost increase relative to MAR-KL DRO

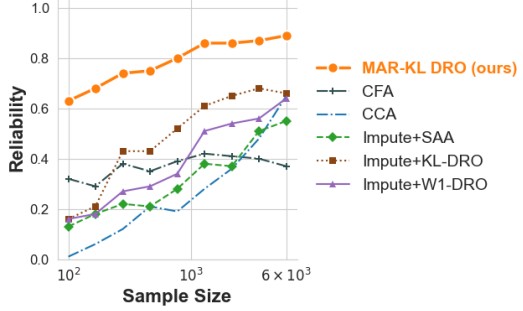

*(b)* Reliability across data sizes

*Figure 4.* Experiment II: Overall performance comparison under varying training sample sizes ($n = 100\text{--}10{,}000$) with fixed missingness strength $\alpha = 0.8$. Values above the zero line in panel (1) indicate higher cost than MAR–KL DRO, i.e., MAR–KL DRO is superior. For a granular breakdown, see Figure 7.

Across nearly the entire data-size range, MAR-KL DRO achieves the lowest (or near-lowest) cost and the highest reliability. Imputation-based methods approach MAR-KL DRO only for large $n$, while CCA and CFA estimators remain unreliable and substantially suboptimal. These findings confirm that MAR-KL DRO provides both statistical stability and practical performance advantages. In Appendix E.6, we report more results and details of this experiment, including a granular breakdown of Figure 4 to compare MAR-KL DRO with each benchmark approach (see Figure 7).

**Experiment III: Sensitivity Analyses** We further assess sensitivity by varying one factor at a time while conditioning on a fixed representative context $\bar{x}$ (results are robust to this choice). We report the results using radii selected via a 25%/75% holdout.

We vary $n$ on a logarithmic grid between $10^2$ and $10^4$ under fixed $\alpha = 0.8$, and report the mean out-of-sample cost with

$\pm 1$ standard deviation bands (see Figure 5).

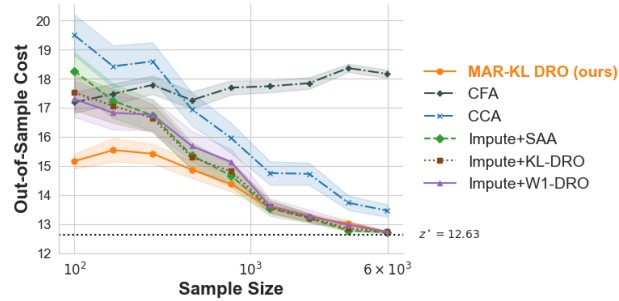

*(a)* Out-of-sample cost (holdout)

*Figure 5.* Design 1: Sensitivity to data size under fixed missingness strength $\alpha = 0.8$.

Then, we fix $n = 270$ and vary $\alpha \in [0.1, 1.0]$, reporting the same metrics (see Figure 6).

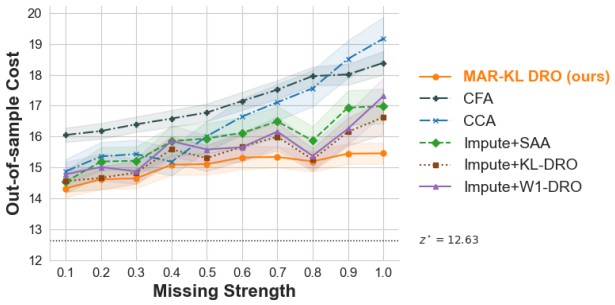

*(a)* Out-of-sample cost (holdout)

*Figure 6.* Design 2: Sensitivity to missingness strength with fixed sample size $n = 270$.

Across both designs and tuning regimes, MAR-KL DRO remains reliable and cost-competitive, while the benchmarks degrade more rapidly as the effective information decreases, either due to smaller data sizes or higher missingness strength. In Appendix E.7, we additionally report the out-of-sample cost under oracle-selected radii, as well as reliability under both tuning settings.

## Impact Statement

This paper proposes a robust framework for contextual decision-making with missing covariates. By explicitly modeling uncertainty arising from incomplete data, the approach aims to improve the reliability of data-driven decisions in imperfect-data regimes. There are many potential societal consequences of our work, none of which we feel must be specifically highlighted here.

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

# A. Notations

*Table 3.* Summary of notation (general setting).

| Symbol | Description |
|---|---|
| $X \in \mathcal{X}$ | Covariate (context) of dimension $d_x$, possibly non-numeric (e.g., categorical features). |
| $Y \in \mathcal{Y}$ | Numerical outcome variable taking values in $\mathbb{R}^{d_y}$. |
| $z \in \mathcal{Z}$ | Decision variable; $\mathcal{Z}$ denotes the feasible set. |
| $c(z, y)$ | Cost function incurred when decision $z$ is taken and outcome $y$ is realized. |
| $\bar{x}$ | Target covariate at the decision time. |
| $(X_o, X_m)$ | Decomposition of covariates into always observed and potentially-missing components. |
| $M$ | Missingness mask |
| $\tilde{X}_m$ | Observed covariates with missing entries. |
| $\mathcal{Q}_{\mathrm{MAR}}$ | Class of missing-at-random (MAR) missingness mechanisms. |
| $\mathcal{H}(\mathbb{P}_{XY}, q)$ | Distribution of observables induced by latent distribution $\mathbb{P}_{XY}$ and missingness mechanism $q$. |
| $\mathcal{A}(\tilde{x}_m, m)$ | Set of latent covariates compatible with observation $(\tilde{x}_m, m)$. |
| $\widehat{\mathbb{P}}_n^{\mathrm{obs}}$ | Empirical distribution of observed data. |

*Table 4.* Notation for ambiguity sets and distributionally robust optimization.

| Symbol | Description |
|---|---|
| $\mathcal{P}_n(\varepsilon)$ | Latent ambiguity set with radius $\varepsilon$, with concrete forms $\mathcal{P}_{\mathrm{KL}}(\varepsilon)$ and $\mathcal{P}_{\mathrm{PKL}}(\varepsilon, \delta)$, the latter smoothed via an LP radius $\delta$. |
| $\mathcal{D}(\gamma)$ | Equivalent ambiguity set obtained via transformation of $\mathcal{P}_{\mathrm{KL}}(\varepsilon)$, with radius $\gamma$ derived from $\varepsilon$. |
| $\mathcal{K}(\gamma, \delta)$ | Tractable ambiguity set induced by the $\mathcal{P}_{\mathrm{PKL}}(\varepsilon, \delta)$, with radius $\gamma$ derived from $\varepsilon$. |
| $\hat{\Phi}_n(\bar{x})$ | Optimal value of (Robust-CSO) at context $\bar{x}$. |
| $\hat{z}_n(\bar{x})$ | Optimal solution of (Robust-CSO) at context $\bar{x}$. |
| $\Phi^\star(\bar{x})$ | Optimal value of (CSO) under the true distribution. |
| $D_{\mathrm{KL}}(P\|Q)$ | Kullback–Leibler divergence. |
| $d_{\mathrm{LP}}(P, Q)$ | Lévy–Prokhorov metric. |
| $D_{\mathrm{BinPKL}}(\mu; P, \delta)$ | LP-projected KL divergence. |
| $\mathcal{B}_{\mathrm{LP}}(P_0, \delta)$ | Lévy–Prokhorov ball centered at $P_0$ with radius $\delta$. |

*Table 5.* Notation for reformulations and statistical guarantees.

| Symbol | Description |
|---|---|
| $\eta$ | with confidence level $1 - \eta$, the worst-case conditional cost acts as an upper confidence bound on the true conditional cost |
| $\Delta(\mathcal{X} \times \mathcal{Y})$ | Probability simplex over $\mathcal{X} \times \mathcal{Y}$. |
| $T$ | Compatibility indicator matrix. |
| $\mathbf{c}(z)_j = c(z, y_j)\mathbb{I}\{x_j = \bar{x}\}$ | Cost vector used to express conditional expectation. |
| $d_j = \mathbb{I}\{x_j = \bar{x}\}$ | Indicator vector selecting atoms with covariate $\bar{x}$. |
| $(v, u, t)$ | Lifted variables in convex reformulation of the inner problem. |
| $y_\infty^{\mathcal{N}}(z)$ | Worst-case outcome maximizing $c(z, y)$. |

# B. Additional Literature Review

## B.1. Missing Data Treatment

In this section, we provide a review of existing strategies for handling missing covariates. We discuss the missingness mechanism in the main text, but include a brief introduction here for completeness: Missing data mechanisms are commonly classified into three categories (Rubin, 1976; Little & Rubin, 2019).

- Missing Completely at Random (MCAR): The probability of a value being missing is independent of both observed and unobserved data. This is the simplest missingness assumption, yet it is rarely satisfied in real-world datasets (Chen & Cummings, 2025). When MCAR holds, several simple approaches—such as deletion strategies and naive imputation—can be justified.

- Missing at Random (MAR): The missingness depends only on the observed data. For example, in a medical study, older patients might be less likely to complete a certain physical test, but this is explainable by their observed age. In such cases, naive analysis can yield biased estimates of the full-data distribution due to covariate shift, and may even bias the conditional distribution $\mathbb{P}(Y \mid X)$ when the missingness depends on the outcome $Y$. Fortunately, observable data can generally identify the distribution of interest under the mild ignorability conditions described by (Rubin, 1976), enabling estimators with valid asymptotic properties.

- Missing Not at Random (MNAR): The missingness depends on the values of the missing variables themselves (e.g., high-income individuals refusing to report their salary). This is the trickiest one that addressing MNAR typically requires additional structural assumptions or external information(Miao & Tchetgen, 2018); in their absence, the model can be observationally equivalent to an MAR model (Molenberghs et al., 2008) .

### B.1.1. DELETION

The most straightforward approach is deletion. A common example is Complete-Case Analysis (CCA) (Little & Rubin, 2019), which discards any observation containing at least one missing value. Although CCA is unbiased under MCAR, it substantially reduces the effective sample size and can lead to severe bias when the missingness mechanism is Missing at Random (MAR) or Missing Not at Random (MNAR) (Little et al., 2024). Another deletion-based approach, which we refer to as Complete-Feature Analysis (CFA), removes entire features with missing entries, often resulting in the loss of critical predictive information for downstream optimization tasks.

More broadly, deletion strategies can become practically infeasible when the missingness rate is moderate to high. In such regimes, CCA may discard the majority of observations (if not all) and CFA can remove most informative covariates thus reducing the problem to a non-contextual one. As a result, deletion-based methods often fail operationally, as the remaining data are often insufficient to support meaningful decision-making.

### B.1.2. SIMPLE POINT-IMPUTATION

Simple point-imputation methods are also widely used as baselines due to their ease of implementation, though they also frequently introduce bias or efficiency loss under MAR or MNAR. Common heuristics include mean, median or mode imputation, which replace missing values with summary statistics. Other heuristics include hot-deck and cold-deck imputation (Little & Rubin, 2019), which substitute missing entries with observed values from similar "donor" cases within the same dataset or from external datasets, respectively. While popular, such methods can attenuate correlations in multivariate settings (Enders, 2022), and can also introduce bias when data is not MCAR (Chen & Cummings, 2025).

### B.1.3. OTHER IMPUTATION METHODS

While simple imputation methods have well-known limitations, a broad class of more sophisticated imputation approaches has been developed to predict missing covariates from observational data. Depending on the modeling assumptions, the imputation distribution may be specified parametrically, semiparametrically, or nonparametrically, including maximum-likelihood-based methods (Little & Rubin, 2019), $k$-nearest neighbors approaches (Troyanskaya et al., 2001), and random-forest-based procedures such as MissForest (Stekhoven & Bühlmann, 2012). The completed datasets are then constructed by drawing from, or approximating, the estimated conditional distribution of the missing covariates. These methods typically rely on structural assumptions about the data-generating process and commonly require the missingness mechanism to

satisfy the Missing at Random (MAR) condition (Chen & Cummings, 2025). When missingness is MNAR, one typically requires additional assumptions, such as selection models or pattern-mixture models, to identify the latent statistical model. In addition, many imputation procedures—particularly single-imputation methods—either ignore or inadequately represent uncertainty in the imputation step. As a result, they can induce substantial residual variability.

Multiple imputation (MI) extends this idea by repeatedly sampling $X_m$ from an estimated conditional distribution to generate several completed datasets (Van Buuren & Groothuis-Oudshoorn, 2011; Carpenter et al., 2023). Each dataset reflects a different plausible realization of the missing covariates, and inference is aggregated to account for between-imputation variability. Under correct specification and MAR assumptions, MI provides valid inference for many estimands. However, its performance critically depends on the correctness of the imputation model, and the Rubin-style combining rules are primarily justified for estimation and hypothesis testing, rather than for decision-focused objectives.

More recently, a growing body of methods has been proposed to impute missing covariates using flexible generative modeling techniques, including variational autoencoders and GAN-based approaches (Mattei & Frellsen, 2019; Yoon et al., 2018); some works formulate missing data imputation as a distribution-matching problem using optimal transport distances (Muzellec et al., 2020). These methods can capture complex dependencies and scale to high-dimensional settings, but they typically lack finite-sample guarantees and offer limited theoretical interpretability.

### B.1.4. INVERSE PROBABILITY WEIGHTED METHODS

Inverse probability weighting (IPW) addresses missing covariates by reweighting complete cases to represent the target population. The central idea is to explicitly model the missingness or response mechanism and to construct observation weights as the inverse of the estimated response probabilities. Under correct specification of the missingness model and appropriate regularity conditions—most commonly under the Missing at Random (MAR) assumption—IPW yields consistent estimators for a wide range of parameters (Seaman & White, 2013; Chesnaye et al., 2022).

Despite its conceptual simplicity, IPW is known to suffer from practical and statistical challenges. In finite samples, estimated response probabilities close to zero can lead to extreme weights, resulting in unstable estimators and inflated variance. This issue is closely related to violations, or near-violations, of the positivity assumption. To mitigate these effects, stabilized or truncated weights are commonly employed. More advanced variants, such as augmented inverse probability weighting (AIPW), combine weighting with an outcome regression model to improve efficiency and provide a degree of robustness against misspecification of either component (Zhong et al., 2021).

While IPW methods are well understood under MAR, extending them to Missing Not at Random (MNAR) settings is substantially more challenging. In MNAR regimes, the missingness mechanism depends on unobserved covariates, rendering the full-data distribution non-identifiable without additional assumptions. A substantial statistical literature has therefore focused on identification and inference under MNAR by introducing auxiliary structures (Miao & Tchetgen, 2018). However, the validity of such methods hinges on strong structural assumptions that are often difficult to verify in practice, and inference can be sensitive to violations of these assumptions.

## C. Technical Proofs

### C.1. Proof of Proposition 3.2

*Proof.* We denote the two sets of interest as follows:

$$A := \left\{ \mathbb{P}_{XY} \in \Delta(\mathcal{X} \times \mathcal{Y}) \,\middle|\, \exists q \in \mathcal{Q}_{\text{MAR}} \text{ s.t. } D_{\text{KL}}\left(\widehat{\mathbb{P}}_n^{\text{obs}} \,\|\, \text{H}(\mathbb{P}_{XY}, q)\right) \leq \epsilon \right\},$$

$$B := \left\{ \mathbb{P}_{XY} \in \Delta(\mathcal{X} \times \mathcal{Y}) \,\middle|\, \frac{1}{n} \sum_{i=1}^{n} \log\left( \sum_{x_{m,i} \in \mathcal{A}(\tilde{x}_{m,i}, m_i)} \mathbb{P}_{XY}(x_{o,i}, x_{m,i}, y_i) \right) \geq \gamma \right\},$$

where $\gamma$ is defined as in the main text in Proposition 3.2. We proceed by showing mutual inclusion that $A \subseteq B$ and $B \subseteq A$.

**(1) $A \subseteq B$.** Suppose $\mathbb{P}_{XY} \in A$. Then by definition there exists $q \in \mathcal{Q}_{\text{MAR}}$ such that $D_{\text{KL}}\left(\widehat{\mathbb{P}}_n^{\text{obs}} \,\|\, \text{H}(\mathbb{P}_{XY}, q)\right) \leq \epsilon$. Since $q \in \mathcal{Q}_{\text{MAR}}$, the model-induced observable distribution is:

$$\text{H}(\mathbb{P}_{XY}, q)(\tilde{x}, y, m) = q(m \mid x_o, y) \sum_{x_m \in \mathcal{A}(\tilde{x}_m)} \mathbb{P}_{XY}(x_o, x_m, y),$$

As KL divergence is finite, we can expand the KL divergence as

$$D_{\text{KL}}\left(\widehat{\mathbb{P}}_n^{\text{obs}} \,\|\, \text{H}(\mathbb{P}_{XY}, q)\right) = \sum_{\xi \in \widehat{\Sigma}} \frac{n_\xi}{n} \log \frac{n_\xi/n}{\text{H}(\mathbb{P}_{XY}, q)(\xi)}$$

$$= \sum_{\xi \in \widehat{\Sigma}} \frac{n_\xi}{n} \log\left(\frac{n_\xi}{n}\right) - \frac{1}{n} \sum_{i=1}^{n} \log q(m_i \mid x_{o,i}, y_i) - \frac{1}{n} \sum_{i=1}^{n} \log\left( \sum_{x_{m,i} \in \mathcal{A}(\tilde{x}_{m,i})} \mathbb{P}_{XY}(x_{o,i}, x_{m,i}, y_i) \right)$$

Let $L(q) := -\frac{1}{n} \sum_{i=1}^{n} \log q(m_i \mid x_{o,i}, y_i)$. From feasibility $D_{\text{KL}} \leq \epsilon$ we know

$$-\frac{1}{n} \sum_{i=1}^{n} \log\left( \sum_{x_{m,i} \in \mathcal{A}(\tilde{x}_{m,i})} \mathbb{P}_{XY}(x_{o,i}, x_{m,i}, y_i) \right) \leq \epsilon - \sum_{\xi \in \widehat{\Sigma}} \frac{n_\xi}{n} \log\left(\frac{n_\xi}{n}\right) - L(q)$$

Since $L(q) \geq \inf_{q' \in \mathcal{Q}_{\text{MAR}}} L(q') = -\frac{1}{n} \sum_{i=1}^{n} \log \widehat{\mathbb{P}}(m_i \mid x_{o,i}, y_i)$, and substitute the definition of $\gamma$ back, we obtain

$$\frac{1}{n} \sum_{i=1}^{n} \log\left( \sum_{x_{m,i} \in \mathcal{A}(\tilde{x}_{m,i})} \mathbb{P}_{XY}(x_{o,i}, x_{m,i}, y_i) \right) \geq \gamma$$

Hence $\mathbb{P}_{XY} \in B$. Thus $A \subseteq B$.

**(2) $B \subseteq A$** Suppose $\mathbb{P}_{XY} \in B$. Then

$$\frac{1}{n} \sum_{i=1}^{n} \log\left( \sum_{x_{m,i} \in \mathcal{A}(\tilde{x}_{m,i})} \mathbb{P}_{XY}(x_{o,i}, x_{m,i}, y_i) \right) \geq \gamma$$

where $\gamma \in \mathbb{R}$ is finite.

Since $\gamma$ is finite, for every $i = 1, \ldots, n$, $\sum_{x_{m,i} \in \mathcal{A}(\tilde{x}_{m,i})} \mathbb{P}_{XY}(x_{o,i}, x_{m,i}, y_i) > 0$, because otherwise for some $i$ the corresponding logarithm would be $\log 0 = -\infty$. In particular, all the inner sums in the logarithms are strictly positive.

Next, consider $\hat{q}(m \mid x_o, y) := \widehat{\mathbb{P}}(m \mid x_o, y)$, when $(x_o, y)$ is observed in the sample. For any $(x_o, y)$ *not* observed in the sample, choose an arbitrary valid distribution on $\mathcal{M}$. Note that for each observed triple $(x_{o,i}, y_i, m_i)$, we have $\widehat{\mathbb{P}}(m_i \mid x_{o,i}, y_i) > 0$ because the empirical conditional places strictly positive mass on any actually observed pattern. Combining these two observations, for every $i$ we obtain

$$\text{H}(\mathbb{P}_{XY}, \hat{q})(\tilde{x}_i, y_i, m_i) = \hat{q}(m_i \mid x_{o,i}, y_i) \sum_{x_{m,i} \in \mathcal{A}(\tilde{x}_{m,i})} \mathbb{P}_{XY}(x_{o,i}, x_{m,i}, y_i) > 0$$

Hence $D_{\mathrm{KL}}(\widehat{\mathbb{P}}_n^{\mathrm{obs}} \,\|\, \mathrm{H}(\mathbb{P}_{XY}, \hat{q}))$ is finite, and the usual finite KL expansion is valid. Substituting $\hat{q}$ into the KL decomposition,

$$D_{\mathrm{KL}} = \sum_{\xi \in \widehat{\Xi}} \frac{n_\xi}{n} \log\left(\frac{n_\xi}{n}\right) - \left(\frac{1}{n} \sum_{i=1}^{n} \log \widehat{\mathbb{P}}(m_i \mid x_{o,i}, y_i)\right) - \frac{1}{n} \sum_{i=1}^{n} \log\left(\sum_{x_{m,i} \in \mathcal{A}(\tilde{x}_{m,i})} \mathbb{P}_{XY}(x_{o,i}, x_{m,i}, y_i)\right)$$

Using the assumed inequality defining set $B$ and definition of $\gamma$, this yields $D_{\mathrm{KL}} \leq \epsilon$. Thus there exists a feasible $q$ such that the KL divergence constraint holds, implying $\mathbb{P}_{XY} \in A$. Therefore $B \subseteq A$.

**(3)** Thus $A = B$. Finally, the objective $\mathbb{E}[c(z, Y) \mid X = \bar{\mathbf{x}}]$ depends only on the conditional law induced by $\mathbb{P}_{XY}$; thus, the optimization over $\mathbb{P}_{XY} \in \mathcal{P}_{\mathrm{KL}}(\epsilon)$ is equivalent to the optimization over $p \in \mathcal{D}(\gamma)$.

$\square$

### C.2. Proof of Proposition 3.3

To prove Proposition 3.3, we first recall the following lemma:

**Lemma C.1** (Weierstrass Extreme-Value Theorem; see, e.g., Rudin (1976)). *Let $K$ be a nonempty compact set and let $f : K \to \mathbb{R}$. If $f$ is upper semicontinuous, then there exists $x^{\mathrm{max}} \in K$ such that $f(x^{\mathrm{max}}) = \sup_{x \in K} f(x)$. If $f$ is lower semicontinuous, then there exists $x^{\mathrm{min}} \in K$ such that $f(x^{\mathrm{min}}) = \inf_{x \in K} f(x)$*

*In particular, if $f$ is continuous, then it attains both its maximum and minimum on $K$.*

We are now ready to prove the main result.

*Proof.* Proof of Proposition 3.3. Fix a feasible likelihood threshold $\gamma$, so that $\mathcal{D}(\gamma) \neq \emptyset$. (As a side note, define $L_1^\star := \sup\left\{\frac{1}{n} \sum_{i=1}^{n} \log\left((Tp)_i\right) : p \in \Delta(\mathcal{X} \times \mathcal{Y})\right\}$. This quantity can be computed via a convex program, and $\gamma \leq \gamma_{\mathrm{max}} := L_1^\star$ guarantees feasibility.) By assumption, there exists $\varepsilon_1 > 0$ such that $p(\bar{\mathbf{x}}) \geq \varepsilon_1$ for all $p \in \mathcal{D}(\gamma)$.

For fixed $z \in \mathcal{Z}$, write

$$F(p; z) := \mathbb{E}_p[c(z, Y) \mid X = \bar{\mathbf{x}}] = \frac{\sum_{y \in \mathcal{Y}} p(\bar{\mathbf{x}}, y) c(z, y)}{\sum_{y \in \mathcal{Y}} p(\bar{\mathbf{x}}, y)}$$

**(1) Compactness of $\mathcal{D}(\gamma)$.** The probability simplex $\Delta(\mathcal{X} \times \mathcal{Y}) \subset \mathbb{R}^{|\mathcal{X}||\mathcal{Y}|}$ is compact. The log-likelihood map $\ell(p) := \frac{1}{n} \sum_{i=1}^{n} \log\left((Tp)_i\right)$ is upper semicontinuous as an extended-real-valued function. (Indeed, the map $p \mapsto (Tp)_i$ is linear and hence continuous. With the convention $\log 0 = -\infty$, the function $t \mapsto \log t$ is upper semicontinuous on $[0, \infty)$. Hence each map $p \mapsto \log((Tp)_i)$ is upper semicontinuous, and so $\ell(p)$ is upper semicontinuous as a finite sum of upper semicontinuous functions ) Consequently, for any finite $\gamma$, its super-level set is closed. Therefore $\mathcal{D}(\gamma)$ is compact as a closed subset of the compact simplex.

**(2) Attainment of the inner supremum.** For fixed $z$, we first show $F(p)$ is continuous over $\mathcal{D}(\gamma)$ under the standard Euclidean topology (which coincides with the weak topology on the probability simplex in the finite discrete case).

Let $\{p_n\} \subset \mathcal{D}(\gamma)$ be any sequence converging to $p \in \mathcal{D}(\gamma)$. Since the support $\mathcal{X} \times \mathcal{Y}$ is finite, pointwise convergence $p_n(x, y) \to p(x, y)$ for all $(x, y)$ implies that both the numerator and denominator of $F(p_n)$ converge:

$$\sum_y p_n(\bar{\mathbf{x}}, y) \, c(z, y) \to \sum_y p(\bar{\mathbf{x}}, y) \, c(z, y), \quad \text{and} \quad p_n(\bar{\mathbf{x}}) \to p(\bar{\mathbf{x}}).$$

Since $p_n(\bar{\mathbf{x}}) \geq \varepsilon_1 > 0$ for all $n$, the denominator is uniformly bounded away from zero. Therefore, by the continuity of the quotient of convergent sequences with strictly positive denominators, we conclude $F(p_n) \to F(p)$. Hence, $F$ is continuous over $\mathcal{D}(\gamma)$.

By Lemma C.1, for each fixed $z$, the continuous function $F(\cdot, z)$ attains its maximum over the nonempty compact set $\mathcal{D}(\gamma)$. Thus there exists $p_z^{\mathrm{wc}} \in \mathcal{D}(\gamma)$ such that $\sup_{p \in \mathcal{D}(\gamma)} F(p, z) = F(p_z^{\mathrm{wc}}, z)$

This ensures that the worst–case expectation is attained for each decision $z$. We thus denote

$$\Psi(z) := \sup_{p \in \mathcal{D}(\gamma)} F(p, z)$$

**(3) Lower semicontinuity of** $\Psi$**.** We now prove that the worst-case cost $\Psi(z)$ is lower semicontinuous on $\mathcal{Z}$. Let $\{z_k\}_{k=1}^{\infty} \subset \mathcal{Z}$ be a sequence converging to $z \in \mathcal{Z}$. Fix an arbitrary distribution $p \in \mathcal{D}(\gamma)$. The objective function $F(p, z)$ is a linear combination of the costs $c(z, y)$ with non-negative weights:

$$F(p, z) = \sum_{y \in \mathcal{Y}} \frac{p(\bar{\mathbf{x}}, y)}{p(\bar{\mathbf{x}})} c(z, y)$$

By assumption 2.1 , $c(\cdot, y)$ is lower semicontinuous for every $y$. Since a non-negative linear combination of LSC functions is LSC, we have:

$$\liminf_{k \to \infty} F(p, z_k) \geq F(p, z)$$

By the definition of the worst-case value, for every $k$, we have $\Psi(z_k) = \sup_{\tilde{p} \in \mathcal{D}(\gamma)} F(\tilde{p}, z_k) \geq F(p, z_k)$. Taking the limit inferior as $k \to \infty$ on both sides :

$$\liminf_{k \to \infty} \Psi(z_k) \geq \liminf_{k \to \infty} F(p, z_k) \geq F(p, z)$$

Since the inequality $\liminf_{k \to \infty} \Psi(z_k) \geq F(p, z)$ holds for *any arbitrary* $p \in \mathcal{D}(\gamma)$, it must also hold for the supremum over all such $p$:

$$\liminf_{k \to \infty} \Psi(z_k) \geq \sup_{p \in \mathcal{D}(\gamma)} F(p, z) = \Psi(z)$$

Thus $\Psi$ is lower semicontinuous on $\mathcal{Z}$.

**(4) Attainment of the outer minimum.** Recall that $\mathcal{Z}$ is nonempty and compact and $\Psi$ is lower semicontinuous, Lemma C.1 implies that there exists $z^{\star} \in \mathcal{Z}$ such that $\Psi(z^{\star}) = \inf_{z \in \mathcal{Z}} \Psi(z)$ Together with the attainment of the inner supremum for every $z$, this proves that (Robust-CSO) attains both its worst-case distribution and an optimal solution.

$\square$

### C.3. Proof of Proposition 3.5 and Theorem 3.6

*Proof.* Proof of Proposition 3.5.

Given the notation in the main text, the inner robust optimization problem (for fixed $z$) can be written as the following linear-fractional form:

$$\max_{p \in \mathbb{R}^{|\mathcal{S}|}, \tilde{p} \in \mathbb{R}^n} \quad \frac{c^\top p}{d^\top p}$$

$$\text{s.t.} \quad \frac{1}{n} \sum_{i=1}^{n} \log(\tilde{p}_i) \geq \gamma, \tag{8}$$

$$\mathbf{e}^\top p = 1, \quad p \succeq 0,$$

$$\tilde{p} = Tp,$$

where $\mathbf{e}$ is the all-ones vector.

Define the new variables:

$$v := \frac{p}{d^\top p}, \quad t := \frac{1}{d^\top p}, \quad u := \frac{\tilde{p}}{d^\top p} = \frac{Tp}{d^\top p}.$$

Consider the following optimization problem:

$$\max_{v, u, t} \quad c^\top v$$

$$\text{s.t.} \quad n\gamma t + \sum_{i=1}^{n} t \log\left(\frac{t}{u_i}\right) \leq 0,$$

$$u = Tv, \tag{9}$$

$$\mathbf{e}^\top v = t,$$

$$d^\top v = 1,$$

$$t \geq 0, \quad v \succeq 0, \quad u \succeq 0.$$

We aim to prove that (8) and (9) are equivalent :

**(1)** Take any feasible $(p, \tilde{p})$ of (8) , the constraint $\frac{1}{n} \sum_{i=1}^{n} \log(\tilde{p}_i) \geq \gamma$ with finite $\gamma$ implies $\tilde{p}_i > 0$ for all $i$, hence $\tilde{p} \succ 0$. We also assume from Proposition 3.3 that $d^\top p > 0$. Define associated $(t, u, v)$. Then $t > 0$, and $v \succeq 0$ because $p \succeq 0$. Also $u \succeq 0$ because $\tilde{p} \succ 0$.

The linear constraints transform as follows:

$$u = t\tilde{p} = t(Tp) = T(tp) = Tv,$$

$$\mathbf{e}^\top v = \mathbf{e}^\top (tp) = t(\mathbf{e}^\top p) = t,$$

$$d^\top v = d^\top (tp) = t(d^\top p) = 1.$$

Since $\tilde{p}_i = u_i/t$ for all $i$, the original constraint implies

$$\sum_{i=1}^{n} \log\left(\frac{u_i}{t}\right) \geq n\gamma.$$

Rearranging yields

$$-n\gamma \leq \sum_{i=1}^{n} \log\left(\frac{u_i}{t}\right) = -\sum_{i=1}^{n} \log\left(\frac{t}{u_i}\right),$$

hence

$$n\gamma + \sum_{i=1}^{n} \log\left(\frac{t}{u_i}\right) \leq 0.$$

Multiplying by $t > 0$ gives exactly the nonlinear constraint in (9):

$$n\gamma t + \sum_{i=1}^{n} t\log\left(\frac{t}{u_i}\right) \leq 0.$$

Hence $(v, u, t)$ satisfies all constraints of (9).

Finally, the objective is preserved:

$$c^\top v = c^\top \left(\frac{p}{d^\top p}\right) = \frac{c^\top p}{d^\top p}.$$

**(2)** Take any feasible $(v, u, t)$ of (9) . We first note that $t > 0$. Although the constraint in (9) is written as $t \geq 0$, any feasible solution $(v, u, t)$ must in fact satisfy $t > 0$. Indeed, suppose by contradiction that $t = 0$ for some feasible $(v, u, t)$. Then the constraint $\mathbf{e}^\top v = t$ together with $v \succeq 0$ implies $v = 0$. However, this contradicts the constraint $d^\top v = 1$. Hence no feasible solution of (9) can satisfy $t = 0$, and necessarily $t > 0$ holds for all feasible solutions.

Then define $p := \frac{v}{t}, \tilde{p} := \frac{u}{t}$.. It follows that $p \succeq 0$ and $\mathbf{e}^\top p = \frac{\mathbf{e}^\top v}{t} = \frac{t}{t} = 1$,

$$\tilde{p} = \frac{u}{t} = \frac{Tv}{t} = T\left(\frac{v}{t}\right) = Tp.$$

Moreover

$$d^\top p = d^\top \left(\frac{v}{t}\right) = \frac{d^\top v}{t} = \frac{1}{t} > 0,$$

The nonlinear constraint in (9):

$$n\gamma t + \sum_{i=1}^{n} t\log\left(\frac{t}{u_i}\right) \leq 0.$$

Divide by $t > 0$ to obtain

$$n\gamma + \sum_{i=1}^{n} \log\left(\frac{t}{u_i}\right) \leq 0 \quad \Longleftrightarrow \quad \sum_{i=1}^{n} \log\left(\frac{u_i}{t}\right) \geq n\gamma.$$

Since $\tilde{p}_i = u_i/t$, this is exactly $\frac{1}{n}\sum_{i=1}^{n} \log(\tilde{p}_i) \geq \gamma$, which is the nonlinear constraint of (8). In all, $(p, \tilde{p})$ is feasible for (8). Objective preservation follows immediately:

$$\frac{c^\top p}{d^\top p} = \frac{c^\top(v/t)}{d^\top(v/t)} = \frac{c^\top v}{d^\top v} = c^\top v,$$

where we used $d^\top v = 1$.

In summary, the two mappings are inverses of each other preserve feasibility, and preserve objective value. Therefore the optimal values coincide and optimal solutions can be transferred between formulations as stated.

$\square$

**Observation.** Problem (9) is a convex optimization problem:

- The objective $c^\top v$ is linear.

- The constraint $n\gamma t + \sum_{i=1}^{n} t \log(t/u_i) \leq 0$ is convex in $(u, t)$ because $t \log(t/u)$ is jointly convex on $\mathbb{R}_{++} \times \mathbb{R}_{++}$.

- The remaining constraints are affine or conic.

*Proof.* Proof of Theorem 3.6.

We establish the result by deriving the dual of the convex reformulation in Proposition 3.5. The inner problem is a convex program with a strictly feasible point under attainability condition by construction. Therefore, Slater's condition holds, and strong duality applies.

We proceed to derive the dual problem. Let $\lambda \geq 0$, $\mu \in \mathbb{R}^n$, $\nu \in \mathbb{R}$, and $\alpha \in \mathbb{R}$ denote the dual variables corresponding to the constraints:

$$\text{perspective constraint:} \qquad n\gamma t + \sum_{i=1}^{n} t \log\left(\frac{t}{u_i}\right) \leq 0,$$

$$\text{linear equality:} \qquad u = Tv, \quad \mathbf{e}^\top v = t, \quad d^\top v = 1.$$

The Lagrangian is given by:

$$L(v, u, t; \lambda, \mu, \nu, \alpha) = c^\top v - \lambda\left(n\gamma t + \sum_{i=1}^{n} t \log\left(\frac{t}{u_i}\right)\right) + \mu^\top(u - Tv) + \nu(\mathbf{e}^\top v - t) + \alpha(d^\top v - 1).$$

The dual function is obtained by maximizing the Lagrangian over $v \succeq 0$, $u \succeq 0$, $t \geq 0$:

$$g(\lambda, \mu, \nu, \alpha) = \sup_{v \succeq 0, u \succeq 0, t \geq 0} L(v, u, t; \lambda, \mu, \nu, \alpha)$$

$$= \sup_{v,u,t}\left\{c^\top v - \lambda\left(n\gamma t + \sum_{i=1}^{n} t \log\left(\frac{t}{u_i}\right)\right) + \mu^\top(u - Tv) + \nu(\mathbf{e}^\top v - t) + \alpha(d^\top v - 1)\right\}$$

$$= \sup_{v,u,t}\left\{-\lambda\left(n\gamma t + \sum_{i=1}^{n} t \log\left(\frac{t}{u_i}\right)\right) + \mu^\top u + \left(c - T^\top\mu + \nu\mathbf{e} + \alpha d\right)^\top v - \nu t - \alpha\right\}.$$

The supremum is finite only if the coefficient of $v$ vanishes, i.e.,

$$c - T^\top\mu + \nu\mathbf{e} + \alpha d \preceq 0,$$

and the remaining part gives:

$$g(\lambda, \mu, \nu, \alpha) = \left[-\lambda\left(n\gamma + \sum_{i=1}^{n} \log\left(\frac{\lambda}{\mu_i}\right)\right) - \nu\right] t + \mu^T u - \alpha,$$

where the supremum over $t > 0$ and $u_i > 0$ is explicitly computed. Fix $\lambda \geq 0$ and $t \geq 0$. Define

$$f(u) := -\lambda \sum_{i=1}^{n} t \log\left(\frac{t}{u_i}\right) + \mu^T u,$$

with partial derivative

$$f'(u_i) = \lambda \frac{t}{u_i} + \mu_i.$$

If any $\mu_i > 0$, then $f(u_i)$ is increasing, and the value diverges as $u_i \to \infty$. Therefore, we must impose: $\mu \preceq 0$. The maximum is then attained at $u_i^* = -\frac{\lambda t}{\mu_i}$.

Substituting $u_i^*$ back, the term becomes:

$$-\lambda \left( n\gamma t + \sum_{i=1}^{n} t \log\left(\frac{t}{u_i^*}\right) \right) + \mu^T u^* - \nu t - \alpha$$

$$= -\lambda \left( n\gamma t + \sum_{i=1}^{n} t \log\left(\frac{t}{-\frac{\lambda t}{\mu_i}}\right) \right) + \sum_{i=1}^{n} \mu_i \cdot \left(-\frac{\lambda t}{\mu_i}\right) - \nu t - \alpha$$

$$= -\lambda \left( n(1+\gamma)t + \sum_{i=1}^{n} \log\left(-\frac{\mu_i}{\lambda}\right) t \right) - \nu t - \alpha.$$

To ensure finiteness of the dual function, we must impose:

$$-\lambda \left( n(1+\gamma) + \sum_{i=1}^{n} \log\left(-\frac{\mu_i}{\lambda}\right) \right) - \nu \leq 0.$$

Substituting into the dual problem, we obtain:

$$\min_{\lambda,\mu,\nu,\alpha} \quad -\alpha$$
$$\text{s.t.} \quad c(z) - T^\top \mu + \nu \mathbf{e} + \alpha d \preceq 0,$$
$$\lambda \left( n(1+\gamma) + \sum_{i=1}^{n} \log\left(\frac{-\mu_i}{\lambda}\right) \right) + \nu \geq 0,$$
$$\mu \leq 0, \quad \lambda \geq 0.$$

We now combine the dual with the outer minimization over $z$. Assuming the cost function $c(z, y)$ is convex in $z$ for every $y$, we obtain the result.

$\square$

## C.4. Proof of Theorem 3.7

We first give the following lemma :

**Lemma C.2.** *Let $\widehat{\mathbb{P}}_n$ be the empirical distribution based on $n$ i.i.d. realizations on a finite support $\mathcal{S} := \mathcal{X} \times \mathcal{Y} \times \mathcal{M}$, generated from the true law $\mathbb{P}^*_{XYM}$. Suppose $|\mathcal{S}| = k < \infty$. Then, for every confidence level $\eta \in (0, 1)$, if*

$$\epsilon_n(\eta) \geq \frac{k \log(n+1) + \log(1/\eta)}{n},$$

*we have*

$$\Pr\left[ D_{\mathrm{KL}}\left(\widehat{\mathbb{P}}_n \| \mathbb{P}^*_{XYM}\right) \leq \epsilon_n(\eta) \right] \geq 1 - \eta.$$

*Proof.* Proof of Lemma C.2. This is the classical result follows from the standard method-of-types concentration bound; see, e.g., (Cover & Thomas, 2006). For completeness we briefly mention it.

For any $\varepsilon > 0$,

$$\Pr\left\{D_{\mathrm{KL}}\left(\widehat{\mathbb{P}}_n \| \mathbb{P}^*_{XYM}\right) > \varepsilon\right\} \leq (n+1)^k e^{-n\varepsilon}.$$

Indeed, if $\mathcal{T}_n$ denotes the set of all empirical distributions (types) generated by $n$ samples on $\mathcal{S}$, then

$$|\mathcal{T}_n| \leq (n+1)^k.$$

Moreover, for any fixed type $q \in \mathcal{T}_n$, the method of types gives

$$\Pr\{\widehat{\mathbb{P}}_n = q\} \leq \exp[-n D_{\mathrm{KL}}(q \| \mathbb{P}^*_{XYM})].$$

Therefore,

$$\begin{aligned}
\Pr\left\{D_{\mathrm{KL}}\left(\widehat{\mathbb{P}}_n \| \mathbb{P}^*_{XYM}\right) > \varepsilon\right\} &= \sum_{q \in \mathcal{T}_n} \Pr\{\widehat{\mathbb{P}}_n = q\} \mathbf{1}\{D_{\mathrm{KL}}(q \| \mathbb{P}^*_{XYM}) > \varepsilon\} \\
&\leq \sum_{q \in \mathcal{T}_n} \exp[-n D_{\mathrm{KL}}(q \| \mathbb{P}^*_{XYM})] \, \mathbf{1}\{D_{\mathrm{KL}}(q \| \mathbb{P}^*_{XYM}) > \varepsilon\} \\
&\leq |\mathcal{T}_n| e^{-n\varepsilon} \\
&\leq (n+1)^k e^{-n\varepsilon}.
\end{aligned}$$

Now set

$$\varepsilon_0 \geq \frac{k \log(n+1) + \log(1/\eta)}{n}.$$

Then

$$(n+1)^k e^{-n\varepsilon_0} \leq \eta.$$

Hence

$$\Pr\left[D_{\mathrm{KL}}\left(\widehat{\mathbb{P}}_n \| \mathbb{P}^*_{XYM}\right) \leq \varepsilon_0\right] \geq 1 - \eta.$$

This proves the claim. $\qquad\square$

*Proof.* Proof of Theorem 3.7.

We next establish the event containment

$$\{\omega : \ D_{\mathrm{KL}}(\widehat{\mathbb{P}}_{XYM}, \mathbb{P}^*_{XYM}) \leq \epsilon_n(\eta)\} \subseteq \{\omega : \ \mathbb{P}^*_{XY} \in \mathcal{D}(\gamma_n(\eta))\},$$

which is the key step linking statistical concentration at the latent level to the feasibility of the true distribution in the observable ambiguity set.

Let $\widehat{\mathbb{P}}_{XYM}$ denote the empirical law of the **latent** complete triples $(X_i, Y_i, M_i)$, used only for the analysis, and recall $\widehat{\mathbb{P}}^{\mathrm{obs}}_n = \widehat{\mathbb{P}}_{\tilde{X}YM}$ denote its **observable pushforward**. We first show the event containment

$$\left\{D_{\mathrm{KL}}\left(\widehat{\mathbb{P}}_{XYM} \,\middle\|\, \mathbb{P}^*_{XYM}\right) \leq \epsilon_n(\eta)\right\} \subseteq \{\mathbb{P}^*_{XY} \in \mathcal{D}(\gamma(\epsilon_n(\eta)))\}$$

where $\gamma(\epsilon_n(\eta))$ is induced by Proposition 3.2 for the KL radius $\epsilon_n(\eta)$.

The mask variable $M$ encodes which components of $X$ are observed and which are missing. Distinct latent triples $(x, y, m) \in \mathcal{X} \times \mathcal{Y} \times \mathcal{M}$ may therefore be mapped to the same observable triple $(\tilde{x}, y, m) \in \tilde{\mathcal{X}} \times \mathcal{Y} \times \mathcal{M}$ once the missing entries in $x$ are replaced by NA symbols. In other words, the observable alphabet is obtained from the latent alphabet by a many–to–one measurable mapping.

By the data–processing inequality for Kullback–Leibler divergence, KL divergence contracts under such measurable coarsenings. Consequently,

$$D_{\mathrm{KL}}\left(\widehat{\mathbb{P}}_{XYM} \,\middle\|\, \mathbb{P}^*_{XYM}\right) \geq D_{\mathrm{KL}}\left(\widehat{\mathbb{P}}^{\mathrm{obs}}_n \,\middle\|\, \mathbb{P}^*_{\tilde{X}YM}\right)$$

Hence, for every radius $\epsilon > 0$,

$$\left\{ D_{\mathrm{KL}}\left(\widehat{\mathbb{P}}_{XYM}, \mathbb{P}^*_{XYM}\right) \le \epsilon \right\} \subseteq \left\{ D_{\mathrm{KL}}\left(\widehat{\mathbb{P}}^{\mathrm{obs}}_n, \mathbb{P}^*_{\tilde{X}YM}\right) \le \epsilon \right\}.$$

By Assumption 2.5, the true missingness mechanism admits a version $q^\star \in \mathcal{Q}_{\mathrm{MAR}}$, and therefore $\mathbb{P}^*_{\tilde{X}YM} = \mathrm{H}(\mathbb{P}^*_{XY}, q^\star)$. Consequently, whenever $D_{\mathrm{KL}}(\widehat{\mathbb{P}}_{XYM} \| \mathbb{P}^*_{XYM}) \le \epsilon$, we also have

$$D_{\mathrm{KL}}\left(\widehat{\mathbb{P}}^{\mathrm{obs}}_n \,\middle\|\, \mathrm{H}(\mathbb{P}^*_{XY}, q^\star)\right) \le \epsilon$$

and equivalently $\mathbb{P}^*_{XY} \in \mathcal{P}_{\mathrm{KL}}(\epsilon)$.

By Lemma C.2,

$$\Pr\left[ D_{\mathrm{KL}}\left(\widehat{\mathbb{P}}_{XYM} \,\middle\|\, \mathbb{P}^*_{XYM}\right) \le \epsilon_n(\eta) \right] \ge 1 - \eta$$

Thus, with probability at least $1 - \eta$, $\mathbb{P}^*_{XY} \in \mathcal{P}_{\mathrm{KL}}(\epsilon_n(\eta))$. By Proposition 3.2, this implies $\mathbb{P}^*_{XY} \in \mathcal{D}(\gamma(\epsilon_n(\eta)))$. The calibrated threshold is clipped as $\gamma_n(\eta) = \min\{\gamma_{\max}, \gamma(\epsilon_n(\eta))\}$, then $\gamma_n(\eta) \le \gamma(\epsilon_n(\eta))$.

Since $\mathcal{D}(\gamma)$ is decreasing in $\gamma$, we have

$$\mathcal{D}\big(\gamma(\epsilon_n(\eta))\big) \subseteq \mathcal{D}\big(\gamma_n(\eta)\big)$$

Therefore,

$$\Pr\big[\mathbb{P}^*_{XY} \in \mathcal{D}\big(\gamma_n(\eta)\big)\big] \ge 1 - \eta$$

On this event, for every $z \in \mathcal{Z}$,

$$\sup_{p \in \mathcal{D}\big(\gamma_n(\eta)\big)} \mathbb{E}_p[c(z, Y) \mid X = \bar{x}] \ge \mathbb{E}_{\mathbb{P}^*}[c(z, Y) \mid X = \bar{x}]$$

Taking $z = \hat{z}_n$, and using the definition of $\Phi^*(\bar{x})$ as the oracle minimum over $z \in \mathcal{Z}$, we obtain

$$\widehat{\Phi}_n(\bar{x}) \ge \mathbb{E}_{\mathbb{P}^*}[c(\hat{z}_n, Y) \mid X = \bar{x}] \ge \Phi^*(\bar{x})$$

with probability at least $1 - \eta$. This proves the result.

$\square$

## C.5. Proof of Theorem 3.8

**Lemma C.3** (Borel–Cantelli Lemma; see, e.g., Durrett (2019)). *Let $(\Omega, \mathcal{F}, \mathbb{P})$ be a probability space, and let $\{A_n\}_{n \ge 1} \subset \mathcal{F}$ be a sequence of events. Then:*

1. *If $\sum_{n=1}^{\infty} \mathbb{P}(A_n) < \infty$, then*

$$\mathbb{P}\left( \limsup_{n \to \infty} A_n \right) = 0$$

   *Equivalently, with probability one, only finitely many of the events $A_n$ occur.*

2. *If the events $\{A_n\}_{n \ge 1}$ are independent and $\sum_{n=1}^{\infty} \mathbb{P}(A_n) = \infty$, then*

$$\mathbb{P}\left( \limsup_{n \to \infty} A_n \right) = 1$$

   *Equivalently, with probability one, infinitely many of the events $A_n$ occur.*

*Proof.* Proof of Theorem 3.8.

**Preparation.** For clarity, we first recall the definition of $\gamma_n := \gamma_n(\eta_n)$.

Let $L_1^\star := \sup\left\{\frac{1}{n}\sum_{i=1}^n \log\left((Tp)_i\right) : p \in \Delta(\mathcal{X} \times \mathcal{Y})\right\}$ and set $\gamma_{\max} := L_1^\star$. By Proposition 3.2, the likelihood threshold induced by a KL radius $\epsilon$ is

$$\gamma(\epsilon) = \sum_{\xi \in \widehat{\Sigma}} \frac{n_\xi}{n} \log\left(\frac{n_\xi}{n}\right) - \frac{1}{n}\sum_{i=1}^n \log \widehat{q}_n(m_i \mid x_{o,i}, y_i) - \epsilon$$

and $\widehat{q}_n$ is the empirical conditional mask law. Hence the minimal KL radius for which the likelihood set is nonempty is

$$\epsilon_{\min,n} = \sum_{\xi \in \widehat{\Sigma}} \frac{n_\xi}{n} \log\left(\frac{n_\xi}{n}\right) - \frac{1}{n}\sum_{i=1}^n \log \widehat{q}_n(m_i \mid x_{o,i}, y_i) - \gamma_{\max}$$

Equivalently, $\gamma(\epsilon_{\min,n}) = \gamma_{\max}$. Therefore the calibrated threshold $\gamma_n(\eta) := \min\{\gamma(\epsilon_n(\eta)), \gamma_{\max}\}$ can be written as $\gamma_n(\eta) = \gamma\left(\max\{\epsilon_n(\eta), \epsilon_{\min,n}\}\right)$ and $\mathcal{D}(\gamma_n(\eta))$ is nonempty.

Moreover, $\epsilon_{\min,n} \to 0$ almost surely. Indeed, under the MAR assumption, the true observable law satisfies $\mathbb{P}_{\text{obs}}^* = \mathrm{H}(\mathbb{P}_{XY}^*, q^\star)$ for some $q^\star \in \mathcal{Q}_{\text{MAR}}$. Since $\epsilon_{\min,n}$ is the infimum over the model class,

$$0 \leq \epsilon_{\min,n} \leq D_{\text{KL}}\left(\widehat{\mathbb{P}}_n^{\text{obs}} \,\Big\|\, \mathbb{P}_{\text{obs}}^*\right)$$

The observable support is finite, so $\widehat{\mathbb{P}}_n^{\text{obs}}(\xi) \to \mathbb{P}_{\text{obs}}^*(\xi)$ for every observable atom $\xi$, almost surely. Moreover, if $\mathbb{P}_{\text{obs}}^*(\xi) = 0$, then $\widehat{\mathbb{P}}_n^{\text{obs}}(\xi) = 0$ for all $n$ almost surely. Thus

$$D_{\text{KL}}\left(\widehat{\mathbb{P}}_n^{\text{obs}} \,\Big\|\, \mathbb{P}_{\text{obs}}^*\right) \to 0$$

almost surely, and therefore $\epsilon_{\min,n} \to 0$ almost surely.

For simplicity of notaions, let the calibrated radius be $\gamma_n := \gamma_n(\eta_n)$, $\tilde{\epsilon}_n = \min\{\varepsilon_{\min,n}, \epsilon_n(\eta)\}$ and denote by $\mathbb{P}_{\text{obs}}^*$ the true observable law and write $\tilde{p}^* := \mathbb{P}_{\text{obs}}^*$.

**A.** Since $\hat{z}_n \in \mathcal{Z}$, we have $\Phi^* \leq \mathbb{E}_{\mathbb{P}^*}\left[c(\hat{z}_n, Y) \mid X = \bar{x}\right]$. Moreover, Theorem 3.7 with $z = \hat{z}_n$ yields

$$\Pr\left[\mathbb{E}_{\mathbb{P}^*}\left[c(\hat{z}_n, Y) \mid X = \bar{x}\right] \leq \Phi_n\right] \geq 1 - \eta_n$$

Hence

$$\Pr\left[\Phi^* \leq \mathbb{E}_{\mathbb{P}^*}\left[c(\hat{z}_n, Y) \mid X = \bar{x}\right] \leq \Phi_n\right] \geq 1 - \eta_n \tag{A}$$

Because $\sum_{n=1}^\infty \eta_n < \infty$, the Borel–Cantelli lemma C.3 implies that the event in (A) occurs for all sufficiently large $n$ almost surely. We therefore confine ourselves to the complementary inequality $\limsup_{n \to \infty} \Phi_n \leq \Phi^*$ and the convergence of $\hat{z}_n$.

**B.** We are interested in properties that hold for every $\mathbb{Q}_{XY} \in \mathcal{D}_n$. Given such a $\mathbb{Q}_{XY}$, replace its original mask law $\mathbb{Q}_{M|X_o,Y}$ by the empirical version $\bar{\mathbb{Q}}_{M|X_o,Y}(m \mid x_o, y) := \widehat{\mathbb{P}}_{\text{obs}}\left(M = m \mid x_o, y\right)$ does not increase the observable KL divergence, hence $\mathrm{H}(\mathbb{Q}_{XY}, \bar{\mathbb{Q}}_{M|X_o,Y})$ remains in the same KL–ball with respect to the empirical observable distribution. Consequently, analysing the modified pair therefore suffices to characterise all members of $\mathbb{Q}_{XY} \in \mathcal{D}_n$.

Pinsker's inequality together with the KL constraint yields $\|\tilde{q} - \hat{p}\|_1 \leq \sqrt{2\tilde{\epsilon}_n}$, where $\tilde{q} := \mathrm{H}(\mathbb{Q}_{XY}, \bar{\mathbb{Q}}_{M|X_o,Y})$. We also know from lemma C.2 together with pinsker's inequality $\Pr\left[\|\tilde{p}^\star - \hat{p}\|_1 \leq \sqrt{2\tilde{\epsilon}_n}\right] \geq 1 - \eta_n$

By triangular inequality and the fact that $\tilde{\epsilon}_n \geq \epsilon_n$, we have

$$\Pr\left[\|\tilde{p}^\star - \tilde{q}\|_1 \leq 2\sqrt{2\tilde{\epsilon}_n}\right] \geq \Pr\left[\|\tilde{p}^\star - \tilde{q}\|_1 \leq (\sqrt{2\epsilon_n} + \sqrt{2\tilde{\epsilon}_n})\right] \geq 1 - \eta_n$$

As $\sum_{i=1}^\infty \eta_n < \infty$, the Borel–Cantelli Lemma C.3 further implies that with probability 1, for all sufficiently large $n \geq N_0(\omega)$,

$$\|\tilde{p}^\star - \tilde{q}\|_1 \leq 2\sqrt{2\tilde{\epsilon}_n} \tag{B1}$$

In the $m = 0$ slice we have

$$\tilde{q}(x_m, x_o, y, 0) = \mathbb{Q}_{XY}(x_o, x_m, y)\,\bar{\mathbb{Q}}_{M|X_o,Y}(0 \mid x_o, y)$$

Assumption 2.4 ensures

$$\lambda_{\min} := \min_{\substack{(x_o, y) \in \mathcal{X}_o \times \mathcal{Y} \\ \mathbb{P}^*(x_o, y) > 0}} \mathbb{P}^*\big(M = 0 \mid x_o, y\big) \; > \; 0$$

For every pair $(x_o, y)$ with $\mathbb{P}^*(x_o, y) > 0$ define

$$\widehat{\pi}_n(x_o, y) := \frac{1}{N_n(x_o, y)} \sum_{i=1}^{n} \mathbf{1}\big\{X_{o,i} = x_o,\, Y_i = y,\, M_i = 0\big\},$$

$$N_n(x_o, y) := \sum_{i=1}^{n} \mathbf{1}\{X_{o,i} = x_o,\, Y_i = y\}$$

whenever $N_n(x_o, y) \geq 1$. Because the support of $(X_o, Y)$ is finite, there are only finitely many pairs $(x_o, y)$ with $\mathbb{P}^*(x_o, y) > 0$. For each such pair, since $N_n(x_o, y)/n \to \mathbb{P}^*(x_o, y) > 0$ almost surely, there exists an almost surely finite random index $N_{x_o, y}(\omega)$ such that $N_n(x_o, y) \geq 1$ for all $n \geq N_{x_o, y}(\omega)$. Define

$$N_1(\omega) := \max_{(x_o, y):\, \mathbb{P}^*(x_o, y) > 0} N_{x_o, y}(\omega),$$

which is almost surely finite because the index set is finite. Then $\hat{\pi}_n(x_o, y)$ is well-defined for all $(x_o, y)$ with $\mathbb{P}^*(x_o, y) > 0$ and all $n \geq N_1(\omega)$. Moreover, by the strong law of large numbers, for each such pair

$$\hat{\pi}_n(x_o, y) \xrightarrow{\text{a.s.}} \mathbb{P}^*(M = 0 \mid x_o, y)$$

Since the index set is finite, this convergence holds simultaneously for all $(x_o, y)$ with $\mathbb{P}^*(x_o, y) > 0$. Consequently, on an event of probability one, there exists $N_2(\omega) \geq N_1(\omega)$ such that for all $n \geq N_2(\omega)$,

$$\widehat{\pi}_n(x_o, y) \; \geq \; \tfrac{1}{2}\lambda_{\min}, \quad \forall (x_o, y)\colon \mathbb{P}^*(x_o, y) > 0. \tag{B2}$$

Recall on the slice $m = 0$ we also have

$$\tilde{p}^*(x_m, x_o, y, 0) = \mathbb{P}^*_{XY}(x_o, x_m, y)\,\mathbb{P}^*\big(M = 0 \mid x_o, y\big).$$

Summing the absolute difference and using (B1) together with (B2) gives for $n \geq \max(N_0(\omega), N_2(\omega))$

$$\big\|\mathbb{Q}_{XY} - \mathbb{P}^*_{XY}\big\|_1 = \sum_{x_o, y} \sum_{x_m} \left| \frac{\tilde{q}(x_m, x_o, y, 0)}{\widehat{\pi}_n(x_o, y)} - \frac{\tilde{p}^*(x_m, x_o, y, 0)}{\mathbb{P}^*(M = 0 \mid x_o, y)} \right|$$

$$\leq \sum_{x_o, y} \left( \sum_{x_m} \frac{\big|\tilde{q}(x_m, x_o, y, 0) - \tilde{p}^*(x_m, x_o, y, 0)\big|}{\widehat{\pi}_n(x_o, y)} + \left| \frac{1}{\widehat{\pi}_n(x_o, y)} - \frac{1}{\mathbb{P}^*(M = 0 \mid x_o, y)} \right| \sum_{x_m} \tilde{p}^*(x_m, x_o, y, 0) \right)$$

$$\leq \frac{2}{\lambda_{\min}} \sum_{x_o, x_m, y} \big|\tilde{q}(x_m, x_o, y, 0) - \tilde{p}^*(x_m, x_o, y, 0)\big| + \frac{2\beta_n}{\lambda_{\min}^2}$$

$$\leq \frac{2}{\lambda_{\min}} \|\tilde{q} - \tilde{p}^*\|_1 + \frac{2\beta_n}{\lambda_{\min}^2} \leq \frac{4\sqrt{2\tilde{\epsilon}_n}}{\lambda_{\min}} + \frac{2\beta_n}{\lambda_{\min}^2}$$

where $\beta_n := \sum_{x_o, y} |\hat{\pi}_n(x_o, y) - \mathbb{P}^*(M = 0 | x_o, y)| \to 0$. Hence, for all sufficiently large $n \geq \max(N_0(\omega), N_2(\omega))$,

$$\big\|\mathbb{Q}_{XY} - \mathbb{P}^*_{XY}\big\|_1 \; \leq \; \frac{4\sqrt{2\tilde{\epsilon}_n}}{\lambda_{\min}} + \frac{2\beta_n}{\lambda_{\min}^2} \tag{B}$$

holds for **every** $Q_{XY} \in \mathcal{D}_n$. Inequality (B) is the key input for the subsequent bound on the conditional cost and for establishing the almost–sure relation $\limsup_{n \to \infty} \Phi_n \leq \Phi^*$.

**C.** Fix $z \in \mathcal{Z}$ and recall the conditional–cost functional

$$F_z(p) := \mathbb{E}_p[c(z, Y) \mid X = \bar{x}] = \frac{\sum_{y \in \mathcal{Y}} p(\bar{x}, y)\, c(z, y)}{\sum_{y \in \mathcal{Y}} p(\bar{x}, y)},$$

Write $\mathbb{Q}_X(\bar{x}) := \sum_y \mathbb{Q}_{XY}(\bar{x}, y)$ and $\mathbb{P}_X^*(\bar{x}) := \varepsilon_0 > 0$ (Assumption 2.2). By total–variation domination (B),

$$\left|\mathbb{Q}_X(\bar{x}) - \mathbb{P}_X^*(\bar{x})\right| \leq \|\mathbb{Q}_{XY} - \mathbb{P}_{XY}^*\|_1 \leq \frac{4\sqrt{2\tilde{\epsilon}_n}}{\lambda_{\min}} + \frac{2\beta_n}{\lambda_{\min}^2}$$

Choose $n \geq N_3$ large enough that $\frac{4\sqrt{2\tilde{\epsilon}_n}}{\lambda_{\min}} + \frac{2\beta_n}{\lambda_{\min}^2} \leq \frac{\varepsilon_0}{2}$; then

$$\mathbb{Q}_X(\bar{x}) \geq \varepsilon_0 - \frac{4\sqrt{2\tilde{\epsilon}_n}}{\lambda_{\min}} - \frac{2\beta_n}{\lambda_{\min}^2} \geq \frac{\varepsilon_0}{2}. \tag{C1}$$

Because $|c(z, y)| \leq C_{\max}$ by assumption 2.1,

$$\left|\sum_y \mathbb{Q}_{XY}(\bar{x}, y) c(z, y) - \sum_y \mathbb{P}_{XY}^*(\bar{x}, y) c(z, y)\right| \leq C_{\max} \|\mathbb{Q}_{XY} - \mathbb{P}_{XY}^*\|_1 \leq \frac{4C_{\max}\sqrt{2\tilde{\epsilon}_n}}{\lambda_{\min}} + \frac{2C_{\max}\beta_n}{\lambda_{\min}^2} \tag{C2}$$

With the denominator lower bound (C1) and the numerator difference bound (C2) we obtain, for any $z \in \mathcal{Z}$,

$$\begin{aligned}
&\left|F_z(\mathbb{Q}_{XY}) - F_z(\mathbb{P}_{XY}^*)\right| \\
&= \left|\frac{\sum_y \mathbb{Q}_{XY}(\bar{x}, y) c(z, y)}{\mathbb{Q}_{XY}(\bar{x})} - \frac{\sum_y \mathbb{P}_{XY}^*(\bar{x}, y) c(z, y)}{\mathbb{P}_{XY}^*(\bar{x})}\right| \\
&= \left|\frac{\sum_y [\mathbb{Q}_{XY} - \mathbb{P}_{XY}^*](\bar{x}, y) c(z, y)}{\mathbb{Q}_{XY}(\bar{x})} + \sum_y \mathbb{P}_{XY}^*(\bar{x}, y) c(z, y)\left(\frac{1}{\mathbb{Q}_{XY}(\bar{x})} - \frac{1}{\mathbb{P}_{XY}^*(\bar{x})}\right)\right| \\
&\leq \frac{C_{\max} \|\mathbb{Q}_{XY} - \mathbb{P}_{XY}^*\|_1}{\mathbb{Q}_{XY}(\bar{x})} + \frac{C_{\max}\left|\mathbb{Q}_X(\bar{x}) - \mathbb{P}_X^*(\bar{x})\right|}{\mathbb{Q}_{XY}(\bar{x})\, \mathbb{P}_{XY}^*(\bar{x})} \\
&\overset{\text{(C1),(C2)}}{\leq} \frac{4C_{\max}\sqrt{2\tilde{\epsilon}_n}/\lambda_{\min} + \frac{2C_{\max}\beta_n}{\lambda_{\min}^2}}{\varepsilon_0/2} + \frac{4C_{\max}\sqrt{2\tilde{\epsilon}_n}}{\lambda_{\min}\varepsilon_0^2/2} + \frac{2C_{\max}\beta_n}{\lambda_{\min}^2\varepsilon_0^2/2} \\
&= \left(\frac{8C_{\max}}{\varepsilon_0\lambda_{\min}} + \frac{8C_{\max}}{\lambda_{\min}\varepsilon_0^2}\right)\sqrt{2\tilde{\epsilon}_n} + \left(\frac{4C_{\max}}{\varepsilon_0\lambda_{\min}^2} + \frac{4C_{\max}}{\lambda_{\min}^2\varepsilon_0^2}\right)\beta_n
\end{aligned}$$

Define constant $L_1 := \left(\frac{8C_{\max}}{\varepsilon_0\lambda_{\min}} + \frac{8C_{\max}}{\lambda_{\min}\varepsilon_0^2}\right)$ and $L_2 := \left(\frac{4C_{\max}}{\varepsilon_0\lambda_{\min}^2} + \frac{4C_{\max}}{\lambda_{\min}^2\varepsilon_0^2}\right)$; we obtain

$$\left|F_z(\mathbb{Q}_{XY}) - F_z(\mathbb{P}_{XY}^*)\right| \leq L_1\sqrt{2\tilde{\epsilon}_n} + L_2\beta_n \tag{C}$$

holds for **every** $\mathbb{Q}_{XY} \in \mathcal{D}_n$ and sufficiently large $n$.

**D.** Fix an arbitrary tolerance $\zeta > 0$. Because $\Phi^* = \min_{z \in \mathcal{Z}} F_z(\mathbb{P}_{XY}^*)$, there exists a decision $z_\zeta \in \mathcal{Z}$ satisfying

$$F_{z_\zeta}(\mathbb{P}_{XY}^*) \leq \Phi^* + \zeta$$

For each sample size $n$ select a distribution $p_n^{(\zeta)} \in \arg\max_{p \in \mathcal{D}_n} F_{z_\zeta}(p)$, that is, $p_n^{(\zeta)}$ attains the inner supremum of the robust objective at the fixed action $z_\zeta$, which is ensured by Propostion 3.3 and corresponding assumptions.

By definition of $\Phi_n$,

$$\Phi_n = \inf_{z \in \mathcal{Z}} \sup_{p \in \mathcal{D}_n} F_z(p) \leq \sup_{p \in \mathcal{D}_n} F_{z_\zeta}(p) = F_{z_\zeta}(p_n^{(\zeta)}) \tag{D1}$$

The pair $(z_\zeta, p_n^{(\zeta)})$ lies within the range covered by inequality (C); hence

$$\left| F_{z_\zeta}\big(p_n^{(\zeta)}\big) - F_{z_\zeta}\big(\mathbb{P}_{XY}^*\big) \right| \;\leq\; L_1 \sqrt{2\tilde{\epsilon}_n} + L_2 \beta_n,$$

Combining this with (D1) yields

$$\Phi_n \;\leq\; F_{z_\zeta}\big(\mathbb{P}_{XY}^*\big) + L_1\sqrt{2\tilde{\epsilon}_n} + L_2\beta_n \;\leq\; \Phi^* + \zeta + L_1\sqrt{2\tilde{\epsilon}_n} + L_2\beta_n \tag{D2}$$

Because $\tilde{\epsilon}_n \to 0$ and $\beta_n \to 0$ as $n \to \infty$, (D2) implies

$$\limsup_{n\to\infty} \Phi_n \;\leq\; \Phi^* + \zeta$$

Since $\zeta > 0$ was arbitrary, we obtain the desired inequality

$$\limsup_{n\to\infty} \Phi_n \leq \Phi^* \tag{D}$$

This completes the upper–bound portion of the proof; together with the almost–sure lower bound established via Borel–Cantelli (A), we conclude that $\Phi_n \downarrow \Phi^*$ almost surely.

**E.** Assume a cluster point exist , i.e., there are indices $n_k \uparrow \infty$ and a vector $\bar{z} \in \mathbb{R}^d$ such that $\hat{z}_{n_k} \xrightarrow{k\to\infty} \bar{z}.$, and each $\hat{z}_{n_k}$ belongs to the decision set $\mathcal{Z}$. Because $\mathcal{Z}$ is *closed*, its limit $\bar{z}$ also lies in $\mathcal{Z}$; hence $\bar{z} \in \mathcal{Z}$.

For all sufficiently large $k$ we have $\mathbb{P}_{XY}^* \in \mathcal{D}(\gamma_{n_k})$; therefore

$$F_{\hat{z}_{n_k}}\big(\mathbb{P}_{XY}^*\big) \;\leq\; \sup_{p\in\mathcal{D}(\gamma_{n_k})} F_{\hat{z}_{n_k}}(p) \;=\; \Phi_{n_k} \tag{E1}$$

By lower semicontinuity of $z \mapsto F_z(\mathbb{P}_{XY}^*)$,

$$F_{\bar{z}}\big(\mathbb{P}_{XY}^*\big) \;\leq\; \liminf_{k\to\infty} F_{\hat{z}_{n_k}}\big(\mathbb{P}_{XY}^*\big) \tag{E2}$$

Combining E1 and E2 the almost-sure value convergence $\Phi_{n_k} \longrightarrow \Phi^*$ gives

$$F_{\bar{z}}\big(\mathbb{P}_{XY}^*\big) \;\leq\; \liminf_{k\to\infty} \Phi_{n_k} = \Phi^* \;\leq\; F_{\bar{z}}\big(\mathbb{P}_{XY}^*\big),$$

so equality holds: $F_{\bar{z}}\big(\mathbb{P}_{XY}^*\big) = \Phi^*$ Hence $\bar{z} \in \mathcal{Z}^* := \arg\min_{z\in\mathcal{Z}} F_z\big(\mathbb{P}_{XY}^*\big)$. Because the argument applies to *any* cluster point, we conclude

$$\mathrm{Acc}\{\hat{z}_n\} \subseteq \mathcal{Z}^* \quad \text{almost surely.}$$

$\square$

### C.6. Proof of Proposition 3.11

It suffices to prove that for

$$\mathcal{K}'(\gamma, \delta; J) := \left\{ \mathbb{P}(y \mid \bar{\mathbf{x}}) \,\middle|\, \mathbb{P}(y \mid \bar{\mathbf{x}}) \in B_{\mathrm{LP}, \frac{2\delta}{\mathbb{Q}(\bar{\mathbf{x}})}}\left(\mathbb{Q}(y \mid \bar{\mathbf{x}})\right), \; \mathbb{Q} \in \mathcal{B}_J \,,\, \frac{1}{n}\sum_{i=1}^n \log\left( \sum_{x_{m,i}\in\mathcal{A}(\tilde{x}_{m,i}, m_i)} \mathbb{Q}(y_i \mid x_{o,i}, x_{m,i})\mathbb{Q}(x_{o,i}, x_{m,i}) \right) \geq \gamma \right\}$$

We have

$$\{\mathbb{P}(y \mid \bar{\mathbf{x}}) \mid \mathbb{P}_{XY} \in \mathcal{P}_{\mathrm{BinPKL}}(\epsilon, \delta; J)\} \subseteq \mathcal{K}'(\gamma, \delta; J) \subseteq \{\mathbb{P}(y \mid \bar{\mathbf{x}}) \mid \mathbb{P}_{XY} \in \mathcal{P}_{\mathrm{BinPKL}}(\epsilon, 3\delta; J)\}.$$

As a direct consequence, we obtain the following sandwiching relations:

$$\mathcal{K}(\gamma, 2\delta; J) \;\subseteq\; \mathcal{K}'(\gamma, \delta; J) \;\subseteq\; \{\mathbb{P}(y \mid \bar{\mathbf{x}}) \mid \mathbb{P}_{XY} \in \mathcal{P}_{\mathrm{BinPKL}}(\epsilon, 3\delta; J)\},$$

$$\mathcal{K}(\gamma, 2\delta; J) \;\supseteq\; \mathcal{K}'(\gamma, \delta \cdot \epsilon_1; J) \;\supseteq\; \{\mathbb{P}(y \mid \bar{\mathbf{x}}) \mid \mathbb{P}_{XY} \in \mathcal{P}_{\mathrm{BinPKL}}(\epsilon, \delta\epsilon_1; J)\}.$$

These inclusions recover exactly the containment relations stated in Proposition 3.11.

*Proof.* Proof of the claim. Recall that the ambiguity set $\mathcal{P}_{\text{BinPKL}}(\epsilon, \delta; J)$ can be written as:

$$\mathcal{P}_{\text{BinPKL}}(\epsilon, \delta; J) := \left\{ \mathbb{P}_{XY} \in \mathcal{P}(\mathcal{X} \times \mathcal{Y}) \; \middle| \; \begin{array}{l} \exists \, \mathbb{Q}_{XY} \in \mathcal{B}_J, \mathbb{Q}_{M|X_o,Y} \in \mathcal{Q}_{\text{MAR}} \text{ such that:} \\ \text{(i) } \mathbb{P}_{XY} \in B_{\text{LP},\delta}(\mathbb{Q}_{XY}), \\ \text{(ii) } D_{\text{KL}}\left( \widehat{\mathbb{P}}_n^{\text{obs}} \, \middle\| \, \mathbb{H}(\mathbb{Q}_{XY}, \mathbb{Q}_{M|X_o,Y}) \right) \leq \epsilon. \end{array} \right\}.$$

To analyze the KL divergence term in the definition, we first note that since the KL radii is finite, the empirical distribution $\widehat{\mathbb{P}}$ is absolutely continuous with respect to candidate model $\mathbb{H}(\mathbb{Q}_{XY}, \mathbb{Q}_{M|X_o,Y})$ and note that $\mathbb{Q}_{M|X_o,Y} \in \mathcal{Q}_{\text{MAR}}$. This allows us to decompose the KL divergence as

$$D_{\text{KL}}\left( \widehat{\mathbb{P}}_n^{\text{obs}} \, \| \, \mathbb{H}(\mathbb{Q}_{XY}, \mathbb{Q}_{M|X_o,Y}) \right) = \int \log \left( \frac{d\widehat{\mathbb{P}}}{d\mathbb{H}}(\tilde{x}, y, m) \right) \, d\widehat{\mathbb{P}}(\tilde{x}, y, m)$$

$$= \sum_{\xi \in \widehat{\Sigma}} \frac{n_\xi}{n} \log \left( \frac{n_\xi}{n} \right) - \frac{1}{n} \sum_i \log \mathbb{Q}(m_i \mid x_{o,i}, y_i) \qquad \text{(Decomposition 2)}$$

$$- \frac{1}{n} \sum_{i=1}^n \log \left( \sum_{x_{m,i} \in \mathcal{A}(\tilde{x}_{m,i}, m_i)} \mathbb{Q}_{XY}(x_{o,i}, x_{m,i}, y_i) \right)$$

At this point, we observe that a decomposed structure similar to that in the proof of C.1 also arises in the present setting. Consequently, by setting $\gamma$ as in Proposition 3.11, we obtain an equivalent characterization of the ambiguity set. The remaining arguments closely follow those in the proof of C.1. This leads to the following reformulation of the ambiguity set:

$$\left\{ \mathbb{P}_{XY} \in \mathcal{P}(\mathcal{X} \times \mathcal{Y}) \; \middle| \; \begin{array}{l} \exists \, \mathbb{Q}_{XY} \in \mathcal{B}_J, \mathbb{Q}_{M|X_o,Y} \in \mathcal{Q}_{\text{MAR}} \text{ such that:} \\ \text{(i) } \mathbb{P}_{XY} \in B_{\text{LP},\delta}(\mathbb{Q}_{XY}), \\ \text{(ii) } \frac{1}{n} \sum_{i=1}^n \log \left( \sum_{x_{m,i} \in \mathcal{A}(\tilde{x}_{m,i}, m_i)} \mathbb{Q}(y_i \mid x_{o,i}, x_{m,i}) \mathbb{Q}(x_{o,i}, x_{m,i}) \right) \geq \gamma \end{array} \right\}.$$

It remains to consider the LP ball layer (i). In the following we demonstrate that any distribution $\mathbb{P}_{XY}$ lying within an LP neighborhood of some reference model $\mathbb{Q}_{XY}$ can be equivalently represented through a marginal distribution $\mathbb{P}(Y \mid X = \bar{\mathbf{x}})$ parameterized by the same Lévy–Prokhorov tolerance $\delta$ scaled by the reference marginal mass at $\bar{\mathbf{x}}$.

Since $X$ takes values in a finite discrete set while $Y$ is continuous, we express the joint distributions $\mathbb{P}$ and $\mathbb{Q}$ using mixture decompositions:

$$\mathbb{P} = \sum_{i \in |\mathcal{X}|} p_i \, \mathbb{P}^i, \qquad \mathbb{Q} = \sum_{i \in |\mathcal{X}|} q_i \, \mathbb{Q}^i,$$

where $\mathcal{X}$ denotes the finite support of $X$, $p_i$ and $q_i$ represent the marginal probabilities of $X = x_i$, and $\mathbb{P}^i, \mathbb{Q}^i$ denote the conditional laws of $Y$ given $X = x_i$. We consider the nontrivial regime $\delta < 1$; for $\delta \geq 1$, the LP ball is already trivial since for any two probability measures $\mu$ and $\nu$ on a metric space $(M, d)$, the Lévy–Prokhorov distance satisfies $d_{\text{LP}}(\mu, \nu) \leq 1$.

The metric $d$ on $\mathcal{X} \times \mathcal{Y}$ is defined as

$$d\big((x, y), (x', y')\big) := \mathbf{1}\{x \neq x'\} + \|y - y'\|,$$

which induces the LP distance on $\mathcal{X} \times \mathcal{Y}$.

For any Borel set $A \subseteq \mathcal{X} \times \mathcal{Y}$, define $A_i := \{y \in \mathcal{Y} \mid (x_i, y) \in A\}$. Beacuse $\delta < 1$, the $\delta-$neighborhood of A does not mix different X-coordinates. Thus $A^\delta = \bigcup_{i \in |\mathcal{X}|}(\{x_i\} \times A_i^\delta)$

$$\mathbb{P}(A) = \sum_i p_i \, \mathbb{P}^i(A_i), \qquad \mathbb{Q}(A^\delta) = \sum_i q_i \, \mathbb{Q}^i(A_i^\delta),$$

where $A^\delta$ denotes the $\delta$-neighborhood of $A$ under $d$.

**Necessary Conditions (Left Inclusion)**    We now derive necessary conditions for $\mathbb{P} \in B_{\mathrm{LP},\delta}(\mathbb{Q})$ . Assume $\mathbb{P} \in B_{\mathrm{LP},\delta}(\mathbb{Q})$. By definition, for all Borel set $A \subseteq S \times \mathcal{Y}$, we have $\mathbb{P}(A) \leq \mathbb{Q}(A^\delta) + \delta$ , we compute the difference as

$$\mathbb{P}(A) - \mathbb{Q}(A^\delta) = \sum_i (p_i - q_i)\mathbb{P}^i(A_i) + \sum_i q_i \left[\mathbb{P}^i(A_i) - \mathbb{Q}^i(A_i^\delta)\right].$$

Similarly, $\mathbb{Q}(A) \leq \mathbb{P}(A^\delta) + \delta$ . We can derive two implications from this expression:

- Marginal condition: By choosing $A$ to lie entirely within a single $x$-coordinate, i.e. $A = \{x_i\} \times \mathcal{Y}$, then $A^\delta = A$,and we obtain $p_i \leq q_i + \delta$, $q_i \leq p_i + \delta$ for all $i$, i.e.

$$|p_i - q_i| \leq \delta.$$

- Conditional condition: Fix $i \in |\mathcal{X}|$ , and an arbituary Borel set $\tilde{A} \subseteq \mathcal{Y}$. Define $A := \{i\} \times \tilde{A}$, the LP constraint $\mathbb{Q}(A) \leq \mathbb{P}(A^\delta) + \delta$ yields

$$q_i \, \mathbb{Q}^i(\tilde{A}) \leq p_i \, \mathbb{P}^i(\tilde{A}^\delta) + \delta.$$

Using the bound $p_i \leq q_i + \delta$ (from the marginal constraint) and the fact that $\mathbb{P}^i(A^\delta) \leq 1$, we obtain

$$q_i\mathbb{Q}^i(\tilde{A}) \leq p_i \, \mathbb{P}^i(\tilde{A}^\delta) + \delta \leq (q_i + \delta) \, \mathbb{P}^i(\tilde{A}^\delta) + \delta \leq q_i\mathbb{P}^i(\tilde{A}^\delta) + 2\delta$$

For every coordinate $i$ with $q_i > 0$, dividing by $q_i$ yields

$$\mathbb{Q}^i(\tilde{A}) \leq \mathbb{P}^i(\tilde{A}^\delta) + \frac{2\delta}{q_i}$$

To establish the reverse inequality, we use the symmetric LP constraint $\mathbb{P}(A) \leq \mathbb{Q}(A^\delta) + \delta$ yields

$$p_i \, \mathbb{P}^i(\tilde{A}) \leq q_i \, \mathbb{Q}^i(\tilde{A}^\delta) + \delta.$$

Using the bound $p_i \geq q_i - \delta$ (from the marginal constraint) :

$$(q_i - \delta) \, \mathbb{P}^i(\tilde{A}) \leq q_i \, \mathbb{Q}^i(\tilde{A}^\delta) + \delta.$$

Expand the left side and move the $-\delta\mathbb{P}^i(\tilde{A})$ term to the right, and use the fact that $\mathbb{P}^i(\tilde{A}^\delta) \leq 1$

$$q_i \, \mathbb{P}^i(\tilde{A}) \leq q_i \, \mathbb{Q}^i(\tilde{A}^\delta) + \delta(1 + \mathbb{P}^i(\tilde{A})) \leq q_i \, \mathbb{Q}^i(\tilde{A}^\delta) + 2\delta$$

For all $q_i > 0$ and dividing by $q_i$ yields

$$\mathbb{P}^i(\tilde{A}) \leq \mathbb{Q}^i(\tilde{A}^\delta) + \frac{2\delta}{q_i}$$

Finally, to strictly satisfy the definition of the Lévy-Prokhorov metric, the spatial expansion radius and the additive penalty must match. Since $q_i \leq 1$, we have $\delta \leq \frac{2\delta}{q_i}$, meaning $\tilde{A}^\delta \subseteq \tilde{A}^{\frac{2\delta}{q_i}}$. By the monotonicity of probability measures, $\mathbb{P}^i(\tilde{A}^\delta) \leq \mathbb{P}^i(\tilde{A}^{\frac{2\delta}{q_i}})$. Substituting this into our bounds yields:

$$\mathbb{Q}^i(\tilde{A}) \leq \mathbb{P}^i(\tilde{A}^{\frac{2\delta}{q_i}}) + \frac{2\delta}{q_i} \quad \text{and} \quad \mathbb{P}^i(\tilde{A}) \leq \mathbb{Q}^i(\tilde{A}^{\frac{2\delta}{q_i}}) + \frac{2\delta}{q_i}.$$

This implies that each conditional law $\mathbb{P}^i$ lies in a robust neighborhood of $\mathbb{Q}^i$ under radius $\frac{2\delta}{q_i}$, proving $\mathbb{P}^i \in B_{\mathrm{LP}, \frac{2\delta}{q_i}}(\mathbb{Q}^i)$.

It follows that

$$\{\mathbb{P}(y \mid \bar{\mathbf{x}}) \mid \mathbb{P}_{XY} \in \mathcal{P}_{\mathrm{BinPKL}}(\epsilon, \delta; J)\} \subseteq \mathcal{K}'(\gamma, \delta; J) \tag{Left}$$

**Sufficient Conditions (Right Inclusion by Construction)**   We now show that for any target conditional law $\mathbb{P}^1 \in B_{\mathrm{LP}, \frac{2\delta}{q_1}}(\mathbb{Q}^1)$ at covariate $\bar{\mathbf{x}} = 1$ (notation with convention), there exists a valid joint distribution $\mathbb{P}_{XY}$ within $B_{\mathrm{LP}, 3\delta}(\mathbb{Q}_{XY})$ that harbors this conditional.

Assume $q_1 \geq \delta$ (if $q_1 < \delta$, the margin of error $\frac{2\delta}{q_1} > 2$, meaning the target neighborhood contains all probability measures, trivializing the bound).If $|\mathcal{X}| = 1$, the construction is immediate by taking $p_1 = q_1 = 1$ and $P^1$ as the target conditional law, since $P^1 \in B_{\mathrm{LP}, 2\delta}(Q^1) \subseteq B_{\mathrm{LP}, 3\delta}(Q^1)$. Hence, we assume $|\mathcal{X}| \geq 2$ and there is another state $i = 2$ to absorb mass. Construct the mixture $\mathbb{P}$ as follows:

$$
\begin{aligned}
p_1 &= q_1 - \delta, \quad \mathbb{P}^1 \text{ is the target distribution,} \\
p_2 &= q_2 + \delta, \quad \mathbb{P}^2 = \mathbb{Q}^2, \\
p_i &= q_i, \quad \mathbb{P}^i = \mathbb{Q}^i \quad \text{for all (if exists) } i \geq 3.
\end{aligned}
$$

This defines a valid distribution. Indeed, $p_1 \geq 0$ because $q_1 \geq \delta$, and $p_2 = q_2 + \delta \leq q_2 + q_1 \leq 1$ again because $q_1 \geq \delta$. All other coordinates are unchanged, and the total mass is preserved since the amount $\delta$ removed from coordinate 1 is added to coordinate 2:

$$
\sum_{i \in \mathcal{S}} p_i = (q_1 - \delta) + (q_2 + \delta) + \sum_{i \geq 3} q_i = 1
$$

Hence $(p_i)_{i \in |X|}$ is a probability vector.

To verify $\mathbb{P} \in B_{\mathrm{LP}, 3\delta}(\mathbb{Q})$, we first evaluate the one-sided bound $\mathbb{Q}(A) - \mathbb{P}(A^\delta)$ for an arbitrary Borel set $A \subseteq \mathcal{S} \times \mathcal{Y}$. Expanding the difference yields:

$$
\mathbb{Q}(A) - \mathbb{P}(A^\delta) = \sum_{i \in \mathcal{S}} \left[ q_i \mathbb{Q}^i(A_i) - p_i \mathbb{P}^i(A_i^\delta) \right].
$$

Substituting our construction (noting that terms for $i \geq 3$ perfectly cancel):

$$
\begin{aligned}
\mathbb{Q}(A) - \mathbb{P}(A^\delta) &= q_1 \mathbb{Q}^1(A_1) - (q_1 - \delta)\mathbb{P}^1(A_1^\delta) + q_2 \mathbb{Q}^2(A_2) - (q_2 + \delta)\mathbb{Q}^2(A_2^\delta) \\
&= q_1 \left[ \mathbb{Q}^1(A_1) - \mathbb{P}^1(A_1^\delta) \right] + \delta\mathbb{P}^1(A_1^\delta) + q_2 \left[ \mathbb{Q}^2(A_2) - \mathbb{Q}^2(A_2^\delta) \right] - \delta\mathbb{Q}^2(A_2^\delta).
\end{aligned}
$$

Because $A_2 \subseteq A_2^\delta$, by the monotonicity of probability measures, $\mathbb{Q}^2(A_2) - \mathbb{Q}^2(A_2^\delta) \leq 0$. Additionally, $-\delta\mathbb{Q}^2(A_2^\delta) \leq 0$. Dropping these non-positive terms establishes an upper bound:

$$
\mathbb{Q}(A) - \mathbb{P}(A^\delta) \leq q_1 \left[ \mathbb{Q}^1(A_1) - \mathbb{P}^1(A_1^\delta) \right] + \delta\mathbb{P}^1(A_1^\delta).
$$

By our initial hypothesis for the conditional law $\mathbb{P}^1$, we know $\mathbb{Q}^1(A_1) - \mathbb{P}^1(A_1^\delta) \leq \frac{2\delta}{q_1}$. Substituting this alongside the probability bound $\mathbb{P}^1(A_1^\delta) \leq 1$:

$$
\mathbb{Q}(A) - \mathbb{P}(A^\delta) \leq q_1 \left( \frac{2\delta}{q_1} \right) + \delta(1) = 3\delta.
$$

Finally, since $\delta \leq 3\delta$, we have $A^\delta \subseteq A^{3\delta}$, which implies $\mathbb{P}(A^\delta) \leq \mathbb{P}(A^{3\delta})$. Therefore, $\mathbb{Q}(A) - \mathbb{P}(A^{3\delta}) \leq \mathbb{Q}(A) - \mathbb{P}(A^\delta) \leq 3\delta$. The reverse inequality is obtained by the same calculation, using the other half of $P^1 \in B_{\mathrm{LP}, 2\delta/q_1}(Q^1)$, namely $P^1(A_1) \leq Q^1(A_1^{2\delta/q_1}) + 2\delta/q_1$. By symmetry, we can also derive $\mathbb{P}(A) - \mathbb{Q}(A^{3\delta}) \leq 3\delta$. This satisfies the definition of the Lévy-Prokhorov metric ball, confirming $\mathbb{P} \in B_{\mathrm{LP}, 3\delta}(\mathbb{Q})$.

Thus

$$
\mathcal{K}'(\gamma, \delta; J) \subseteq \{\mathbb{P}(y \mid \bar{\mathbf{x}}) \mid \mathbb{P}_{XY} \in \mathcal{P}_{\mathrm{BinPKL}}(\epsilon, 3\delta; J)\} \tag{Right}
$$

Combining (Left) and (Right), we obtain the desired result.

$\square$

### C.7. Proof of Proposition 3.12

We begin with the following representation, which simplifies the computation of the supremum over an LP ball.

**Lemma C.4** (CVaR Representation, see e.g. (Bennouna & Van Parys, 2022, Theorem A.1) ). *Let $\delta \in [0, 1]$ and $\mathbb{Q}(y|\bar{\mathbf{x}})$ be a nominal distribution. Then, for any measurable cost function $c(z, \cdot) : \mathcal{Y} \to \mathbb{R}$ and any fixed $\bar{\mathbf{x}} \in \mathcal{X}$,*

$$\sup \left\{ \mathbb{E}_{\mathbb{P}}[c(z, Y) \mid X = \bar{\mathbf{x}}] \mid \mathbb{P}(y \mid \bar{\mathbf{x}}) \in B_{LP,\delta}(\mathbb{Q}(y \mid \bar{\mathbf{x}})) \right\}$$
$$= (1 - \delta) \, \text{CVaR}^{\delta}_{\mathbb{Q}(y|\bar{\mathbf{x}})}(c^{\mathcal{N}}(z, Y)) + \delta \cdot \max_{y \in \mathcal{Y}} c(z, y),$$

*where $\mathcal{N} := \mathbb{B}_1(0, \delta)$ denotes the closed $\ell_1$-norm ball of radius $\delta$, and the inflated cost function is defined as*

$$c^{\mathcal{N}}(z, Y) := \sup \left\{ c(z, Y - \Delta) \mid \Delta \in \mathcal{N}, \, Y - \Delta \in \mathcal{Y} \right\}.$$

Fix $z \in \mathcal{Z}$. Now we consider the following optimization problem, where $\mathbb{Q}$ denotes a candidate probability measure over the joint space of $(x, y)$. In particular, $\mathbb{Q}(x_j)$ represents the marginal probability of covariate $x_j$, and $\mathbb{Q}(y \mid x_j)$ denotes the conditional probability measure over outcomes given $x_j$.

$$\sup_{\mathbb{Q}} \quad (1 - \delta) \, \text{CVaR}^{\delta}_{\mathbb{Q}(y|\bar{\mathbf{x}})} \left( c^{\mathcal{N}}(z, Y) \right) \ + \ \delta \cdot \max_{y \in \mathcal{Y}} c(z, y)$$

$$\text{s.t.} \quad \frac{1}{n} \sum_{i=1}^{n} \log \left( \sum_{x_{m,i} \in \mathcal{A}(\tilde{x}_{m,i})} \mathbb{Q}(y_i \mid x_{o,i}, x_{m,i}) \cdot \mathbb{Q}(x_{o,i}, x_{m,i}) \right) \geq \gamma,$$

$$\sum_{j=1}^{|\mathcal{X}|} \mathbb{Q}(x_j) = 1, \quad \mathbb{Q}(x_j) \geq 0, \quad \forall j \in |\mathcal{X}|,$$

$$\mathbb{Q}(y \mid x_j) \in \mathcal{P}(\mathcal{Y}), \mathbb{Q}(y \mid x) = \mathbb{Q}(y' \mid x), \forall y, y' \in \mathcal{Y}_j, \forall j \quad \mathbb{Q}(A \mid x_j) \geq 0, \quad \forall j \in |\mathcal{X}|, \, \forall A \in \mathcal{B}(\mathcal{Y}).$$

$$(10)$$

Let $S_{\text{obs}} := \{y_1, \ldots, y_K\}$. Choose $y^{\mathcal{N}}_{\infty}(z) \in \arg\max_{y \in \mathcal{Y}} c(z, y)$, and let $j_{\infty}$ be such that $y^{\mathcal{N}}_{\infty}(z) \in \mathcal{Y}_{j_{\infty}}$. Since $\Delta = 0$ is feasible in the definition of $c^{\mathcal{N}}$, we have

$$c^{\mathcal{N}}(z, y^{\mathcal{N}}_{\infty}(z)) = \max_{y \in \mathcal{Y}} c^{\mathcal{N}}(z, y) = \max_{y \in \mathcal{Y}} c(z, y).$$

We show that every feasible $\mathbb{Q}$ in (10) can be replaced by another feasible distribution $\widetilde{\mathbb{Q}}$ with no smaller objective value and such that

$$\text{supp}\left( \widetilde{\mathbb{Q}}(\cdot \mid \bar{\boldsymbol{x}}) \right) \subseteq \underbrace{\{y_1, \ldots, y_K\}}_{S_{\text{obs}}} \cup \{y^{\mathcal{N}}_{\infty}(z)\}.$$

The transformation keeps the marginal law $\mathbb{Q}_X$ fixed and only redistributes the conditional outcome mass within each $\mathbb{Q}(\cdot \mid x)$.

Define the auxiliary part of the support by

$$S_{\text{aux}} := \mathcal{Y} \setminus (S_{\text{obs}} \cup \{y^{\mathcal{N}}_{\infty}(z)\})$$

We distinguish two cases.

1. **Suppose first that** $S_{\text{obs}} \cap \mathcal{Y}_{j_{\infty}} = \varnothing$. For each $x \in \mathcal{X}$, construct $\widetilde{\mathbb{Q}}(\cdot \mid \bar{x})$ by keeping the probabilities of all observed outcomes unchanged, setting the mass of all auxiliary points to zero, and adding the total mass originally assigned to $S_{\text{aux}}$ to $y^{\mathcal{N}}_{\infty}(z)$. Since the bin containing $y^{\mathcal{N}}_{\infty}(z)$ contains no observed outcome, no probability appearing in the empirical part changes. Hence the log-sum-exp constraint remains satisfied. The within-bin equality constraint is also preserved: points assigned zero mass are removed from the support, and the only newly active point in $\mathcal{Y}_{j_{\infty}}$ is $y^{\mathcal{N}}_{\infty}(z)$. Moreover, all removed mass is moved to a point with maximal inflated cost. Therefore the loss distribution under $\widetilde{\mathbb{Q}}(\cdot \mid \bar{x})$ first-order stochastically dominates that under $\mathbb{Q}(\cdot \mid \bar{x})$, and by monotonicity of CVaR the objective does not decrease.

2. **Suppose now that** $\mathcal{S}_{\text{obs}} \cap \mathcal{Y}_{j_\infty} \neq \varnothing$. In this case, moving all auxiliary mass only to $y_\infty^{\mathcal{N}}(z)$ may violate the within-bin equality constraint. Instead, for each $x \in \mathcal{X}$, we set the mass of all points in $\mathcal{S}_{\text{aux}} \cup \{y_\infty^{\mathcal{N}}(z)\}$ to zero, keep the masses of observed outcomes outside $\mathcal{Y}_{j_\infty}$ unchanged, and add to each observed outcome in $\mathcal{S}_{\text{obs}} \cap \mathcal{Y}_{j_\infty}$ an equal share of the total mass originally assigned to $\mathcal{S}_{\text{aux}} \cup \{y_\infty^{\mathcal{N}}(z)\}$. This preserves the simplex and nonnegativity constraints. It also preserves the within-bin equality constraint, because all observed outcomes in $\mathcal{Y}_{j_\infty}$ receive the same additional mass, while the other observed bins are unchanged.

The likelihood constraint is weakly improved. Observed outcomes outside $\mathcal{Y}_{j_\infty}$ keep the same conditional probabilities, while observed outcomes inside $\mathcal{Y}_{j_\infty}$ receive weakly larger conditional probabilities. Hence every term inside the empirical log-likelihood is weakly larger under $\widetilde{\mathbb{Q}}$ than under $\mathbb{Q}$.

It remains to check the objective. Since $\delta \geq h_J$, every $y \in \mathcal{Y}_{j_\infty}$ satisfies $\|y - y_\infty^{\mathcal{N}}(z)\|_1 \leq h_J \leq \delta$. Therefore $y_\infty^{\mathcal{N}}(z)$ is feasible in the definition of $c^{\mathcal{N}}(z, y)$, which implies

$$c^{\mathcal{N}}(z, y) = \max_{u \in \mathcal{Y}} c(z, u), \qquad \forall y \in \mathcal{Y}_{j_\infty}$$

Thus every observed outcome in $\mathcal{Y}_{j_\infty}$ has the same maximal inflated cost as $y_\infty^{\mathcal{N}}(z)$. Mass removed from outside $\mathcal{Y}_{j_\infty}$ is moved to points with maximal inflated cost, and mass removed from inside $\mathcal{Y}_{j_\infty}$ is moved to points with the same inflated cost. Therefore the transformed loss distribution under $\widetilde{\mathbb{Q}}(\cdot \mid \bar{x})$ first-order stochastically dominates the original one, and the CVaR term does not decrease.

Combining the two cases, every feasible distribution $\mathbb{Q}$ can be replaced by a feasible distribution $\widetilde{\mathbb{Q}}$ with no smaller objective value and with conditional outcome support contained in $\mathcal{S}_{\text{obs}} \cup \{y_\infty^{\mathcal{N}}(z)\}$. Therefore, under the within-bin equality constraint and the condition $\delta \geq h_J$, it is without loss of optimality to restrict attention to distributions supported on $\{y_1, \ldots, y_K\} \cup \{y_\infty^{\mathcal{N}}(z)\}$.

### C.8. Proof of Theorem 3.13

Now we introduce the following variable substitutions: Let $B_J : \widehat{\mathcal{Y}} \to [J]$ denote the bin map, i.e., $B_J(y) = j$ if $y \in \mathcal{Y}_j$, and let

$$\widehat{\mathcal{J}} := \{j \in [J] : \widehat{\mathcal{Y}} \cap \mathcal{Y}_j \neq \varnothing\}$$

be the set of bins occupied by the empirical support. Because the projected law is constrained to take the same value within each outcome bin for a fixed covariate value, we introduce one variable per covariate-bin pair:

$$v_{x,j} := \frac{\mathbb{Q}(x, y)}{\mathbb{Q}(\bar{x})}, \qquad x \in \mathcal{X},\ j \in \widehat{\mathcal{J}}, \forall\, y \in \widehat{\mathcal{Y}} \cap \mathcal{Y}_j$$

We also define

$$t := \frac{1}{\mathbb{Q}(\bar{x})}, \qquad u_i := \frac{\widehat{\mathbb{Q}}(\tilde{x}_i, y_i)}{\mathbb{Q}(\bar{x})} = \sum_{x_{m,i} \in \mathcal{A}(\tilde{x}_{m,i})} v_{(x_{o,i}, x_{m,i}), B_J(y_i)}, \qquad i = 1, \ldots, n$$

and we introduce the same matrix $T$ and vector $d$ as defined in the discrete case now with dimensions $T \in \mathbb{R}^{n \times |\mathcal{X} \times \widehat{\mathcal{J}}|}$ and $d \in \mathbb{R}^{|\mathcal{X} \times \widehat{\mathcal{J}}|}$.

Define $e \in \mathbb{R}^{|\mathcal{X}||\widehat{\mathcal{J}}|}$ for each x and bin $j \in \widehat{\mathcal{J}}$ as

$$e_{x,j} := \#\{(x, y) \in \mathcal{Y}_j \text{ for some } y \in \widehat{\mathcal{Y}}\}.$$

We also introduce $w := \mathbb{Q}(y_\infty \mid \bar{x})$. Let $j_\infty := B_J(y_\infty^{\mathcal{N}}(z))$. If $j_\infty \in \widehat{\mathcal{J}}$, then the within-bin equality constraint implies $w = v_{\bar{x}, j_\infty}$.

The feasible region can be reformulated as:

$$\sum_{i=1}^{n} \log\left(\frac{u_i}{t}\right) \geq n\gamma$$
$$u = Tv$$
$$w = v_{\bar{x},j_\infty}$$
$$e^\top v + w \leq t$$
$$d^\top v + w = 1$$
$$t \geq 0, \quad v \succeq 0, \quad u \succeq 0, \quad w \geq 0$$

This constraint can be equivalently expressed as: $n\gamma t + t\sum_{i=1}^{n} \log\left(\frac{t}{u_i}\right) \leq 0$

The objective function can be written as: $\sum_{k=1}^{K} s_k \cdot c^{\mathcal{N}}(z, y_k) + (\delta + s_\infty) \cdot \max_{y \in \mathcal{Y}} c(z, y)$ subject to:

$$\sum_{k=1}^{K} s_k + s_\infty = 1 - \delta$$
$$s_k \leq v(y_k \mid \bar{\mathbf{x}}), \quad s_\infty \leq w$$
$$s_k \geq 0, \quad s_\infty \geq 0, \quad \forall k \in \{1, \ldots, K\}$$

Combine them together, we get a convex representation of primal problem of inner sup: Notice that for fixed z, the inflated unit costs $c^{\mathcal{N}}(z, y_k)$ as well as $\max_{y \in \mathcal{Y}} c(z, y)$ are constants

$$\sup_{\Theta_1} \quad \sum_{k=1}^{K} s_k \cdot c^{\mathcal{N}}(z, y_k) + (\delta + s_\infty) \cdot \max_{y \in \mathcal{Y}} c(z, y)$$

$$\text{s.t.} \quad n\gamma t + t\sum_{i=1}^{n} \log\left(\frac{t}{u_i}\right) \leq 0$$

$$u = Tv, \quad e^\top v + w \leq t, \quad d^\top v + w = 1 \tag{11}$$

$$\sum_{k=1}^{K} s_k + s_\infty = 1 - \delta$$

$$s_k \leq v(y_k \mid \bar{\mathbf{x}}), \quad s_\infty \leq w$$

$$w = v_{\bar{x},j_\infty}$$

where $\Theta_1 := \{t \geq 0, v \succeq 0, u \succeq 0, w \geq 0, s_k \geq 0 \quad \forall k \in \{1, \ldots, K\}, s_\infty \geq 0\}$

**Observation.** Problem (11) is a convex optimization problem

- The objective function is linear in $(s_i, s_\infty)$.

- The constraint $n\gamma t + t\sum_i \log(t/u_i) \leq 0$ is jointly convex in $(u, t)$ because the function $t\log(t/u)$ is jointly convex over $\mathbb{R}_{++} \times \mathbb{R}_{++}$.

- All other constraints are affine or conic in the decision variables $(v, u, t, w, s, s_\infty)$.

This motivates us to re-express the model in a dual form. For simplicity , we write $w = f^T v$ to represent $w = v_{\bar{x},j_\infty}$ . Let $Y_{(j)}$ denotes the outcome region associated with the $j$-th cell $(x, Y)_j$, and $f_j = 1$ if $y^{\mathcal{N}}_\infty(z) \in Y_{(j)}$ otherwise 0

Let $\lambda \geq 0$, $\varphi$, $\mu \geq 0$, $\eta \in \mathbb{R}$, $\rho \in \mathbb{R}$, $\pi_k \geq 0$ for all $k \in K$, and $\pi_\infty \geq 0$, $\zeta \in \mathbb{R}$, denote the dual variables corresponding to the constraints. The Lagrangian of the problem (11) is given by:

$$
L(\lambda, \varphi, \mu, \eta, \rho, \boldsymbol{\pi}, \pi_\infty, \nu_1, \nu_2) = \sum_{k=1}^{K} s_k \cdot c^{\mathcal{N}}(z, y_k) + (\delta + s_\infty) \cdot \max_{y \in \mathcal{Y}} c(z, y)
$$

$$
- \lambda \left( n\gamma t + t \sum_{i=1}^{n} \log\left(\frac{t}{u_i}\right) \right) + \varphi^\top (u - Tv)
$$

$$
- \mu(e^\top v + w - t) + \eta(d^\top v + w - 1)
$$

$$
+ \rho\left( \sum_{k=1}^{K} s_k + s_\infty - 1 + \delta \right) - \sum_{k=1}^{K} \pi_k(s_k - v(y_k \mid \bar{\mathbf{x}})) - \pi_\infty(s_\infty - w) + \zeta(w - f^T v)
$$

The dual function is obtained by maximizing over $\Theta_1 := \{ t \geq 0,\ v \succeq 0,\ u \succeq 0,\ w \geq 0,\ s_k \geq 0,\ s_\infty \geq 0 \}$ and minimizing over the dual variables $\Theta_2 := \{ \lambda \geq 0,\ \varphi,\ \mu \geq 0,\ \eta \in \mathbb{R},\ \rho \in \mathbb{R},\ \pi_k \geq 0,\ \pi_\infty \geq 0, \zeta \in \mathbb{R} \}$:

$$
\min_{\Theta_2}\ g(\lambda, \varphi, \mu, \eta, \rho, \boldsymbol{\pi}, \pi_\infty, \nu_1, \nu_2) = \min_{\Theta_2} \max_{\Theta_1} \left\{ \left( -\varphi^\top T - \mu e^\top + \eta d^\top - \zeta f^\top \right) v + \sum_{k=1}^{K} \pi_k v(y_k \mid \bar{\mathbf{x}}) \right.
$$

$$
+ w(-\mu + \eta + \pi_\infty + \zeta)
$$

$$
+ \sum_{k=1}^{K} s_k \left( c^{\mathcal{N}}(z, y_k) - \pi_k + \rho \right) + s_\infty \left( \max_{y \in \mathcal{Y}} c(z, y) + \rho - \pi_\infty \right)
$$

$$
- \lambda \left( n\gamma t + t \sum_{i=1}^{n} \log\left(\frac{t}{u_i}\right) \right) + \varphi^\top u + \mu t
$$

$$
\left. + \delta \cdot \max_{y \in \mathcal{Y}} c(z, y) - \eta - \rho(1 - \delta) \right\}.
$$

For $\boldsymbol{\nu} \succeq 0$, we can prove equivalence by adding the constraints

$$
-(T^T \varphi)_j - \mu e_j + (\eta + \sum_{k: y_k \in Y_{(j)}} )d_j - \zeta f_j \leq 0, \quad \forall j
$$

Then the term

$$
\left[ -\varphi^T T - \mu e^T + \eta d^T - \zeta f^T \right] v\ +\ \sum_{k=1}^{K} \pi_k\, v(y_k \mid \bar{\mathbf{x}}) \quad \longrightarrow \quad 0
$$

For $w \geq 0$, we pose the condition

$$
-\mu + \eta + \pi_\infty + \zeta \leq 0 \quad \implies \quad w(-\mu + \eta + \pi_\infty + \zeta) \longrightarrow 0
$$

For $s_k \geq 0$ and $\pi_\infty \geq 0$, we pose

$$
c^{\mathcal{N}}(z, y_k) - \pi_k + \rho \leq 0, \quad \forall k \in [K], \qquad \max_{y \in \mathcal{Y}} c(z, y) + \rho - \pi_\infty \leq 0,
$$

then we have

$$
\sum_{k=1}^{K} s_k \left( c^{\mathcal{N}}(z, y_i) - \pi_k + \rho \right)\ +\ s_\infty \left( \max_{y \in \mathcal{Y}} c(z, y) + \rho - \pi_\infty \right) \quad \longrightarrow \quad 0.
$$

We now examine the term involving the log-likelihood constraint:

$$
-\lambda \left( n\gamma t + \sum_{i=1}^{n} t \log\left(\frac{t}{u_i}\right) \right)\ +\ \varphi^\top u + \mu t.
$$

Fix $\lambda \geq 0$ and $t \geq 0$. Define

$$f(u) := -\lambda \sum_{i=1}^{n} t \log\left(\frac{t}{u_i}\right) + \boldsymbol{\varphi}^\top u,$$

with partial derivative

$$f'(u_i) = \lambda \frac{t}{u_i} + \varphi_i.$$

If any $\varphi_i > 0$, then $f(u_i)$ is increasing, and the value diverges as $u_i \to \infty$. Therefore, we must impose: $\boldsymbol{\varphi} \preceq 0$. The maximum is then attained at $u_i^* = -\frac{\lambda t}{\varphi_i}$.

Substituting $u_i^*$ back, the term becomes:

$$-\lambda\left(n\gamma t + \sum_{i=1}^{n} t \log\left(\frac{t}{u_i^*}\right)\right) + \boldsymbol{\varphi}^\top u^* + \mu t$$

$$= -\lambda\left(n\gamma t + \sum_{i=1}^{n} t \log\left(\frac{t}{-\frac{\lambda t}{\varphi_i}}\right)\right) + \sum_{i=1}^{n} \varphi_i \cdot \left(-\frac{\lambda t}{\varphi_i}\right) + \mu t$$

$$= -\lambda\left(n(1+\gamma)t + \sum_{i=1}^{n} \log\left(-\frac{\varphi_i}{\lambda}\right)t\right) + \mu t.$$

To ensure finiteness of the dual function, we must impose:

$$-\lambda\left(n(1+\gamma) + \sum_{i=1}^{n} \log\left(-\frac{\varphi_i}{\lambda}\right)\right) + \mu \leq 0.$$

By reparametrizing with $\varphi_i := -\psi_i$ and requiring $\boldsymbol{\psi} \succeq 0$, we obtain the equivalent constraint:

$$-\lambda\left(n(1+\gamma) + \sum_{i=1}^{n} \log\left(\frac{\psi_i}{\lambda}\right)\right) + \mu \leq 0, \quad \text{with } \psi_i \geq 0.$$

We now combine the Lagrangian dual derived in the previous steps with the original decision variable z. To simplify notation, we define $M := \max_{y \in \mathcal{Y}} c(z, y)$ as the worst-case cost associated with decision z. The original minimax formulation is thus equivalently reformulated as the following optimization problem:

$$\min_{\lambda \geq 0,\, \boldsymbol{\varphi} \succeq 0,\, \mu \geq 0,\, \eta,\, \rho,\, z \in \mathcal{Z},\, M} \quad \delta M - (1-\delta)\rho - \eta$$

$$\begin{aligned}
\text{s.t.} \quad & M = \max_{y \in \mathcal{Y}} c(z, y), \\
& -\mu + \eta + M + \rho + \zeta \leq 0, \\
& -(T^\top \boldsymbol{\varphi})_j - \mu e_j + \left(\eta + \sum_{k: y_k \in Y_{(j)}} \pi_k\right) d_j - \zeta f_j \leq 0, \ \forall j \\
& c^{\mathcal{N}}(z, y_k) - \pi_k + \rho \leq 0, \quad \forall k \in [K] \\
& M + \rho - \nu_1 - \nu_2 \leq 0, \\
& -\lambda\left(n(1+\gamma) + \sum_{i=1}^{n} \log\left(\frac{\psi_i}{\lambda}\right)\right) + \mu \leq 0.
\end{aligned}$$

We relax the equality constraint $M = \max_{y \in \mathcal{Y}} c(z, y)$ to an inequality: $M \geq \max_{y \in \mathcal{Y}} c(z, y)$, which defines the epigraph of the pointwise maximum. This relaxation is justified because $\delta > 0$ and with smaller $M$ the feasible region is strictly

larger.

$$\min_{\lambda \geq 0,\ \boldsymbol{\varphi} \succeq 0,\ \mu \geq 0,\ \eta,\ \rho,\ z \in \mathcal{Z},\ M} \quad \delta M - (1-\delta)\rho - \eta$$

$$\text{s.t.} \quad M \geq \max_{y \in \mathcal{Y}} c(z, y),$$

$$-\mu e_i + \eta + M + \rho + \zeta \leq 0,$$

$$-(T^\top \boldsymbol{\varphi})_j - \mu e_j + \Big(\eta + \sum_{k:y_k \in Y_{(j)}} \pi_k\Big) d_j - \zeta f_j \leq 0,\ \forall j$$

$$c^{\mathcal{N}}(z, y_k) - \pi_k + \rho \leq 0, \quad \forall k \in [K]$$

$$M + \rho - \nu_1 - \nu_2 \leq 0,$$

$$-\lambda \left( n(1 + \gamma) + \sum_{i=1}^n \log\left(\frac{\psi_i}{\lambda}\right) \right) + \mu \leq 0.$$

When the cost function $c(z, y)$ is convex in $z$ for each fixed $y$, since pointwise maxima of convex functions remain convex. In addition, the inflated cost $c^{\mathcal{N}}(z, y_i)$ accounts for uncertainty in the outcome through local perturbations around $y_i$, defined by

$$c^{\mathcal{N}}(z, y_i) := \sup\{c(z, y_i - \Delta) : \Delta \in \mathcal{N},\ y_i - \Delta \in \mathcal{Y}\}$$

When $c(z, y)$ is convex in $z$, this supremum over a fixed perturbation set $\mathcal{N}$ also preserves convexity in $z$, as it represents the pointwise supremum of convex functions indexed by $\Delta$.

Moreover, since $c(z, y)$ is convex in $y$, the maximum over $\mathcal{Y}$ is attained at a boundary point. As $\mathcal{Y}$ is constructed as the outward $h_J$-enlargement of the data support, with $h_J = \max_j \operatorname{diam}(\mathcal{Y}_j)$, the bin $j_\infty$ containing any maximizer $y_\infty(z)$ is disjoint from the observed-bin set $\widehat{\mathcal{J}}$. Therefore $j_\infty \notin \widehat{\mathcal{J}}$, and the term $f$ is identically zero.

Thus, under the mild assumption the entire reformulated problem admits a convex program representation.

### C.9. Proof of Proposition 3.14

We begin by providing the following preparatory materials.

**Lemma C.5** (Dembo, 2009). *For any measurable set $\mathcal{A} \subset \Delta(\mathcal{S})$ and empirical distribution $\widehat{\mathbb{P}}_n$ drawn i.i.d. from the true law $\mathbb{P}$,*

$$\Pr\big(\widehat{\mathbb{P}}_n \in \mathcal{A}\big) \ \leq \ m_{\mathrm{LP}}(\mathcal{A}, \zeta) \exp\Big(-n \inf_{\mathbb{P}' \in \mathcal{A}^\zeta} D_{\mathrm{KL}}(\mathbb{P}' \,\|\, \mathbb{P})\Big),$$

*where $m_{\mathrm{LP}}(\mathcal{A}, \zeta)$ is the covering number of $\mathcal{A}$ by Lévy–Prokhorov balls of radius $\zeta$, and*

$$\mathcal{A}^\zeta := \big\{\mathbb{P}' : \pi_{\mathrm{LP}}(\mathbb{P}', \mathbb{P}'') \leq \zeta,\ \mathbb{P}'' \in \mathcal{A}\big\}.$$

**Lemma C.6** (Dembo, 2009). *Let $\Sigma$ be a compact metric space and $M_1(\Sigma)$ the set of all Borel probability measures on $\Sigma$. Then, for any set $\mathcal{A} \subset \mathcal{M}_1(\Sigma)$ and any $\zeta \in (0, 1)$,*

$$m_{\mathrm{LP}}(\mathcal{A}, \zeta) \ \leq \ m_{\mathrm{LP}}(\mathcal{M}_1(\Sigma), \zeta) \ \leq \ \left(\tfrac{4}{\zeta}\right)^{m(\Sigma, \zeta)},$$

*where $m(\Sigma, \zeta)$ is the metric entropy, i.e. the minimal number of balls of radius $\zeta$ that cover $\Sigma$*

*Proof.* Proof Outline of Proposition 3.14: We aim to establish a lower bound for the probability

$$\Pr\Big\{\mathbb{P}^* \in \big\{\mathbb{P} : \exists \mathbb{Q},\ D_{\mathrm{KL}}(\widehat{\mathbb{P}}, \mathbb{Q}) \leq \varepsilon,\ \pi_{\mathrm{LP}, B(0,\delta)}(\mathbb{P}, \mathbb{Q}) \leq \delta\big\}\Big\} = \Pr\Big\{\exists \mathbb{Q},\ D_{\mathrm{KL}}(\widehat{\mathbb{P}}, \mathbb{Q}) \leq \varepsilon,\ \pi_{\mathrm{LP}, B(0,\delta)}(\mathbb{P}^*, \mathbb{Q}) \leq \delta\Big\}$$

which can be rewritten as

$$\Pr\Big\{\widehat{\mathbb{P}} \in \big\{\mathbb{P} : \exists \mathbb{Q},\ D_{\mathrm{KL}}(\mathbb{P}, \mathbb{Q}) \leq \varepsilon,\ \pi_{\mathrm{LP}, B(0,\delta)}(\mathbb{P}^*, \mathbb{Q}) \leq \delta\big\}\Big\}.$$

For brevity, denote

$$\mathcal{A}^c := \Big\{ \mathbb{P} : \exists \mathbb{Q}, \ D_{\mathrm{KL}}(\mathbb{P}, \mathbb{Q}) \le \varepsilon, \ \pi_{\mathrm{LP}, B(0,\delta)}(\mathbb{P}^*, \mathbb{Q}) \le \delta \Big\}.$$

Our task is therefore to control the tail probability $\Pr\{\widehat{\mathbb{P}} \in \mathcal{A}\}$. Combining Lemma C.5 with Lemma C.6, the task reduces to bounding $\inf\limits_{\mathbb{P}' \in \mathcal{A}^\varsigma} D_{\mathrm{KL}}(\mathbb{P}' \parallel \mathbb{P})$. Choosing $\varsigma = \frac{\delta}{2}$ one shows that

$$\inf_{\mathbb{P}' \in \mathcal{A}^{\frac{\delta}{2}}} D_{\mathrm{KL}}(\mathbb{P}' \parallel \mathbb{P}) \ \ge \ \frac{\delta^2}{2} \left( e^{-\varepsilon} + e^{-2\delta^2} \right)^{-1}$$

**Proof detail.** Assume $\mathbb{P}'' \in \mathcal{A}^{\frac{\delta}{2}}$ and let us derive a contradiction under the counter-hypothesis $D_{\mathrm{KL}}(\mathbb{P}'', \mathbb{P}) \le t$ for a parameter $t > 0$ to be chosen.

As a preparation, we first write a target contradicting event as

$$\mathbb{P}'' : \ \forall \mathbb{P}''' \in B_{\mathrm{LP}, \varsigma}(\mathbb{P}''), \ \mathbb{P}''' \in \mathcal{A}^c \iff \forall \mathbb{P}''' \in B_{\mathrm{LP}, \varsigma}(\mathbb{P}'') : \exists \mathbb{Q}, \ D_{\mathrm{KL}}(\mathbb{P}''', \mathbb{Q}) \le \varepsilon, \ \pi_{\mathrm{LP}}(\mathbb{Q}, \mathbb{P}) \le \delta, \quad (12)$$

and carry out the following steps:

From $D_{\mathrm{KL}}(\mathbb{P}'', \mathbb{P}) \le t$ we have the Pinsker bound $\pi_{\mathrm{LP}}(\mathbb{P}'', \mathbb{P}) \le \sqrt{t/2}$.

For any $\mathbb{P}''' \in B_{\mathrm{LP}, \varsigma}(\mathbb{P}'')$, it follows by the triangle inequality,

$$\pi_{\mathrm{LP}}(\mathbb{P}''', \mathbb{P}) \ \le \ \sqrt{t/2} + \varsigma$$

Construct the mixture $\mathbb{Q} := (1 - \alpha)\mathbb{P}''' + \alpha\mathbb{P}$ with $\alpha \in (0, 1)$. Direct calculation shows

$$D_{\mathrm{KL}}(\mathbb{P}''', \mathbb{Q}) = \int \mathbb{P}''' \log \frac{\mathbb{P}'''}{(1 - \alpha)\mathbb{P}''' + \alpha\mathbb{P}} \ = \ -\log(1 - \alpha),$$
$$\text{and choosing } \alpha := 1 - e^{-\varepsilon} \text{ yields } D_{\mathrm{KL}}(\mathbb{P}''', \mathbb{Q}) = \varepsilon$$

Moreover, Pinsker's inequality gives

$$\pi_{\mathrm{LP}}(\mathbb{P}, \mathbb{Q}) \ \le \ \sqrt{\tfrac{1}{2} D_{\mathrm{KL}}(\mathbb{P}, \mathbb{Q})} \ \le \ \sqrt{\frac{1}{2} \log \frac{e^\varepsilon}{e^\varepsilon - 1}}$$

Using $\pi_{\mathrm{LP}}(\mathbb{Q}, \mathbb{P}) \le \max\{\pi_{\mathrm{LP}}(\mathbb{P}''', \mathbb{P}), \pi_{\mathrm{LP}}(\mathbb{P}, \mathbb{P})\} = \pi_{\mathrm{LP}}(\mathbb{P}''', \mathbb{P})$, we obtain

$$\pi_{\mathrm{LP}}(\mathbb{Q}, \mathbb{P}) \ \le \ \min\Big\{ \sqrt{\tfrac{1}{2} \log \tfrac{e^\varepsilon}{e^\varepsilon - 1}}, \sqrt{\tfrac{t}{2}} + \varsigma \Big\}.$$

Setting $\varsigma = \delta/2$, and we require $\min\{\ \cdot\ , \ \cdot\ \} \le \delta$. This is ensured by taking, e.g.

$$t \ = \ \frac{\delta^2}{2} \left( e^{-\varepsilon} + e^{-2\delta^2} \right)^{-1}.$$

With this choice of $t$, the assumption $D_{\mathrm{KL}}(\mathbb{P}'', \mathbb{P}) \le t$ is sufficient for event (12) (since we can construct such a $\mathbb{Q}$ for any $\mathbb{P}'''$) , which contradicts the fact that $\mathbb{P}'' \in \mathcal{A}^{\frac{\delta}{2}} = \mathcal{A}^\varsigma$. This proves that any $\mathbb{P}'' \in \mathcal{A}^{\delta/2}$ must satisfy $D_{\mathrm{KL}}(\mathbb{P}'', \mathbb{P}) \ge \frac{\delta^2}{2}(e^{-\varepsilon} + e^{-2\delta^2})^{-1}$.

It follows that

$$\Pr(\widehat{\mathbb{P}}_n \in \mathcal{A}) \ \le \ \left( \tfrac{8}{\delta} \right)^{m(\Sigma, \delta)} \exp\Big( -n \frac{\delta^2}{2} \left( e^{-\varepsilon} + e^{-2\delta^2} \right)^{-1} \Big)$$

We now have a closer look at the metric entropy in our case. We equip

$$\Sigma \ = \ \mathcal{S} = \mathcal{X} \times \mathcal{Y}, \quad d\big((x,y),(x',y')\big) \ = \ \mathbf{1}\{x \ne x'\} + \|y - y'\|,$$

with the $\delta$-covering machinery.

Since we assume $\mathcal{X} \subset \mathbb{R}^{d_x}$ is finite , $\mathcal{Y} \subset \mathbb{R}^{d_y}$ is bounded with Euclidean diameter $diam(\mathcal{Y}) := \sup_{y,y' \in \mathcal{Y}} \|y - y'\| < \infty$. For radii $\delta < 1$ the discrete coordinates cannot be mixed inside a single ball, so

$$m(\Sigma, \delta) = |\mathcal{X}| \, N(\mathcal{Y}, \delta), \qquad m(\mathcal{Y}, \delta) \leq \left(1 + \tfrac{diam(\mathcal{Y})}{\delta}\right)^{d_y}.$$

Hence

$$m(\Sigma, \delta) \ \leq \ |\mathcal{X}| \left(1 + \frac{diam(\mathcal{Y})}{\delta}\right)^{d_y}, \ \ 0 < \delta < 1.$$

Insert the covering bound into the deviation estimate, and we obtain the desired result. $\qquad\square$

## C.10. Proof of Theorem 3.15

*Proof.* Recall the containment result established in Proposition 3.11, which ensures that

$$\{\mathbb{P}(Y \mid \bar{\mathbf{x}}) \mid \mathbb{P}_{XY} \in \mathcal{P}_{\mathrm{BinPKL}}(\varepsilon, \delta \cdot \epsilon_1; J)\} \ \subseteq \ \mathcal{K}(\gamma, 2\delta; J).$$

We next establish that the following implication holds:

$$\left\{ D_{\mathrm{RKL}}\big(\widehat{\mathbb{P}}_{XY}, B_{\mathrm{LP}}(\mathbb{P}^*_{XY}, \delta)\big) \leq \varepsilon \right\} \implies \left\{ \mathbb{P}^*_{XY} \in \mathcal{P}_{\mathrm{BinPKL}}(\varepsilon, \delta + h_J) \right\} \qquad \text{(includsion)}$$

*Proof of the claim* (includsion). We begin by interpreting the left–hand side event. By definition of the robust KL divergence over an LP neighborhood,

$$D_{\mathrm{RKL}}\big(\widehat{\mathbb{P}}_{XY}, B_{\mathrm{LP}}(\mathbb{P}^*_{XY}, \delta)\big) \leq \varepsilon$$

is equivalent to the existence of a distribution $\mathbb{P}_{XY}$ such that

$$\mathrm{LP}(\mathbb{P}_{XY}, \mathbb{P}^*_{XY}) \leq \delta, \qquad D_{\mathrm{KL}}\Big(\widehat{\mathbb{P}}_{XY} \,\big\|\, \mathbb{P}_{XY}\Big) \leq \varepsilon.$$

Hence, on the event on the left-hand side of (includsion), there exists at least one joint distribution $\mathbb{P}_{XY}$ satisfying the above two conditions.

Fix such a distribution $\mathbb{P}_{XY}$ and find a construction of missingness mechanism $q_{M|XY} \in \mathcal{Q}_{\mathrm{MAR}}$ . Define a missingness mechanism $\widehat{\mathbb{P}}_{M|XY} \in \mathcal{Q}_{\mathrm{MAR}}$ : For every $(x_o, y)$ that occurs in the data , set the missingness mechanism

$$\widehat{\mathbb{P}}(M = m \mid X_o = x_o, Y = y) := \frac{\sum_i \mathbf{1}\{m_i = m, x_{o,i} = x_o, y_i = y\}}{\sum_i \mathbf{1}\{x_{o,i} = x_o, y_i = y\}}, \quad m \in \mathcal{M}, \qquad \text{(construction)}$$

For any $(x_o, y)$ not appearing in the sample, i.e., $\{i : x_{o,i} = x_o, \ y_i = y\} = \varnothing$, choose an arbitrary distribution on $\mathcal{M}$. The KL divergence between the joint observable distributions admits the decomposition

$$D_{\mathrm{KL}}\Big(\widehat{\mathbb{P}}_{\tilde{X},Y,M} \,\big\|\, \mathrm{H}(\mathbb{P}_{XY}, \widehat{\mathbb{P}}_{M|XY})\Big) = D_{\mathrm{KL}}\Big(\widehat{\mathbb{P}}_{\tilde{X},Y} \,\big\|\, \mathbb{P}_{\tilde{X}Y}\Big) \leq D_{\mathrm{KL}}\Big(\widehat{\mathbb{P}}_{X,Y} \,\big\|\, \mathbb{P}_{XY}\Big) \leq \varepsilon$$

The first equality is obtained by the KL chain rule. Since the NA pattern in $\widetilde{X}$ uniquely determines the mask $M$, there exists a deterministic map $h$ such that $M = h(\widetilde{X}, Y)$. Hence both $\widehat{\mathbb{P}}_{\widetilde{X},Y,M}$ and $\mathrm{H}(\mathbb{P}_{XY}, \widehat{q})$ are obtained by applying the same deterministic augmentation $(\widetilde{X}, Y) \mapsto (\widetilde{X}, Y, h(\widetilde{X}, Y))$ to their $(\widetilde{X}, Y)$-marginals. By the chain rule for KL divergence, the conditional KL term is zero, and thus

$$D_{\mathrm{KL}}\Big(\widehat{\mathbb{P}}_{\widetilde{X},Y,M}\,\Big\|\,\mathrm{H}(\mathbb{P}_{XY}, \widehat{q})\Big) = D_{\mathrm{KL}}\Big(\widehat{\mathbb{P}}_{\widetilde{X},Y}\,\Big\|\,\mathbb{P}_{\widetilde{X},Y}\Big).$$

The second inequality follows from the data-processing inequality, since $\widehat{\mathbb{P}}_{\widetilde{X},Y}$ and $\mathbb{P}_{\widetilde{X},Y}$ are obtained from $\widehat{\mathbb{P}}_{X,Y}$ and $\mathbb{P}_{XY}$ through the same coarsening kernel induced by $\widehat{q}$.

We note that $P_{XY}$ is not our target, because in the definition of BinPKL, the reference distribution must lie in the $B_J$ class. Therefore, based on $P_{XY}$, we perform the following transformation: Let $B_J : \mathcal{Y} \to [J]$ denote the binning map induced by the partition $\{\mathcal{Y}_j\}_{j=1}^J$, i.e., $B_J(y) = j \iff y \in \mathcal{Y}_j$. Applying this map to the observed samples yields

$$(\widetilde{X}_i, Y_i, M_i) \longmapsto (\widetilde{X}_i, B_J(Y_i), M_i).$$

Since $D_{\mathrm{KL}}\left(\widehat{\mathbb{P}}_{\tilde{X},Y,M} \,\big\|\, \mathrm{H}(\mathbb{P}_{XY}, \widehat{\mathbb{P}}_{M|XY})\right) \leq \varepsilon,,$ the data-processing inequality gives

$$D_{\mathrm{KL}}\left(B_{J\#}\widehat{\mathbb{P}}_{\tilde{X},Y,M} \,\big\|\, B_{J\#}\mathrm{H}(\mathbb{P}_{XY}, \widehat{\mathbb{P}}_{M|XY})\right) \leq \varepsilon,$$

where $B_{J\#}$ denotes the pushforward induced by applying $B_J$ to the $Y$-coordinate. Given $B_J$, define the bin mass $b(x,j) := P_{XY}\big(X = x, \, Y \in \mathcal{Y}_j\big)$. We construct $\widetilde{P}_{XY}$ by redistributing each mass $b(x,j)$ within the cell $\{x\} \times \mathcal{Y}_j$: if the cell contains observed samples, namely if $\mathcal{I}_{xj} := \{i : X_i = x, \, Y_i \in \mathcal{Y}_j\} \neq \varnothing$, then $b(x,j)$ is assigned uniformly to the observed sample support in that cell; otherwise, $b(x,j)$ is spread uniformly over the whole bin $\mathcal{Y}_j$. Equivalently, for any measurable $A \subseteq \mathcal{Y}$,

$$\widetilde{P}_{XY}(X = x, Y \in A) = \sum_{j=1}^{J} b(x,j) \left[ \mathbf{1}_{\{\mathcal{I}_{xj} \neq \varnothing\}} \frac{1}{|\mathcal{I}_{xj}|} \sum_{i \in \mathcal{I}_{xj}} \mathbf{1}_{\{Y_i \in A\}} + \mathbf{1}_{\{\mathcal{I}_{xj} = \varnothing\}} \frac{\lambda(A \cap \mathcal{Y}_j)}{\lambda(\mathcal{Y}_j)} \right],$$

where $\lambda$ denotes Lebesgue measure.

We also define the conditional distribution $\widetilde{P}_{M|X_o,Y}$ in a binwise manner. For every observed pair $(x_o, \mathcal{Y}_j)$, i.e. whenever $\{i : X_{o,i} = x_o, \, Y_i \in \mathcal{Y}_j\} \neq \varnothing$, and for any $y \in \mathcal{Y}_j$, set

$$\widetilde{P}_{M|X_o,Y}(M = m \mid X_o = x_o, Y = y) := \frac{\sum_i \mathbf{1}\{M_i = m, \, X_{o,i} = x_o, \, Y_i \in \mathcal{Y}_j\}}{\sum_i \mathbf{1}\{X_{o,i} = x_o, \, Y_i \in \mathcal{Y}_j\}}, \qquad m \in \mathcal{M}.$$

For any cell $(x_o, \mathcal{Y}_j)$ that is not observed in the sample, namely $\mathcal{I}_{x_o j} = \varnothing$, we choose an arbitrary distribution on $\mathcal{M}$. We note that $\widetilde{\mathbb{P}}_{M|XY} \in \mathcal{Q}_{MAR}$

It follows that

$$D_{\mathrm{KL}}\left(\widehat{\mathbb{P}}_{\tilde{X},Y,M} \,\big\|\, \mathrm{H}(\widetilde{\mathbb{P}}_{XY}, \widehat{\mathbb{P}}_{M|XY})\right) = D_{\mathrm{KL}}\left(B_{J\#}\widehat{\mathbb{P}}_{\tilde{X},Y,M} \,\big\|\, B_{J\#}\mathrm{H}(\widetilde{\mathbb{P}}_{XY}, \widetilde{\mathbb{P}}_{M|XY})\right)$$

$$= D_{\mathrm{KL}}\left(B_{J\#}\widehat{\mathbb{P}}_{\tilde{X},Y,M} \,\big\|\, B_{J\#}\mathrm{H}(\mathbb{P}_{XY}, \widehat{\mathbb{P}}_{M|XY})\right) \leq \varepsilon,$$

The first equality holds with probability one: since $Y$ is continuous, the observed $Y_i$'s have no ties almost surely, and our construction matches the empirical conditional law within each fiber of $(\widetilde{X}, Y, M) \mapsto (\widetilde{X}, B_J(Y), M)$. Thus no extra KL divergence is created inside the bins, so the KL is determined entirely by the binned pushforward laws. The replacement of $\widehat{\mathbb{P}}_{M|XY}$ by $\widetilde{\mathbb{P}}_{M|XY}$ after pushforward follows from the binwise construction, since both induce the same missingness mechanism once $Y$ is replaced by $B_J(Y)$. The second equality follows from the construction of $\widetilde{\mathbb{P}}_{XY}$ and $\widetilde{\mathbb{P}}_{M|XY}$: the former preserves the bin masses of $\mathbb{P}_{XY}$, while the latter depends on $Y$ only through its bin $B_J(Y)$ and coincides with the empirical binwise missingness mechanism. Therefore, after applying $B_J$, the two reference distributions induce the same binned law. The final inequality is then exactly the KL bound for the binned construction.

Next, recall $h_J := \max_{1 \leq j \leq J} \mathrm{diam}(\mathcal{Y}_j)$. We claim that

$$d_{\mathrm{LP}}(\widetilde{\mathbb{P}}_{XY}, \mathbb{P}_{XY}) \leq h_J.$$

Indeed, by construction, $\widetilde{\mathbb{P}}_{XY}$ preserves the same mass as $\mathbb{P}_{XY}$ on every cell $\{x\} \times \mathcal{Y}_j$. Hence we may couple $(X,Y) \sim \mathbb{P}_{XY}$ and $(\widetilde{X}, \widetilde{Y}) \sim \widetilde{\mathbb{P}}_{XY}$ so that $\widetilde{X} = X$ and, whenever $Y \in \mathcal{Y}_j$, also $\widetilde{Y} \in \mathcal{Y}_j$. Therefore, under this coupling,

$$d\big((X,Y), (\widetilde{X}, \widetilde{Y})\big) = d_{\mathcal{Y}}(Y, \widetilde{Y}) \leq \mathrm{diam}(\mathcal{Y}_j) \leq h_J \qquad \text{almost surely.}$$

By the coupling characterization of the Lévy–Prokhorov distance, it follows that $d_{\mathrm{LP}}(\widetilde{\mathbb{P}}_{XY}, \mathbb{P}_{XY}) \leq h_J$. It follows that

$$\mathrm{LP}(\widetilde{\mathbb{P}}_{XY}, \mathbb{P}_{XY}^*) \leq \delta + h_J$$

Finally, since $\mathrm{LP}(\tilde{\mathbb{P}}_{XY}, \mathbb{P}^*_{XY}) \le \delta + h_J$ and $\widehat{\mathbb{P}}_{M|XY} \in \mathcal{Q}_{\mathrm{MAR}}$, the definition of the Bin-projected KL ambiguity set implies that there exists $q \in \mathcal{Q}_{\mathrm{MAR}}$ such that

$$D_{\mathrm{BinPKL}}\left(\widehat{\mathbb{P}}^{\mathrm{obs}}_n \, ; \, \mathrm{H}(\mathbb{P}^*_{XY}, q), \delta + h_J; J\right) \le \varepsilon.$$

Consequently, $\mathbb{P}^*_{XY} \in \mathcal{P}_{\mathrm{BinPKL}}(\varepsilon, \delta + h_J; J)$, which establishes the claimed event inclusion (includsion). $\qquad\square$

Combine together current results we have

$$\Pr\left[\mathbb{P}^*_{Y|\bar{\mathbf{x}}} \notin \mathcal{K}(\gamma_n, \delta_n)\right] \le \Pr\left[\mathbb{P}^*_{XY} \notin \mathcal{P}_{\mathrm{BinPKL}}(\varepsilon_n, \frac{\epsilon_1 \delta_n}{2})\right] \le \Pr\left[D_{\mathrm{RKL}}\left(\widehat{\mathbb{P}}_{XY}, B_{\mathrm{LP}}(\mathbb{P}^*_{XY}, \frac{\epsilon_1 \delta_n}{2} - h_J)\right) > \varepsilon_n\right]$$

It therefore suffices to establish that the event $\left\{D_{\mathrm{RKL}}\left(\widehat{\mathbb{P}}_{XY}, B_{\mathrm{LP}}(\mathbb{P}^*_{XY}, \frac{\epsilon_1 \delta_n}{2} - h_J)\right) > \varepsilon_n\right\}$ holds with probability at most $\eta_n$, for the sequence $(\varepsilon_n, \delta_n)$ calibrated according to the confidence level $\eta_n$. By Proposition 3.14, we have

$$\Pr\left[D_{\mathrm{RKL}}(\widehat{\mathbb{P}}_{XY}, B_{\mathrm{LP}}(\mathbb{P}^*_{XY}, \delta)) > \varepsilon\right] \le \left(\frac{8}{\delta}\right)^{|\mathcal{X}|\left(1 + \frac{\mathrm{diam}(\mathcal{Y})}{\delta}\right)^{d_y}} \cdot \exp\left(-\frac{n\delta^2}{2}\left(e^{-\varepsilon} + e^{-2\delta^2}\right)^{-1}\right).$$

To ensure that this upper bound is less than or equal to $\eta_n = \exp\left(-4CL^2 \left(\frac{n}{\log n}\right)^{\frac{d_y}{d_y+2}} \cdot \log\log n\right)$, where $\kappa = 1 + \mathrm{diam}(\mathcal{Y}), L = \left(\frac{4|\mathcal{X}|\kappa^{d_y}}{d_y+2}\right)^{\frac{1}{d_y+2}}$. Substitute the definition of $\delta_n$ and $\varepsilon_n$ and write $\delta'_n = \frac{\epsilon_1 \delta_n}{2} - h_J = L\left(\frac{\log n}{n}\right)^{\frac{1}{d_y+2}}$

We take **log** of RHS :

$$\le |\mathcal{X}|\kappa^{d_y}(\delta'_n)^{-d_y} \log\left(\frac{8}{\delta'_n}\right) - \frac{n{\delta'_n}^2}{2}\left(e^{-\varepsilon_n} + e^{2{\delta'_n}^2}\right)^{-1}$$

$$\le |\mathcal{X}|\kappa^{d_y}(\delta'_n)^{-d_y} \log\left(\frac{8}{\delta'_n}\right) - \frac{n{\delta'_n}^2}{4} \qquad \text{(since } \left(e^{-\varepsilon_n} + e^{2{\delta'_n}^2}\right)^{-1} \le 2\text{)}$$

$$= |\mathcal{X}|\kappa^{d_y}\left(\frac{4|\mathcal{X}|\kappa^{d_y}\log n}{(d_y+2)n}\right)^{-\frac{d_y}{d_y+2}} \cdot \log\left(\frac{8}{\left(\frac{4|\mathcal{X}|\kappa^{d_y}\log n}{(d_y+2)n}\right)^{\frac{1}{d_y+2}}}\right) - \frac{n}{4}\left(\frac{4|\mathcal{X}|\kappa^{d_y}\log n}{(d_y+2)n}\right)^{\frac{2}{d_y+2}}$$

$$= |\mathcal{X}|\kappa^{d_y}\left(\frac{(d_y+2)n}{4|\mathcal{X}|\kappa^{d_y}\log n}\right)^{\frac{d_y}{d_y+2}} \cdot \left[\log 8 + \frac{1}{d_y+2}\left(\log n - \log\log n + \log(d_y+2) - \log(4|\mathcal{X}|\kappa^{d_y})\right) - \frac{\log n}{d_y+2}\right]$$

$$= |\mathcal{X}|\kappa^{d_y}\left(\frac{(d_y+2)n}{4|\mathcal{X}|\kappa^{d_y}\log n}\right)^{\frac{d_y}{d_y+2}} \cdot \left[\log 8 + \frac{1}{d_y+2}\left(-\log\log n + \log(d_y+2) - \log(4|\mathcal{X}|\kappa^{d_y})\right)\right]$$

$$\left(\lesssim -4\left(\frac{4|\mathcal{X}|\kappa^{d_y}}{d_y+2}\right)^{\frac{2}{d_y+2}}\left(\frac{n}{\log n}\right)^{\frac{d_y}{d_y+2}} \cdot \log\log n, \quad \text{(dominant term)}\right)$$

$$\le -4CL^2 \left(\frac{n}{\log n}\right)^{\frac{d_y}{d_y+2}} \cdot \log\log n,$$

where the constant $L = \left(\frac{4|\mathcal{X}|\kappa^{d_y}}{d_y+2}\right)^{\frac{1}{d_y+2}}$ corresponds exactly to the one used in the definition of $\eta_n$.

Hence, combine together we have

$$\Pr\left[\mathbb{P}^*_{Y|\bar{\mathbf{x}}} \notin \mathcal{K}(\gamma_n, \delta_n)\right] \le \Pr\left[\mathbb{P}^*_{XY} \notin \mathcal{P}_{\mathrm{BinPKL}}(\varepsilon_n, \frac{\epsilon_0 \delta_n}{2})\right] \le \Pr\left[\mathbb{P}^*_{XY} \notin \mathcal{P}_{\mathrm{RKL}}(\varepsilon_n, \frac{\epsilon_0 \delta_n}{2} - h_J)\right] \le \eta_n,$$

completing the proof. $\qquad\square$

## C.11. Proof of Theorem 3.16

We first prove the following lemma:

**Lemma C.7** (Conditional LP bound from joint LP distance). *Let $\mathbb{P}_{XY}$ and $\mathbb{P}_{XY}^*$ be probability measures on the Polish space $\mathcal{X} \times \mathcal{Y}$. Equip $\mathcal{X} \times \mathcal{Y}$ with the product metric*

$$d((x, y), (x', y')) = \mathbf{1}\{x \neq x'\} + d_{\mathcal{Y}}(y, y')$$

*where $\bar{\mathbf{x}}$ is an atom of $\mathcal{X}$.*

*Suppose*

$$d_{\mathrm{LP}}\big(\mathbb{P}_{XY}, \mathbb{P}_{XY}^*\big) \leq \eta, \quad and \quad \mathbb{P}_X^*(\bar{\mathbf{x}}) = \mathbb{P}_{XY}^*(\{\bar{\mathbf{x}}\} \times \mathcal{Y}) \geq \epsilon_0 > 0.$$

*Assuming $\eta < \frac{1}{2}\epsilon_0$ (otherwise the bound below is trivial), the conditional laws satisfy*

$$d_{\mathrm{LP}}\big(\mathbb{P}(Y \mid \bar{\mathbf{x}}), \mathbb{P}^*(Y \mid \bar{\mathbf{x}})\big) \leq \frac{2\eta}{\epsilon_0}.$$

*Proof.* Proof of the lemma. The proof is similar to what we have done in proof of Proposition 3.11.

Let $\mathbb{P}^{\bar{\mathbf{x}}} = \mathbb{P}(Y \mid \bar{\mathbf{x}})$ and $\mathbb{P}^{*\bar{\mathbf{x}}} = \mathbb{P}^*(Y \mid \bar{\mathbf{x}})$ denote the conditional measures, and let $p_{\bar{\mathbf{x}}} = \mathbb{P}_X(\bar{\mathbf{x}})$ and $p_{\bar{\mathbf{x}}}^* = \mathbb{P}_X^*(\bar{\mathbf{x}})$. From the joint LP bound $d_{\mathrm{LP}}(\mathbb{P}_{XY}, \mathbb{P}_{XY}^*) \leq \eta$, the marginals satisfy the LP bound, implying $p_{\bar{\mathbf{x}}} \geq p_{\bar{\mathbf{x}}}^* - \eta$. Since $p_{\bar{\mathbf{x}}}^* \geq \epsilon_0$ and we assumed $\eta < \epsilon_0$, we have $p_{\bar{\mathbf{x}}} \geq \epsilon_0 - \eta > 0$. This strictly guarantees that the conditional distribution $\mathbb{P}^{\bar{\mathbf{x}}}$ is well-defined.

We now use the set-based definition of the LP metric. For any Borel set $\tilde{A} \subseteq \mathcal{Y}$, define the joint set $A = \{\bar{\mathbf{x}}\} \times \tilde{A}$. Because $\eta < 1$ (implied by $\eta < \epsilon_0/2 \leq 1/2 < 1$, the $\eta$-neighborhood of $A$ in the product metric $d((x, y), (x', y')) = \mathbf{1}\{x \neq x'\} + \|y - y'\|$ is exactly $A^\eta = \{\bar{\mathbf{x}}\} \times \tilde{A}^\eta$.

Applying the joint LP condition $\mathbb{P}_{XY}^*(A) \leq \mathbb{P}_{XY}(A^\eta) + \eta$ yields:

$$p_{\bar{\mathbf{x}}}^* \mathbb{P}^{*\bar{\mathbf{x}}}(\tilde{A}) \leq p_{\bar{\mathbf{x}}} \mathbb{P}^{\bar{\mathbf{x}}}(\tilde{A}^\eta) + \eta.$$

Using the upper bound $p_{\bar{\mathbf{x}}} \leq p_{\bar{\mathbf{x}}}^* + \eta$ and the fact that $\mathbb{P}^{\bar{\mathbf{x}}}(\tilde{A}^\eta) \leq 1$, we have:

$$p_{\bar{\mathbf{x}}}^* \mathbb{P}^{*\bar{\mathbf{x}}}(\tilde{A}) \leq (p_{\bar{\mathbf{x}}}^* + \eta)\mathbb{P}^{\bar{\mathbf{x}}}(\tilde{A}^\eta) + \eta \leq p_{\bar{\mathbf{x}}}^* \mathbb{P}^{\bar{\mathbf{x}}}(\tilde{A}^\eta) + 2\eta.$$

Dividing by $p_{\bar{\mathbf{x}}}^*$ and using $p_{\bar{\mathbf{x}}}^* \geq \epsilon_0 > 0$, we obtain the first one-sided bound:

$$\mathbb{P}^{*\bar{\mathbf{x}}}(\tilde{A}) \leq \mathbb{P}^{\bar{\mathbf{x}}}(\tilde{A}^\eta) + \frac{2\eta}{p_{\bar{\mathbf{x}}}^*} \leq \mathbb{P}^{\bar{\mathbf{x}}}(\tilde{A}^\eta) + \frac{2\eta}{\epsilon_0}.$$

To establish the reverse bound, we apply the symmetric joint LP condition $\mathbb{P}_{XY}(A) \leq \mathbb{P}_{XY}^*(A^\eta) + \eta$, which yields $p_{\bar{\mathbf{x}}} \mathbb{P}^{\bar{\mathbf{x}}}(\tilde{A}) \leq p_{\bar{\mathbf{x}}}^* \mathbb{P}^{*\bar{\mathbf{x}}}(\tilde{A}^\eta) + \eta$. Substituting $p_{\bar{\mathbf{x}}} \geq p_{\bar{\mathbf{x}}}^* - \eta$:

$$(p_{\bar{\mathbf{x}}}^* - \eta)\mathbb{P}^{\bar{\mathbf{x}}}(\tilde{A}) \leq p_{\bar{\mathbf{x}}}^* \mathbb{P}^{*\bar{\mathbf{x}}}(\tilde{A}^\eta) + \eta \implies p_{\bar{\mathbf{x}}}^* \mathbb{P}^{\bar{\mathbf{x}}}(\tilde{A}) \leq p_{\bar{\mathbf{x}}}^* \mathbb{P}^{*\bar{\mathbf{x}}}(\tilde{A}^\eta) + \eta + \eta \mathbb{P}^{\bar{\mathbf{x}}}(\tilde{A}).$$

Since $\eta \mathbb{P}^{\bar{\mathbf{x}}}(\tilde{A}) \leq \eta$, dividing by $p_{\bar{\mathbf{x}}}^*$ yields:

$$\mathbb{P}^{\bar{\mathbf{x}}}(\tilde{A}) \leq \mathbb{P}^{*\bar{\mathbf{x}}}(\tilde{A}^\eta) + \frac{2\eta}{p_{\bar{\mathbf{x}}}^*} \leq \mathbb{P}^{*\bar{\mathbf{x}}}(\tilde{A}^\eta) + \frac{2\eta}{\epsilon_0}.$$

To satisfy the strict definition of the LP distance, the spatial expansion radius and additive penalty must match. Since $\epsilon_0 \leq 1$, we have $\eta \leq \frac{2\eta}{\epsilon_0}$, meaning $\tilde{A}^\eta \subseteq \tilde{A}^{\frac{2\eta}{\epsilon_0}}$. By monotonicity of probability measures, $\mathbb{P}(\tilde{A}^\eta) \leq \mathbb{P}(\tilde{A}^{\frac{2\eta}{\epsilon_0}})$. Substituting this into both bounds gives:

$$\mathbb{P}^{*\bar{\mathbf{x}}}(\tilde{A}) \leq \mathbb{P}^{\bar{\mathbf{x}}}\left(\tilde{A}^{\frac{2\eta}{\epsilon_0}}\right) + \frac{2\eta}{\epsilon_0} \quad and \quad \mathbb{P}^{\bar{\mathbf{x}}}(\tilde{A}) \leq \mathbb{P}^{*\bar{\mathbf{x}}}\left(\tilde{A}^{\frac{2\eta}{\epsilon_0}}\right) + \frac{2\eta}{\epsilon_0}.$$

As this holds for all Borel sets $\tilde{A} \subseteq \mathcal{Y}$, we conclude $d_{\mathrm{LP}}\big(\mathbb{P}^{\bar{\mathbf{x}}}, \mathbb{P}^{*\bar{\mathbf{x}}}\big) \leq \frac{2\eta}{\epsilon_0}$.

$\square$

**Lemma C.8** (Likelihood-ratio separation under Hellinger bracketing). *Let $O_1, \ldots, O_n$ be i.i.d. from a density $f_0$ with respect to a dominating measure $\mu$. Let $\mathcal{F}_n$ be a class of densities with respect to $\mu$. For $r > 0$, define*

$$\mathcal{F}_{n,r} := \{f \in \mathcal{F}_n : H(f, f_0) \geq r\},$$

*where*

$$H^2(f, g) := \int (\sqrt{f} - \sqrt{g})^2 \, d\mu.$$

*Assume that, for every fixed $r > 0$, there exists a sequence $\eta_{n,r} \in (0, r^2/8)$ such that*

$$\log N_{[]}(\eta_{n,r}, \mathcal{F}_{n,r}, H) = o(n).$$

*Then there exists a constant $c_r, C_r > 0$, depending only on $r$, such that*

$$P_{f_0}\left(\sup_{f \in \mathcal{F}_{n,r}} \frac{1}{n} \sum_{i=1}^{n} \log \frac{f(O_i)}{f_0(O_i)} \geq -c_r\right) \leq \exp\left(-\frac{nC_r}{2}\right) \to 0.$$

*Proof.* Proof of the lemma.

Fix $r > 0$. If $\mathcal{F}_{n,r} = \varnothing$, the conclusion is trivial. Hence assume $\mathcal{F}_{n,r} \neq \varnothing$. Since $H(f, f_0) \leq \sqrt{2}$ under our convention, it is enough to consider $0 < r \leq \sqrt{2}$.

Let $\eta_{n,r} < \frac{r^2}{8}$ be such that $\log N_{[]}(\eta_{n,r}, \mathcal{F}_{n,r}, H) = o(n)$. For notational simplicity, in the following we write $N_{n,r} := N_{[]}(\eta_{n,r}, \mathcal{F}_{n,r}, H)$. Let $[l_k, u_k], k = 1, \ldots, N_{n,r}$ be Hellinger brackets covering $\mathcal{F}_{n,r}$. We discard any bracket that does not contain an element of $\mathcal{F}_{n,r}$. Thus, for each remaining bracket, there exists $f_k \in \mathcal{F}_{n,r}$ such that $l_k \leq f_k \leq u_k$, and $\left(\int (\sqrt{u_k} - \sqrt{l_k})^2 \, d\mu\right)^{1/2} \leq \eta_{n,r}$

We first show that each upper bracket $u_k$ has small affinity with $f_0$. Fix $k$. Since $H(f_k, f_0) \geq r$,

$$\int \sqrt{f_k f_0} \, d\mu = 1 - \frac{1}{2} H^2(f_k, f_0) \leq 1 - \frac{r^2}{2}.$$

Moreover,

$$\int \sqrt{u_k f_0} \, d\mu = \int \sqrt{f_k f_0} \, d\mu + \int (\sqrt{u_k} - \sqrt{f_k}) \sqrt{f_0} \, d\mu$$

$$\leq 1 - \frac{r^2}{2} + \left(\int (\sqrt{u_k} - \sqrt{f_k})^2 \, d\mu\right)^{1/2}$$

where we also use Cauchy–Schwarz inequality and $\int f_0 \, d\mu = 1$.

Because $l_k \leq f_k \leq u_k$, we have

$$0 \leq \sqrt{u_k} - \sqrt{f_k} \leq \sqrt{u_k} - \sqrt{l_k}$$

Therefore,

$$\int (\sqrt{u_k} - \sqrt{f_k})^2 \, d\mu \leq \int (\sqrt{u_k} - \sqrt{l_k})^2 \, d\mu \leq \eta_{n,r}^2$$

It follows that

$$\int \sqrt{u_k f_0} \, d\mu \leq 1 - \frac{r^2}{2} + \eta_{n,r} \leq 1 - \frac{3r^2}{8},$$

where the last inequality uses $\eta_{n,r} < r^2/8$

Set $c_r := \frac{3r^2}{8}$, since $1 - c_r \leq e^{-c_r}$, we obtain $\int \sqrt{u_k f_0} \, d\mu \leq e^{-c_r}$.

Now define the event

$$E_n := \left\{\sup_{f \in \mathcal{F}_{n,r}} \frac{1}{n} \sum_{i=1}^{n} \log \frac{f(O_i)}{f_0(O_i)} \geq -c_r\right\}.$$

If $E_n$ occurs, then there exists $f \in \mathcal{F}_{n,r}$ such that $\prod_{i=1}^{n} \frac{f(O_i)}{f_0(O_i)} \geq e^{-nc_r}$.

Choose a bracket $[l_k, u_k]$ containing $f$. Since $f \leq u_k$, we have $\prod_{i=1}^{n} \frac{u_k(O_i)}{f_0(O_i)} \geq \prod_{i=1}^{n} \frac{f(O_i)}{f_0(O_i)} \geq e^{-nc_r}$ Therefore,

$$E_n \subseteq \bigcup_{k=1}^{N_{n,r}} \left\{ \prod_{i=1}^{n} \frac{u_k(O_i)}{f_0(O_i)} \geq e^{-nc_r} \right\}$$

For each fixed $k$, apply Markov's inequality to the square-root likelihood ratio:

$$P_{f_0} \left( \prod_{i=1}^{n} \frac{u_k(O_i)}{f_0(O_i)} \geq e^{-nc_r} \right) = P_{f_0} \left( \prod_{i=1}^{n} \sqrt{\frac{u_k(O_i)}{f_0(O_i)}} \geq e^{-nc_r/2} \right)$$

$$\leq e^{nc_r/2} E_{f_0} \left[ \prod_{i=1}^{n} \sqrt{\frac{u_k(O_i)}{f_0(O_i)}} \right]$$

Since $O_1, \ldots, O_n$ are i.i.d. under $f_0$,

$$E_{f_0} \left[ \prod_{i=1}^{n} \sqrt{\frac{u_k(O_i)}{f_0(O_i)}} \right] = \left( \int \sqrt{u_k f_0} \, d\mu \right)^n \leq e^{-nc_r}$$

Thus

$$P_{f_0} \left( \prod_{i=1}^{n} \frac{u_k(O_i)}{f_0(O_i)} \geq e^{-nc_r} \right) \leq e^{-nc_r/2}$$

Taking the union bound over all brackets gives

$$P_{f_0}(E_n) \leq N_{n,r} e^{-nc_r/2} = \exp\left( \log N_{n,r} - \frac{nc_r}{2} \right)$$

Since $\log N_{n,r} = o(n)$ and $c_r > 0$ is fixed for the given $r$, it follows that $P_{f_0}(E_n) \to 0$. This proves the claim.

$\square$

**Next we prove the main theorem**:

*Proof.* Proof of Theorem 3.16. For simplicity of notation, let the calibrated radius be $\gamma_n := \gamma_n(\eta_n)$ and $\delta_n := \delta_n(\eta_n)$, denote by $\mathbb{P}^*_{\mathrm{obs}}$ the true observable law and write $\tilde{p}^* := \mathbb{P}^*_{\mathrm{obs}}$.

**Step A.** Since $\hat{z}_n \in \mathcal{Z}$, we have $\Phi^* \leq \mathbb{E}_{\mathbb{P}^*}[c(\hat{z}_n, Y) \mid X = \bar{x}]$ Moreover, Theorem 3.15 with $z = \hat{z}_n$ yields

$$\Pr\left[ \mathbb{E}_{\mathbb{P}^*}[c(\hat{z}_n, Y) \mid X = \bar{x}] \leq \Phi_n \right] \geq 1 - \eta_n.$$

Hence

$$\Pr\left[ \Phi^* \leq \mathbb{E}_{\mathbb{P}^*}[c(\hat{z}_n, Y) \mid X = \bar{x}] \leq \Phi_n \right] \geq 1 - \eta_n \tag{A1}$$

Because $\sum_{n=1}^{\infty} \eta_n < \infty$, the Borel–Cantelli lemma implies that the event in (A1) occurs for all sufficiently large $n$ almost surely; that is,

$$\mathbb{P}^\infty\left[ \Phi^* \leq \mathbb{E}_{\mathbb{P}^*}[c(\hat{z}_n, Y) \mid X = \bar{x}] \leq \Phi_n \text{ for all sufficiently large } n \right] = 1 \tag{A}$$

It remains to prove the matching upper bound $\limsup_{n\to\infty} \Phi_n \leq \Phi^*$ and then to establish convergence of the robust solutions $\hat{z}_n$.

By Proposition 3.11, with $\delta = \delta_n/2$,

$$\mathcal{K}(\gamma_n, \delta_n; J) \subseteq \left\{ \mathbb{P}(\cdot \mid \bar{x}) : \mathbb{P}_{XY} \in \mathcal{P}_{\mathrm{BinPKL}}\left( \epsilon_n, \frac{3}{2}\delta_n; J \right) \right\}.$$

Thus, by the containment above, we have

$$\Phi_n \;\leq\; \Phi'_n \;:=\; \inf_{z\in\mathcal{Z}} \; \sup_{\mathbb{P}(Y|\bar{\mathbf{x}}):\, \mathbb{P}_{XY}\in\mathcal{P}_{\mathrm{BinPKL}}(\epsilon_n,\frac{3}{2}\delta_n)} \mathbb{E}_{\mathbb{P}}[c(z,Y)].$$

**Step B. LP convergence.** Fix any $\mathbb{P}_{XY} \in \mathcal{P}_{\mathrm{BinPKL}}(\epsilon_n,\frac{3}{2}\delta_n)$. By definition, there exists a reference law $\mathbb{Q}_{XY} \in \mathcal{B}_{J(n)}$ and a MAR kernel $q$ such that $d_{\mathrm{LP}}(\mathbb{P}_{XY},\mathbb{Q}_{XY}) \leq \frac{3}{2}\delta_n$ and the BinPKL constraint holds. We first control the LP distance between this reference law and the true law, and then use the conditional lifting lemma C.7 to obtain a bound for the conditional laws (i.e., $\mathbb{P}(Y \mid \bar{\mathbf{x}})$ and the true conditional law $\mathbb{P}^*(Y \mid \bar{\mathbf{x}})$).

Let $(\mathbb{Q}_{XY}, q_{M|X_o,Y})$ be any feasible pair such that

$$\mathbb{Q}_{XY} \in \mathcal{B}_J, \mathbb{Q}_{XY} \in B_{\mathrm{LP},\frac{3\delta_n}{2}}(\mathbb{P}_{XY}), \quad D_{\mathrm{KL}}\left(\widehat{\mathbb{P}}(\tilde{X},Y,M) \,\|\, \mathrm{H}(\mathbb{Q}_{XY}, q_{M|X_o,Y})\right) \leq \epsilon_{\min,n} + \varepsilon_n$$

We know that there exists an empirical construction $\hat{q}_{M|X_o,Y} \in \mathcal{Q}_{\mathrm{MAR}}$. This is a virtual object introduced only to facilitate the proof; This substitution does not increase the corresponding KL divergence, as we previously showed

$$D_{\mathrm{KL}}\left(\widehat{\mathbb{P}}(\tilde{X},Y,M) \,\|\, \mathrm{H}(\mathbb{Q}_{XY}, \hat{q}_{M|X_o,Y})\right) \leq \epsilon_{\min,n} + \varepsilon_n$$

Let $I_{n,j} := \{i \in [n] : y_i \in Y_{n,j}\}$, $N_{n,j} := |I_{n,j}|$. By properly choosing $J(n)$ s.t. $\mathbb{P}\left(\min_{1\leq j\leq J(n)} N_{n,j} \geq 1\right) \to 1$. On the event $\min_j N_{n,j} \geq 1$, For observation $i$, let $A_i := A(\tilde{x}_{m,i}, m_i)$ be the set of missing-covariate completions compatible with the observed data, and let $j(i)$ be the index such that $y_i \in Y_{n,j(i)}$. For each mass $b = (b_{xj})_{x\in\mathcal{X},\, j\in[J(n)]} \in \Delta(\mathcal{X} \times [J(n)])$, we write

$$\ell_n^A(b) := \frac{1}{n}\sum_{i=1}^{n} \log\left(\sum_{x_m\in A_i} \frac{b_{(x_{o,i},x_m),j(i)}}{N_{n,j(i)}}\right)$$

For a deterministic sequence $\varepsilon_n \downarrow 0$, define the near-maximal likelihood set

$$\mathcal{M}_n(\varepsilon_n) := \left\{ b \in \Delta(\mathcal{X} \times [J(n)]) : \ell_n^A(b) \geq \sup_{\bar{b}\in\Delta(\mathcal{X}\times[J(n)])} \ell_n^A(\bar{b}) - \varepsilon_n \right\}.$$

It is equivalent to represent $\mathbb{Q}_{XY}$ (when $\epsilon_{\min,n} + \varepsilon_n < \infty$) as

$$b \in \mathcal{M}_n(\varepsilon_n), \mathbb{Q}^A = \sum_{x\in\mathcal{X}}\sum_{j=1}^{J(n)}\sum_{i\in I_{n,j}} \frac{b_{xj}}{N_{n,j}}\delta_{(x,y_i)}$$

Define the histogram law $Q_b^H$ (used for analysis) by

$$Q_b^H(x,y)(\{x\}\times A) = \sum_{j=1}^{J(n)} b_{xj} \frac{\lambda(A\cap Y_{n,j})}{\lambda(Y_{n,j})}$$

where $\lambda$ denotes Lebesgue measure on $\mathcal{Y}$.

**We write the proof in several steps.**

**Step B1: Reference and lifted histogram laws are close in $d_{LP}$.**

Fix $b \in \Delta(\mathcal{X} \times [J(n)])$. We construct a coupling between $Q_b^H$ and $Q_b^A$.

Draw $(X, Y^H) \sim Q_b^H$. Conditional on the event $\{X = x, Y^H \in Y_{n,j}\}$, draw $Y^A$ uniformly from the sample atoms in the same bin: $\{y_i : i \in I_{n,j}\}$. Then $(X, Y^A)$ has law $Q_b^A$. Moreover, $Y^H$ and $Y^A$ both belong to $Y_{n,j}$, so

$$\|Y^H - Y^A\| \leq \mathrm{diam}(Y_{n,j}) \leq h_J.$$

Since $X$ is unchanged, under the product metric $d((x,y),(x',y')) = \mathbf{1}\{x \neq x'\} + \|y - y'\|$, we have $d((X,Y^H),(X,Y^A)) \leq h_J$, a.s. By the coupling characterization of the Levy–Prokhorov metric, $d_{LP}(Q_b^A, Q_b^H) \leq h_J$ This bound holds uniformly for $b$. Hence, on the event $\min_j N_{n,j} \geq 1$,

$$\sup_{b \in \Delta(\mathcal{X} \times [J(n)])} d_{LP}(Q_b^A, Q_b^H) \leq h_J \to 0$$

**Step B2: Reference and histogram likelihoods differ only by a constant.**

Define the histogram empirical log-likelihood

$$\ell_n^H(b) := \frac{1}{n} \sum_{i=1}^n \log \left( \sum_{x_m \in A_i} \frac{b_{(x_{o,i},x_m),j(i)}}{\lambda(Y_{n,j(i)})} \right)$$

For each observation $i$,

$$\sum_{x_m \in A_i} \frac{b_{(x_{o,i},x_m),j(i)}}{N_{n,j(i)}} = \frac{\lambda(Y_{n,j(i)})}{N_{n,j(i)}} \sum_{x_m \in A_i} \frac{b_{(x_{o,i},x_m),j(i)}}{\lambda(Y_{n,j(i)})}$$

Therefore,

$$\ell_n^A(b) = \ell_n^H(b) + \frac{1}{n} \sum_{i=1}^n \log \left( \frac{\lambda(Y_{n,j(i)})}{N_{n,j(i)}} \right)$$

The second term is independent of $b$. Consequently, $b \in \mathcal{M}_n(\varepsilon_n)$ if and only if $\ell_n^H(b) \geq \sup_{\bar{b} \in \Delta(\mathcal{X} \times [J(n)])} \ell_n^H(\bar{b}) - \varepsilon_n$.

**Step B3: A sieve likelihood contraction lemma.**

Let $O_i = (\widetilde{X}_{m,i}, X_{o,i}, M_i, Y_i)$ denote the observed data. For $b \in \Delta(\mathcal{X} \times [J(n)])$, define the observed density

$$f_b(O_i) := \pi^\star(m_i \mid x_{o,i}, y_i) \sum_{x_m \in A_i} \frac{b_{(x_{o,i},x_m),j(i)}}{\lambda(Y_{n,j(i)})},$$

The factor $\pi^\star(m_i \mid x_{o,i}, y_i)$ is introduced only for the analysis. Since it is independent of $b$, maximizing $\ell_n^H(b)$ is equivalent to maximizing the empirical log-likelihood $n^{-1} \sum_i \log f_b(O_i)$, up to an additive term independent of $b$.

Let $f^*$ denote the true observed-data density. We make the following cliam :

**Claim**. *Let* $\mathcal{F}_n := \{f_b : b \in \Delta(\mathcal{X} \times [J(n)])\}$. *Suppose that* $\log N_{[]}(\eta, \mathcal{F}_n, H) \leq C|\mathcal{X}|J(n) \log \left( \frac{C|\mathcal{X}|J(n)}{\eta^2} \right)$ *for all* $\eta \in (0,1)$, *and that* $|\mathcal{X}|J(n) \log n / n \to 0$. *Suppose further that there exists* $b_n^\star \in \Delta(\mathcal{X} \times [J(n)])$ *such that* $D_{\mathrm{KL}}(f^* \| f_{b_n^\star}) \to 0$ *and* $E_{f^*}\left[ \left( \log \frac{f^*}{f_{b_n^\star}} \right)^2 \right] \leq n^{-\alpha} \to 0$ *for some* $\alpha > 0$. *Then, for every* $\varepsilon_n \to 0$, $\sup_{b \in \mathcal{M}_n(\varepsilon_n)} \widetilde{H}(P^\star, Q_b^H) \to 0$ *a.s.. Here* $\widetilde{H}(P^\star, Q_b^H) := H(f^*, f_b)$ *is the observed-data Hellinger distance.*

**Proof of the claim** : For any fixed $r > 0$, define the bad set

$$\mathcal{F}_{n,r} := \{f_b : H(f_b, f^*) \geq r\}$$

Since

$$\mathcal{F}_{n,r} \subseteq \{f_b : b \in \Delta(\mathcal{X} \times [J(n)])\} := \mathcal{F}_n,$$

the assumed bracketing bound implies that, for every fixed $\eta \in (0,1)$,

$$\log N_{[]}(\eta, \mathcal{F}_{n,r}, H) \leq C|\mathcal{X}|J(n) \log \left( \frac{C|\mathcal{X}|J(n)}{\eta^2} \right)$$

Choose $\eta_r \in (0,1)$ small enough, for instance $\eta_r < r^2/8$. Since $\frac{|\mathcal{X}|J(n)\log n}{n} \to 0$, we have $|\mathcal{X}|J(n) \leq n$ eventually. Hence

$$\log \left( \frac{C|\mathcal{X}|J(n)}{\eta_r^2} \right) \leq C_r \log n$$

for some constant $C_r < \infty$ depending only on $r$. Therefore,

$$\log N_{[]}(\eta_r, \mathcal{F}_{n,r}, H) \leq C_r |\mathcal{X}| J(n) \log n = o(n).$$

By Lemma C.8, there exists $c_r > 0$ such that

$$P_{f^*}\left(\sup_{f \in \mathcal{F}_{n,r}} \frac{1}{n} \sum_{i=1}^{n} \log \frac{f(O_i)}{f^*(O_i)} \geq -c_r\right) \leq \exp\left(-\frac{nC_r}{2}\right) \to 0.$$

Next, consider the approximating density $f_{b_n^\star}$. Recall we assume that $E_{f^*}\left[\left(\log \frac{f^*}{f_{b_n^\star}}\right)^2\right] \to 0$ and $D_{\mathrm{KL}}(f^*\|f_{b_n^\star}) \to 0$. Let $Z_{n,i} := \log \frac{f_{b_n^\star}(O_i)}{f^*(O_i)}$, then $\mathbb{E}_{f^*}[Z_{n,i}] = -D_{\mathrm{KL}}(f^*\|f_{b_n^\star})$ and

$$\mathrm{Var}_{f^*}(Z_{n,i}) \leq \mathbb{E}_{f^*}\left[\left(\log \frac{f_{b_n^\star}(O_i)}{f^*(O_i)}\right)^2\right] = \mathbb{E}_{f^*}\left[\left(\log \frac{f^*(O_i)}{f_{b_n^\star}(O_i)}\right)^2\right]$$

For all sufficiently large $n$, $D_{\mathrm{KL}}(f^*\|f_{b_n^\star}) \leq \frac{c_r}{4}$. Therefore,

$$-\frac{c_r}{2} - \mathbb{E}_{f^*}[Z_{n,i}] = -\frac{c_r}{2} + D_{\mathrm{KL}}(f^*\|f_{b_n^\star}) \leq -\frac{c_r}{4}$$

Hence, by Chebyshev's inequality,

$$P_{f^*}\left(\frac{1}{n}\sum_{i=1}^{n} \log \frac{f_{b_n^\star}(O_i)}{f^*(O_i)} < -\frac{c_r}{2}\right)$$

$$= P_{f^*}\left(\frac{1}{n}\sum_{i=1}^{n} Z_{n,i} - \mathbb{E}_{f^*}[Z_{n,i}] < -\frac{c_r}{2} - \mathbb{E}_{f^*}[Z_{n,i}]\right)$$

$$\leq P_{f^*}\left(\left|\frac{1}{n}\sum_{i=1}^{n} Z_{n,i} - \mathbb{E}_{f^*}[Z_{n,i}]\right| > \frac{c_r}{4}\right)$$

$$\leq \frac{16}{nc_r^2} \mathrm{Var}_{f^*}(Z_{n,i})$$

$$\leq \frac{16}{nc_r^2} \mathbb{E}_{f^*}\left[\left(\log \frac{f^*(O_i)}{f_{b_n^\star}(O_i)}\right)^2\right] \leq \frac{16}{n^{1+\alpha} c_r^2}$$

Define

$$A_n(r) := \left\{\sup_{f \in \mathcal{F}_{n,r}} \frac{1}{n}\sum_{i=1}^{n} \log \frac{f(O_i)}{f^*(O_i)} \geq -c_r\right\}, B_n(r) := \left\{\frac{1}{n}\sum_{i=1}^{n} \log \frac{f_{b_n^\star}(O_i)}{f^*(O_i)} \geq -\frac{c_r}{2}\right\}.$$

We have obatined that $P_{f^*}(A_n(r)) \to 0$ and $P_{f^*}(B_n(r)^c) \to 0$.

We next show that, on $A_n(r)^c \cap B_n(r)$, no near-maximal likelihood candidate can belong to the bad set . Next, suppose, toward contradiction, that there exists $b \in \mathcal{M}_n(\varepsilon_n)$ such that $H(f_b, f^*) \geq r$, then $f_b \in \mathcal{F}_{n,r}$.

By the preceding equivalence between $\ell_n^H$ and $n^{-1}\sum_i \log f_b(O_i)$, the definition of $\mathcal{M}_n(\varepsilon_n)$ implies

$$\frac{1}{n}\sum_{i=1}^{n} \log f_b(O_i) \geq \sup_{\bar{b} \in \Delta(\mathcal{X} \times [J(n)])} \frac{1}{n}\sum_{i=1}^{n} \log f_{\bar{b}}(O_i) - \varepsilon_n$$

Since $b_n^\star \in \Delta(\mathcal{X} \times [J(n)])$, we have

$$\sup_{\bar{b} \in \Delta(\mathcal{X} \times [J(n)])} \frac{1}{n}\sum_{i=1}^{n} \log f_{\bar{b}}(O_i) \geq \frac{1}{n}\sum_{i=1}^{n} \log f_{b_n^\star}(O_i)$$

Therefore,

$$\frac{1}{n}\sum_{i=1}^{n}\log f_b(O_i) \geq \frac{1}{n}\sum_{i=1}^{n}\log f_{b_n^\star}(O_i) - \varepsilon_n$$

Subtracting $\frac{1}{n}\sum_{i=1}^{n}\log f^*(O_i)$ from both sides gives

$$\frac{1}{n}\sum_{i=1}^{n}\log\frac{f_b(O_i)}{f^*(O_i)} \geq \frac{1}{n}\sum_{i=1}^{n}\log\frac{f_{b_n^\star}(O_i)}{f^*(O_i)} - \varepsilon_n$$

On Event $B_n(r)$, the first term on the right-hand side is at least $-c_r/2$. Since $\varepsilon_n \to 0$, for all sufficiently large $n$, $\varepsilon_n \leq \frac{c_r}{2}$. Hence

$$\frac{1}{n}\sum_{i=1}^{n}\log\frac{f_b(O_i)}{f^*(O_i)} \geq -c_r$$

Since $f_b \in \mathcal{F}_{n,r}$, this implies $\sup_{f \in \mathcal{F}_{n,r}}\frac{1}{n}\sum_{i=1}^{n}\log\frac{f(O_i)}{f^*(O_i)} \geq -c_r$ : That is, the event $A_n(r)$ occurs, contradicting $A_n(r)^c$. Therefore, for all sufficiently large $n$,

$$A_n(r)^c \cap B_n(r) \subseteq \left\{\sup_{b \in \mathcal{M}_n(\varepsilon_n)} H(f_b, f^*) < r\right\}$$

Equivalently,

$$\left\{\sup_{b \in \mathcal{M}_n(\varepsilon_n)} H(f_b, f^*) \geq r\right\} \subseteq A_n(r) \cup B_n(r)^c$$

Thus

$$P_{f^*}\left(\sup_{b \in \mathcal{M}_n(\varepsilon_n)} H(f_b, f^*) \geq r\right) \leq P_{f^*}(A_n(r)) + P_{f^*}(B_n(r)^c) \leq \exp\left(-\frac{nC_r}{2}\right) + \frac{16}{n^{1+\alpha}c_r^2} \to 0$$

Since $\sum_{n=1}^{\infty}\left[\exp\left(-\frac{nC_r}{2}\right) + \frac{16}{n^{1+\alpha}c_r^2}\right] < \infty$, we obtain $\sum_{n=1}^{\infty} P_{f^*}\left(\sup_{b \in \mathcal{M}_n(\varepsilon_n)} H(f_b, f^*) \geq r\right) < \infty$. By the Borel–Cantelli lemma, for each fixed $r > 0$,

$$P_{f^*}\left(\sup_{b \in \mathcal{M}_n(\varepsilon_n)} H(f_b, f^*) \geq r \text{ i.o.}\right) = 0.$$

Equivalently, with $P_{f^*}$-probability one, for every fixed $r > 0$, there exists $N_r(\omega) < \infty$ such that $\sup_{b \in \mathcal{M}_n(\varepsilon_n)} H(f_b, f^*) < r, \forall n \geq N_r(\omega)$. Applying this conclusion to all rational $r > 0$, we obtain $\sup_{b \in \mathcal{M}_n(\varepsilon_n)} H(f_b, f^*) \to 0$ almost surely.

Finally, by definition, $H(f_b, f^*) = \widetilde{H}(P^\star, Q_b^H)$, and therefore $\sup_{b \in \mathcal{M}_n(\varepsilon_n)} \widetilde{H}(P^\star, Q_b^H) \to 0$ almost surely. This proves the claim.

**It remains to verify the conditions for the present histogram class**.

1. First, define the true binned masses $b_{n,xj}^\star := P^\star(X = x, Y \in Y_{n,j})$. The histogram law generated by $b_n^\star$ is denoted $H_n P^\star$. For $y \in Y_{n,j}$, the density of $H_n P^\star$ is (by definition)

   $$p_n^H(x,y) = \frac{1}{\lambda(Y_{n,j})}\int_{Y_{n,j}} p^\star(x,u)\,du, \quad y \in Y_{n,j}$$

   Since $\|u - y\| \leq h_J$ for all $u \in Y_{n,j}$, the Hölder continuity of $\log p^\star$ , (i.e, there exists a $\omega_p$ such that $|\log p^\star(x,y) - \log p^\star(x,y')| \leq \|y - y'\|^\beta$ ) gives that

   $$e^{-\omega_p h_n^\beta} p^\star(x,y) \leq p^\star(x,u) \leq e^{\omega_p h_n^\beta} p^\star(x,y), \qquad u,y \in Y_{n,j}$$

   Averaging the preceding representation over $u \in Y_{n,j}$ gives

$$e^{-\omega_p h_n^\beta} p^\star(x,y) \leq p_n^H(x,y) \leq e^{\omega_p h_n^\beta} p^\star(x,y)$$

Thus $\left| \log \frac{p^\star(x,y)}{p_n^H(x,y)} \right| \leq \omega_p h_J$. Consequently,

$$D_{\mathrm{KL}}(P^\star \| H_n P^\star) = \sum_{x \in \mathcal{X}} \int_{\mathcal{Y}} p^\star(x,y) \log \frac{p^\star(x,y)}{p_n^H(x,y)} \, dy$$

$$\leq \omega_p h_n^\beta \sum_{x \in \mathcal{X}} \int_{\mathcal{Y}} p^\star(x,y) \, dy$$

$$= \omega_p h_n^\beta \to 0$$

Moreover,

$$\int \left( \log \frac{p^\star}{p_n^H} \right)^2 dP^\star \leq (\omega_p h_n^\beta)^2 \leq C n^{-2\alpha\beta} \to 0$$

Because the observed law is obtained from the full law by the same MAR kernel, the same bounds hold for the observed densities:

$$D_{\mathrm{KL}}(f^* \| f_{q_n^\star}) \leq \omega_p h_n^\beta \to 0, \qquad E_{f^*}\left[ \left( \log \frac{f^*}{f_{q_n^\star}} \right)^2 \right] \leq C n^{-2\alpha\beta} \to 0.$$

Indeed, the likelihood ratio over the observed data is an average of full-data likelihood ratios over compatible completions, and the pointwise bound above is preserved under such averaging.

2. We now establish a Hellinger bracketing bound for the observed-density class. Fix $0 < \eta < 1$, and for any $b \in \Delta(\mathcal{X} \times [J(n)])$, define

$$b_{xj}^- := \left( \frac{\eta^2}{|\mathcal{X}|J(n)} \right) \left\lfloor \frac{b_{xj}}{\left( \frac{\eta^2}{|\mathcal{X}|J(n)} \right)} \right\rfloor, \quad b_{xj}^+ := b_{xj}^- + \frac{\eta^2}{|\mathcal{X}|J(n)}, \quad x \in \mathcal{X}, \ j \in [J(n)].$$

Then by construction $b_{xj}^- \leq b_{xj} \leq b_{xj}^+$ for all $x, j$.

Let $\underline{f}_b$ and $\overline{f}_b$ be the observed-data functions obtained from the same formula defining $f_b$, but replacing $b_{xj}$ by $b_{xj}^-$ and $b_{xj}^+$, respectively. That is, for an observed datum $o = (\widetilde{x}_m, x_o, m, y)$, with $y \in Y_{n,j(y)}$ :

$$\underline{f}_b(o) := \pi^\star(m \mid x_o, y) \sum_{x_m \in A(\widetilde{x}_m, m)} \frac{b_{(x_o, x_m), j(y)}^-}{\lambda(Y_{n,j(y)})},$$

and

$$\overline{f}_b(o) := \pi^\star(m \mid x_o, y) \sum_{x_m \in A(\widetilde{x}_m, m)} \frac{b_{(x_o, x_m), j(y)}^+}{\lambda(Y_{n,j(y)})}$$

Since $b_{xj}^- \leq b_{xj} \leq b_{xj}^+$, we naturally have $\underline{f}_b \leq f_b \leq \overline{f}_b$. Thus $[\underline{f}_b, \overline{f}_b]$ is a **bracket** containing $f_b$.

We next bound its Hellinger width, which is quite standard. Using $(\sqrt{u} - \sqrt{v})^2 \leq u - v, 0 \leq v \leq u$, we obtain

$$H^2(\underline{f}_b, \overline{f}_b) = \int \left( \sqrt{\overline{f}_b} - \sqrt{\underline{f}_b} \right)^2 d\mu$$

$$\leq \int (\overline{f}_b - \underline{f}_b) d\mu$$

By linearity of the observation map, the integral of the observed-density function generated by a nonnegative cell-mass vector $c = (c_{xj})$ equals $\sum_{x,j} c_{xj}$. Hence

$$\int (\overline{f}_b - \underline{f}_b) d\mu = \sum_{x \in \mathcal{X}} \sum_{j=1}^{J(n)} (b_{xj}^+ - b_{xj}^-) \leq |\mathcal{X}|J(n) \frac{\eta^2}{|\mathcal{X}|J(n)} = \eta^2,$$

Thus $H(\underline{f}_b, \overline{f}_b) \le \eta$, It remains to count the number of such brackets. For each pair $(x, j)$, the lower endpoint $b_{xj}^-$ takes values in the grid $\{0, \frac{\eta^2}{|\mathcal{X}|J(n)}, 2\frac{\eta^2}{|\mathcal{X}|J(n)}, \ldots, 1\}$ Hence the number of possible lower arrays $(b_{xj}^-)$, and therefore the number of brackets, is at most $\left(1 + \frac{|\mathcal{X}|J(n)}{\eta^2}\right)^{|\mathcal{X}|J(n)}$ Consequently,

$$\log N_{[]}(\eta, \mathcal{F}_n, H) \le |\mathcal{X}|J(n) \log\left(1 + \frac{|\mathcal{X}|J(n)}{\eta^2}\right)$$

**Step B4: Positivity transfers observed Hellinger to full Hellinger.**

We now use complete-case positivity. Let $\pi_0^\star(x_o, y) := \mathbb{P}^\star(M = 0 \mid X_o = x_o, Y = y) \ge \lambda_{\min}$ (by assumption 2.4).

On the complete-case slice $M = 0$, all covariates are observed. Hence, under $P^\star$, the observed density on this slice is $\pi_0^\star(x_o, y)p^\star(x_o, x_m, y)$ while under $Q_q^H$ it is $\pi_0^\star(x_o, y)q^H(x_o, x_m, y)$.

Therefore,

$$\widetilde{H}^2(P^\star, Q_q^H) \ge \sum_{x_o, x_m} \int_{\mathcal{Y}} \left[\sqrt{\pi_0^\star(x_o, y)p^\star(x_o, x_m, y)} - \sqrt{\pi_0^\star(x_o, y)q^H(x_o, x_m, y)}\right]^2 dy$$

$$= \sum_{x_o, x_m} \int_{\mathcal{Y}} \pi_0^\star(x_o, y) \left[\sqrt{p^\star(x_o, x_m, y)} - \sqrt{q^H(x_o, x_m, y)}\right]^2 dy$$

$$\ge \lambda_{\min} \sum_{x_o, x_m} \int_{\mathcal{Y}} \left[\sqrt{p^\star(x_o, x_m, y)} - \sqrt{q^H(x_o, x_m, y)}\right]^2 dy$$

$$= \lambda_{\min} H^2(P^\star, Q_q^H)$$

It follows that $H(P^\star, Q_q^H) \le (\lambda_{\min})^{-1/2}\widetilde{H}(P^\star, Q_q^H)$, and consequently $\sup_{q \in \mathcal{M}_n(\varepsilon_n)} H(P^\star, Q_q^H) \to 0$. Since Hellinger convergence implies total variation convergence, and total variation dominates $d_{LP}$,

$$\sup_{q \in \mathcal{M}_n(\varepsilon_n)} d_{LP}(Q_q^H, P^\star) \to 0$$

**Step B5: Conclude the result.** By the triangle inequality,

$$d_{LP}(Q_q^A, P^\star) \le d_{LP}(Q_q^A, Q_q^H) + d_{LP}(Q_q^H, P^\star),$$

Taking suprema over $q \in \mathcal{M}_n(\varepsilon_n)$ gives

$$\sup_{q \in \mathcal{M}_n(\varepsilon_n)} d_{LP}(Q_q^A, P^\star) \le h_J + \sup_{q \in \mathcal{M}_n(\varepsilon_n)} d_{LP}(Q_q^H, P^\star)$$

Both terms on the right converge to zero. Hence $\sup_{q \in \mathcal{M}_n(\varepsilon_n)} d_{LP}(Q_q^A, P^\star) := r_n \to 0$

We now derive the final bound on the conditional Lévy–Prokhorov distance using only the lower bound $\mathbb{P}_X^*(\bar{x}) \ge \epsilon_0$. Applying Lemma C.7 with sufficiently large $n$ and $d_{LP}(\mathbb{P}_{XY}, \mathbb{P}_{XY}^*) \le (r_n + \frac{3}{2}\delta_n)$ from Step (B4) yields

$$d_{LP}\left(\mathbb{P}(Y \mid \bar{x}), \mathbb{P}^*(Y \mid \bar{x})\right) \le \frac{2}{\epsilon_0}\left(r_n + \frac{3}{2}\delta_n\right) \to 0 \tag{B}$$

**Step C. Upper–bound on $\limsup \Phi_n'$.** Let $z \in \mathcal{Z}$ be arbitrary. Since by (B)

$$d_{LP}\left(\mathbb{P}(Y \mid \bar{x}), \mathbb{P}^*(Y \mid \bar{x})\right) \le \Delta_n := \frac{1}{\epsilon_0}\left(r_n + \frac{3}{2}\delta_n\right),$$

we can control the difference in expectations via the dual characterization of LP distance (see e.g. Corollary 11.6.5 of (Dudley, 2018)): if $f$ is any bounded Lipschitz function on $\mathcal{Y}$ with $\|f\|_{BL}$, then

$$\left|\mathbb{E}_{\mathbb{P}}[f(Y)] - \mathbb{E}_{\mathbb{P}^*}[f(Y)]\right| \le 2\|f\|_{BL} \cdot d_{LP}(\mathbb{P}, \mathbb{P}^*) \le 2\|f\|_{BL} \cdot \Delta_n.$$

By assumption 2.1, for each fixed $z$, the cost function $y \mapsto c(z, y)$ is bounded and Lipschitz, thus it belongs to $\mathrm{BL}(\mathcal{Y})$.

For each $n$, choose $z_n \in \mathcal{Z}$ so that

$$\Phi^\star \;\geq\; \mathbb{E}_{\mathbb{P}^*}[\, c(z_n, Y) \mid X = \bar{x}\,] \;-\; \frac{1}{n}$$

Then

$$\Phi'_n \;\leq\; \sup_{\mathbb{P}(Y|\bar{x})} \mathbb{E}_{\mathbb{P}}[\, c(z_n, Y) \mid X = \bar{x}\,] \;\leq\; \mathbb{E}_{\mathbb{P}^*}[\, c(z_n, Y) \mid X = \bar{x}\,] + L_c\, \Delta_n,$$

where $L_c = 2\sup_{z \in \mathcal{Z}} \|c(z, \cdot)\|_{\mathrm{BL}}$. Hence

$$\Phi'_n \;\leq\; \Phi^\star + L_c\, \Delta_n + \tfrac{1}{n} \tag{C1}$$

Because $\Delta_n, \tfrac{1}{n} \to 0$ as $n \to \infty$, (C1) implies

$$\limsup_{n \to \infty} \Phi_n \;\leq\; \limsup_{n \to \infty} \Phi'_n \leq \Phi^*$$

This completes the upper–bound portion of the proof; together with the almost–sure lower bound established via Borel–Cantelli, we conclude that $\Phi_n \downarrow \Phi^*$ almost surely. The remaining assertion follows the same logic as in the proof C.5.

$\square$

## D. Additional Results

### D.1. Effective Problem Size after Aggregation

Recall that $n_\xi$ denotes the number of samples with $(\tilde{x}_i, y_i, m_i) = \xi$, and let $\widehat{\Sigma} \subseteq \tilde{\mathcal{X}} \times \mathcal{Y} \times \mathcal{M}$ be the empirical support of $(\tilde{x}, y, m)$. Define an indicator matrix $T' \in \{0,1\}^{|\widehat{\Sigma}| \times |\mathcal{X} \times \mathcal{Y}|}$ by

$$T'_{\xi,(x,y)} = 1 \quad \Longleftrightarrow \quad (y = y_\xi, \ x_o = \tilde{x}_{o,\xi}, \ x_m \in \mathcal{A}(\tilde{x}_{m,\xi}, m_\xi)),$$

i.e., $(x, y)$ is compatible with the observation $\xi = (\tilde{x}_\xi, y_\xi, m_\xi)$. Then the feasible set $\mathcal{D}(\gamma)$ in (4) admits the following equivalent aggregated form:

$$\mathcal{D}'(\gamma) := \Big\{ p \in \Delta(\mathcal{X} \times \mathcal{Y}) : \ \sum_{\xi \in \widehat{\Sigma}} \frac{n_\xi}{n} \log\big((T'p)_\xi\big) \geq \gamma \Big\},$$

with the same $\gamma$.

This aggregation yields a reformulation of the resulting conic program in which the auxiliary variables associated with the log terms are indexed by $\xi \in \widehat{\Sigma}$ rather than by samples $i \in [n]$.

$$\inf_{\lambda \geq 0, \ \mu \in \mathbb{R}^{|\widehat{\Sigma}|}_-, \ \nu \in \mathbb{R}, \ \alpha \in \mathbb{R}, \ z \in \mathcal{Z}} \quad -\alpha$$

$$\text{s.t.} \quad c(z) - (T')^\top \mu + \nu \mathbf{1} + \alpha d \leq 0,$$

$$\lambda \left( n(1 + \gamma) + \sum_{\xi \in \widehat{\Sigma}} n_\xi \log\left(\frac{-\mu_\xi}{n_\xi \lambda}\right) \right) + \nu \geq 0.$$

Here $\mu = (\mu_\xi)_{\xi \in \widehat{\Sigma}}$ is indexed by aggregated observations $\xi \in \widehat{\Sigma}$ rather than by individual samples $i \in [n]$. Therefore, the size of the reformulated problem is governed by $|\widehat{\Sigma}|$ instead of $n$.

### D.2. Missing both Covariate and Outcome

We show that the model developed in the section 3.1 extends naturally to a setting in which **both** selected covariate components and the outcome itself may be missing. Let $X = (X_o, X_m), Y = (Y_o, Y_m)$. We retain the standard *MAR* assumption as in 2.5, now adapted to the joint covariate–outcome setting. Specifically, the joint missingness mechanism satisfies

$$\mathbb{P}\big(M \mid X_o = x_o, \ Y_o = y_o, \ X_m = x_m, \ Y_m = y_m\big) = \mathbb{P}\big(M \mid X_o = x_o, \ Y_o = y_o\big).$$

For any joint distribution $\mathbb{Q}_{XY}$ and any missingness kernel $q(M \mid X_o, Y_o)$, the induced observable distribution is given by

$$\mathrm{H}(\mathbb{Q}_{XY}, \mathbb{Q}(M \mid X_o, Y_o))(\tilde{x}, \tilde{y}, m) = \mathbb{Q}(m \mid x_o, y_o) \sum_{(x_m, y_m) \in \mathcal{A}(\tilde{x}_m, \tilde{y}_m)} \mathbb{Q}_{XY}(x_o, x_m, y_o, y_m),$$

where $\mathcal{A}(\tilde{x}_m, \tilde{y}_m)$ denotes the set of latent covariate–outcome completions consistent with the masked values $(\tilde{x}_m, \tilde{y}_m)$ and the missingness pattern $m$. The Kullback–Leibler divergence between the empirical distribution and the model-induced distribution admits the decomposition

$$D_{\mathrm{KL}}\left(\widehat{\mathbb{P}} \,\Big\|\, \mathrm{H}(\mathbb{Q}, q)\right) = \sum_{\xi \in \widehat{\Sigma}} \frac{n_\xi}{n} \log\left(\frac{n_\xi}{n}\right) - \frac{1}{n} \sum_{i=1}^{n} \log \mathbb{Q}(m_i \mid x_{o,i}, y_{o,i})$$

$$- \frac{1}{n} \sum_{i=1}^{n} \log \left( \sum_{(x_m, y_m) \in \mathcal{A}(\tilde{x}_{m,i}, \tilde{y}_{m,i})} \mathbb{Q}_{XY}(x_{o,i}, x_m, y_{o,i}, y_m) \right).$$

Accordingly, we define the ambiguity set

$$\mathcal{D}_{\text{joint-missing}}(\gamma) := \left\{ p \in \Delta(\mathcal{S}) \ \middle|\ \frac{1}{n} \sum_{i=1}^{n} \log \left( \sum_{(x_m, y_m) \in \mathcal{A}(\tilde{x}_{m,i}, \tilde{y}_{m,i})} \mathbb{Q}_{XY}(x_{o,i}, x_m, y_{o,i}, y_m) \right) \geq \gamma \right\}.$$

Relative to the base model, the resulting optimization problem differs only through an updated indicator matrix $T$. For each sample $i = 1, \ldots, n$ and support point $j \equiv (x_j, y_j) = (x_{o,j}, x_{m,j}, y_{o,j}, y_{m,j}) \in \mathcal{S}$, define

$$
T_{ij} = \begin{cases} 1, & \text{if } x_{o,j} = x_{o,i}, \; y_{o,j} = y_{o,i}, \; (x_{m,j}, y_{m,j}) \in \mathcal{A}(\tilde{x}_{m,i}, \tilde{y}_{m,i}), \\ 0, & \text{otherwise.} \end{cases}
$$

The fractional program structure, transformation techniques, and solver complexity therefore remain unchanged. Moreover, under the same regularity conditions as in the discrete model, the convergence arguments and out-of-sample guarantees extend analogously; we omit the proofs for brevity.

As a special case, the framework reduces to a non-contextual optimization problem when $|\mathcal{X}| = 1$ and the covariate is always observable.

## D.3. Additional Analysis of Computational Structure and Reformulations

Similar to the discrete case, the computational bottleneck here is not governed by the sample size $n$, as the aggregation argument in Appendix D.1 applies verbatim.Instead, the main computational challenge arises from the dimensionality of the covariates and the size of the empirical support $|\widehat{\mathcal{Y}}|$. Although the dual formulation in Theorem 3.13 is convex, its direct solution becomes impractical when the covariate dimension is high or when $|\widehat{\mathcal{Y}}|$ is large. Moreover, the presence of exponential-cone constraints and semi-infinite conditions requires additional care in both modeling and numerical resolution. In what follows, we introduce a sequence of structural refinements that together render the resulting optimization program computationally tractable.

### D.3.1. ROW GENERATION FOR THE BULK CONSTRAINT SET

The system in Theorem 3.13 contains $|\mathcal{X} \times \widehat{\mathcal{Y}}| - K$ inequalities of the form

$$
-\left(T^{\mathsf{T}} \varphi\right)_j - \mu \leq 0, \qquad j \notin I,
$$

which can be exorbitantly many when $|\mathcal{X} \times \widehat{\mathcal{Y}}|$ is large (high-dimensional or a large support set) . Because only a handful of these inequalities are ever active at optimality, we enforce them via a standard *row-generation* procedure.

Because each iteration adds at least one new inequality and the candidate set $\{j \notin I\}$ is finite, the procedure converges in no more than $|\mathcal{X} \times \widehat{\mathcal{Y}}| - K$ iterations, while each separation step is a linear scan of the vector $T^{\mathsf{T}} \varphi$ and selecting its minimum entry.

### D.3.2. SEMI−INFINITE PROGRAM

The formulation in Theorem 3.13 can often be simplified further when $c$ exhibits additional structure. In what follows we show how such structure can reduce the computational burden associated with the two most challenging expressions,

$$
\max_{y \in \mathcal{Y}} c(z, y) \tag{$\star$}
$$

$$
c^{\mathcal{N}}(z, y_i) \tag{$\star\star$}
$$

We begin with the separable-cost case and subsequently outline other tractable specialisations.

**(1) Separable cost.**

**Assumption D.1** (Condition for separable form)**.** Let $c : \mathcal{Z} \times \mathcal{Y} \to \mathbb{R}$ satisfy: *(i)* for every $y \in \mathcal{Y}$, the map $c(\,\cdot\,, y)$ is convex in $z$; *(ii)* for every $z \in \mathcal{Z}$, the map $c(z, \cdot)$ is locally Lipschitz and $\mathcal{Y}$ is connected; and *(iii)* the (Clarke) sub-gradient with respect to $y$ is independent of $z$, i.e. $\partial_y c(z, y) = \partial_y c(z', y)$ for all $z, z' \in \mathcal{Z}$ and $y \in \mathcal{Y}$.

Fix $z_1, z_2 \in \mathcal{Z}$ and let $\Delta_{12}(y) := c(z_1, y) - c(z_2, y)$ for $y \in \mathcal{Y}$. Each $c(z, \cdot)$ is locally Lipschitz, hence so is $\Delta_{12}$. Assumption D.1(iii) gives $\partial_y c(z_1, y) = \partial_y c(z_2, y)$; thus $0 \in \partial \Delta_{12}(y)$. By Clarke (1990, Lemma 2.5.1), a locally Lipschitz function on a connected set whose sub-differential contains the origin is constant, so $\Delta_{12}(y) \equiv$ const.

Choose a reference pair $(z^\star, y^\star)$ and define $g(z) := c(z, y^\star)$ and $h(y) := c(z^\star, y) - c(z^\star, y^\star)$. Constancy of $\Delta_{z,z^\star}$ yields $c(z, y) = g(z) + h(y)$ for all $(z, y)$, and convexity of $g$ follows from that of $c(\cdot, y^\star)$.

It fllows that under Assumption D.1, there exist functions $g : \mathcal{Z} \to \mathbb{R}$ and $h : \mathcal{Y} \to \mathbb{R}$ such that

$$c(z, y) = g(z) + h(y) \qquad \forall (z, y) \in \mathcal{Z} \times \mathcal{Y},$$

and $g$ is convex.

We can thus decompose the troublesome terms as

$$\max_{y \in \mathcal{Y}} c(z, y) = g(z) + \max_{y \in \mathcal{Y}} h(y),$$
$$c^{\mathcal{N}}(z, y_i) = g(z) + h^{\mathcal{N}}(y_i), \qquad i \in [K], \tag{13}$$

where $H := \max_{y \in \mathcal{Y}} h(y)$ and $h_i := h^{\mathcal{N}}(y_i)$ can be pre-computed off-line (e.g., by solving $K + 1$ one-dimensional problems).

Substituting (13) into the dual of Theorem 4 yields

$$\begin{aligned} M &\geq g(z) + H, \\ -(T^{\mathsf{T}}\boldsymbol{\varphi})_{I(i)} - \mu + \eta + g(z) + h_i + \rho &\leq 0, \quad \forall i \in [K] \end{aligned} \tag{14}$$

*Remark.* In many applications $H$ and the collection $\{h_i\}$ need not be computed explicitly.

*Example* D.2. Suppose that, for each fixed $z \in \mathcal{Z}$, the cost takes the form of a pointwise maximum over a finite family of affine functions in $y$, viz.

$$c(z, y) = \max_i \{\mathbf{a}_i^{\mathsf{T}} y + b_i(z)\},$$

and that the outcome set $\mathcal{Y}$ is a compact polyhedron whose finite set of extreme points is denoted $\Omega \subset \mathcal{Y}$.

Because the maximand in (13) is affine in $y$, the inflated cost admits the closed form

$$c^{\mathcal{N}}(z, y_i) = \max_i \{\mathbf{a}_i^{\mathsf{T}}(y_i + \delta \mathbf{d}_i^+) + b_i(z), \ \mathbf{a}_i^{\mathsf{T}}(y_i - \delta \mathbf{d}_i^+) + b_i(z)\},$$

where $\mathbf{d}_i^+$ is the unit vector in the direction of $\mathbf{a}_i$.

By convexity of $c(\cdot, y)$ and the linearity in $y$, the maximum in $(\star)$ is attained at an extreme point of $\mathcal{Y}$; hence

$$\max_{y \in \mathcal{Y}} c(z, y) = \max_{y' \in \Omega} c(z, y'),$$

where $c(z, y') = \max_i \{\mathbf{a}_i^{\mathsf{T}} y' + b_i(z)\}$.

Let $c_i(z, y') := \mathbf{a}_i^{\mathsf{T}} y' + b_i(z)$ and $c_j(z, y_i \pm \delta \mathbf{d}_j^+)$ be defined analogously. The block (14) then reduces to

$$\begin{aligned} M &\geq c_i(z, y'), \qquad \forall y' \in \Omega, \ \forall i, \\ -(T^{\mathsf{T}}\boldsymbol{\varphi})_{I(i)} + \eta + c_j(z, y_i + \delta \mathbf{d}_j^+) + \rho &\leq \mu, \quad \forall i \in [K], \ \forall j, \\ -(T^{\mathsf{T}}\boldsymbol{\varphi})_{I(i)} + \eta + c_j(z, y_i - \delta \mathbf{d}_j^+) + \rho &\leq \mu, \quad \forall i \in [K], \ \forall j \end{aligned} \tag{15}$$

Because the sets $\Omega$ and $\{\mathbf{a}_i, b_i(\cdot)\}$ are finite, (15) contains only finitely many linear constraints, and can therefore be solved directly with off-the-shelf conic solvers without computing $H$ or $h_i$ in (13).

**(2) Affine, non-separable cost.**

We next treat the classical setting in which the decision and outcome variables are not separable but $c$ remains affine in $y$.

**Assumption D.3** (Polyhedral uncertainty and affine coefficients). $\mathcal{Y} \subset \mathbb{R}^{d_y}$ is a non-empty polyhedron, $\mathcal{Y} = \{y \in \mathbb{R}^{d_y} : Fy \leq f\}$. For fixed $y \in \mathcal{Y}$ the map $z \mapsto c(z,y)$ is affine and can be written

$$c(z,y) = \max_{i \in \mathcal{J}} \{a_i(y) + b_i(y)^\mathsf{T} z\},$$

where the coefficients are themselves affine in $y$:

$$a_i(y) = a_i^\mathsf{T} y + a_{i,0}, \qquad b_i(y) = B_i y + B_{i,0}. \tag{16}$$

We can now eliminate the semi-infinite constraint. The inner inequality $\max_{y \in \mathcal{Y}} c(z,y) \leq M$ is equivalent to

$$a_i(y) + b_i(y)^\mathsf{T} z \ \leq \ M, \qquad \forall y \in \mathcal{Y}, \ \forall i \in \mathcal{J}.$$

Using (16), $a_i(y) + b_i(y)^\mathsf{T} z = (a_i + B_i^\mathsf{T} z)^\mathsf{T} y + (a_{i,0} + B_{i,0}^\mathsf{T} z)$. For each fixed $i$ the worst-case value of this linear form over the polyhedron $\mathcal{Y}$ is obtained through the LP dual: there exists a non-negative multiplier $g_i \in \mathbb{R}^m$ (with $m$ rows in $F$) such that

$$F^\mathsf{T} g_i = a_i + B_i^\mathsf{T} z, \qquad f^\mathsf{T} g_i + (a_{i,0} + B_{i,0}^\mathsf{T} z) \leq M, \qquad g_i \geq 0, \ \forall i \in \mathcal{J}. \tag{17}$$

No duality gap arises provided $\mathcal{Y}$ has an interior point (Slater's condition).

If the neighbourhood $\mathcal{N}$ is an Euclidean ball of radius $\delta$, the inflated cost $c^\mathcal{N}(z, y_i) = \max_{y \in \mathcal{Y}, \|y-y_i\| \leq \delta} c(z,y)$ translates, after the same dual trick (with a dual variable $h_k$), to

$$\left\| F^\mathsf{T} h_k - (a_k + B_k^\mathsf{T} z) \right\|_2 \leq \delta, \quad f^\mathsf{T} h_k + (a_{k,0} + B_{k,0}^\mathsf{T} z) \leq \mu - \rho - \eta + (T^\mathsf{T} \varphi)_{I(i)}, \ \forall k \in \mathcal{J}, \ \forall i \in [K]. \tag{18}$$

Substituting (17)–(18) into the master conic dual yields a finite system—linear and second-order cone constraints—that is amenable to standard conic solvers without any remaining semi-infinite terms.

**(3) Quadratic, non-separable cost.**

**Assumption D.4** (Polyhedral uncertainty & quadratic cost). $\mathcal{Y} \subset \mathbb{R}^{d_y}$ is a non-empty polyhedron, $\mathcal{Y} = \{y \in \mathbb{R}^{d_y} : Ay \leq b\}$. For each $y \in \mathcal{Y}$ the cost is convex quadratic in $z$:

$$c(z,y) = z^\mathsf{T} Q(y) z + r(y)^\mathsf{T} z + s(y), \qquad Q(y) \succeq 0,$$

with affine dependence

$$Q(y) = Q_0 + \sum_{p=1}^{d_y} y_p Q_p, \quad r(y) = r_0 + \sum_{p=1}^{d_y} y_p r_p, \quad s(y) = s_0 + \sum_{p=1}^{d_y} y_p s_p.$$

Such quadratic specifications frequently appear in ridge-regularized regression and in DC optimal power-flow models for power systems.

Lift $z$ to the matrix variable $Z := zz^\mathsf{T}$ so that $\mathrm{tr}(QZ) = z^\mathsf{T} Q z$. Then

$$c(z,y) = \mathrm{tr}(Q_0 Z) + r_0^\mathsf{T} z + s_0 + \sum_{p=1}^{d_y} y_p \big( \mathrm{tr}(Q_p Z) + r_p^\mathsf{T} z + s_p \big).$$

For fixed $z$ the inner maximisation $\max_y \{c(z,y) \,|\, Ay \leq b\}$ is linear in $y$ and admits the LP dual

$$\min_{g \geq 0} \ b^\mathsf{T} g \quad \text{s.t.} \quad A^\mathsf{T} g = \big[ \mathrm{tr}(Q_p Z) + r_p^\mathsf{T} z + s_p \big]_{p=1}^{d_y}.$$

The only non-linearities that remain are the terms $\mathrm{tr}(Q_p Z)$ and the rank-one constraint $Z = zz^\mathsf{T}$, both of which become linear after lifting and adding a single LMI:

$$\begin{pmatrix} 1 & z^\mathsf{T} \\ z & Z \end{pmatrix} \succeq 0.$$

Combining the above, the semi-infinite constraint $\max_{y \in \mathcal{Y}} c(z, y) \leq M$ is equivalent to the SDP

$$\begin{pmatrix} 1 & z^\mathsf{T} \\ z & Z \end{pmatrix} \succeq 0, \qquad A^\mathsf{T} g = \big[ \mathrm{tr}(Q_p Z) + r_p^\mathsf{T} z + s_p \big]_{p=1}^{d_y},$$

$$b^\mathsf{T} g + \mathrm{tr}(Q_0 Z) + r_0^\mathsf{T} z + s_0 \;\leq\; M, \qquad g \geq 0.$$

When the inflated cost involves a local ball $\|y - y_i\|_2 \leq \delta$, define $w := y - y_i$ and $f(i) := b - A y_i$. Then the robust counterpart becomes

$$\max_{\substack{A w \leq f(i) \\ \|w\|_2 \leq \delta}} \big[ \mathrm{tr}(Q_1 Z) + r_1^\mathsf{T} z + s_1, \ldots, \mathrm{tr}(Q_{d_y} Z) + r_{d_y}^\mathsf{T} z + s_{d_y} \big] w,$$

whose SOCP dual yields the mixed SDP–SOCP constraint

$$\begin{pmatrix} 1 & z^\mathsf{T} \\ z & Z \end{pmatrix} \succeq 0,$$

$$\Big\| g^\mathsf{T} f(i) + \delta \big[ A^\mathsf{T} g - \big( \mathrm{tr}(Q_p Z) + r_p^\mathsf{T} z + s_p \big)_{p=1}^{d_y} \big] \Big\|_2 \leq \mu - \rho - \eta + (T^\mathsf{T} \varphi)_{I(i)}, \quad g \geq 0.$$

The resulting master problem consists solely of linear, SDP and SOCP blocks plus the exponential-cone constraints, and can be handled by modern mixed conic solvers.

## D.4. Continuous Covariates

We briefly discuss how the proposed framework can be extended to settings with continuous covariates. Suppose that both $X$ and $Y$ take values in compact subsets of Euclidean spaces, and that Lebesgue measures serve as the reference measures in Assumption 2.2. This setting introduces two technical difficulties that are absent in the finite-covariate case.

First, exact conditioning on a target covariate value is no longer statistically stable. If $X$ is continuously distributed, then

$$\mathbb{P}(X = \bar{\boldsymbol{x}}) = 0, \qquad \bar{\boldsymbol{x}} \in \mathcal{X},$$

so guarantees relying on a positive mass condition such as $\mathbb{P}(X = \bar{\boldsymbol{x}}) \geq \epsilon_0 > 0$ cannot be used directly. Second, the set of feasible completions of a partially observed covariate is no longer finite. Given a partially observed covariate $\tilde{x}_m$, the set $\mathcal{A}(\tilde{x}_m)$ of compatible completions is typically an uncountable subset of the covariate space.

To address these issues, we use a two-step approximation. We first construct a finite empirical support for the missing covariates, and then smooth the resulting finite-support model in the $x$-coordinate.

**Empirical virtual support.** For each missing-covariate coordinate $j$, let

$$\mathcal{X}_{\mathrm{cand},m}^{(j)} := \Big\{ x_m^{(j)} : x_m^{(j)} \text{ is observed in at least one sample} \Big\}$$

be the empirical candidate support for that coordinate. The full candidate support for the missing block is then

$$\mathcal{X}_{\mathrm{cand},m} := \prod_{j=1}^{d_m} \mathcal{X}_{\mathrm{cand},m}^{(j)}.$$

Given an observation $(x_o, \tilde{x}_m)$, define the compatible empirical completion set

$$\mathcal{X}_{\mathrm{cand}}(x_o, \tilde{x}_m) := \Big\{ (x_o, x_m) : x_m^{(j)} = \tilde{x}_m^{(j)} \text{ if } \tilde{x}_m^{(j)} \neq \mathrm{NA}, \quad x_m^{(j)} \in \mathcal{X}_{\mathrm{cand},m}^{(j)} \text{ otherwise} \Big\}.$$

The overall virtual support is

$$\mathcal{X}_{\text{cand}} := \bigcup_{i=1}^{n} \mathcal{X}_{\text{cand}}(x_{o,i}, \tilde{x}_{m,i}).$$

By construction, $\mathcal{X}_{\text{cand}}$ is finite and contains only covariate vectors that are compatible with the observed data and the empirical coordinate-wise support.

Using this virtual support, we define the empirical likelihood class

$$\mathcal{Q}_{\text{cand},n}(\gamma) := \left\{ \mathbb{Q} : \text{supp}(\mathbb{Q}_X) \subseteq \mathcal{X}_{\text{cand}}, \quad \frac{1}{n} \sum_{i=1}^{n} \log \left( \sum_{x \in \mathcal{X}_{\text{cand}}(x_{o,i}, \tilde{x}_{m,i})} \mathbb{Q}(x, y_i) \right) \geq \gamma \right\}.$$

Here $\mathbb{Q}(x, y_i)$ denotes the mass assigned by $\mathbb{Q}$ to the atom $(x, y_i)$. This likelihood constraint is the finite-support analogue of the KL-based observable likelihood constraint used in the discrete-covariate setting.

As in the continuous-outcome case, we may further restrict the candidate laws to a sieve class. Let

$$\Pi_J = \{\pi_1, \ldots, \pi_J\}$$

be a measurable partition of $\mathcal{X} \times \mathcal{Y}$, and let $\mathcal{B}_J$ denote the class of laws whose density or mass representation is constant on each active support component within every bin $\pi_j$. Define the mesh size

$$h_J := \max_{1 \leq j \leq J} \text{diam}(\pi_j).$$

The sieve restriction controls the complexity of the ambiguity set and provides a finite-dimensional approximation to the continuous joint law.

**Kernel smoothing in the covariate coordinate.** The virtual support construction yields a finite-support law on $\mathcal{X}_{\text{cand}} \times \mathcal{Y}$. However, it still does not define a stable conditional law at a target point $\bar{x} \notin \mathcal{X}_{\text{cand}}$. We therefore smooth the finite-support law in the $x$-coordinate.

Let $K : \mathbb{R}^{d_x} \to \mathbb{R}_+$ be a measurable kernel satisfying

$$\int_{\mathbb{R}^{d_x}} K(u) \, du = 1,$$

and define

$$K_h(u) := h^{-d_x} K(u/h).$$

For a probability measure $\mathbb{Q}$ supported on $\mathcal{X}_{\text{cand}} \times \mathcal{Y}$, define its $x$-smoothed version $\widehat{\mathbb{Q}}_h$ by

$$\widehat{\mathbb{Q}}_h(A \times B) := \sum_{x_i \in \mathcal{X}_{\text{cand}}} \left[ \int_A K_h(x - x_i) \, dx \right] \mathbb{Q}(\{x_i\} \times B),$$

for Borel sets $A \subseteq \mathbb{R}^{d_x}$ and $B \subseteq \mathcal{Y}$. Equivalently, $\widehat{\mathbb{Q}}_h$ is obtained by replacing each atom $x_i$ in the covariate coordinate by the kernel density $K_h(\cdot - x_i)$, while preserving the conditional law of $Y$ given $X = x_i$.

If $\mathbb{Q}(\{x_i\} \times dy) = q_i(y) \, dy$, then the smoothed joint density takes the form

$$\widehat{q}_h(x, y) = \sum_{x_i \in \mathcal{X}_{\text{cand}}} K_h(x - x_i) q_i(y).$$

Consequently, whenever the denominator is positive, the smoothed conditional law of $Y$ given $X = x$ has density

$$\widehat{q}_h(y \mid x) = \frac{\sum_{x_i \in \mathcal{X}_{\text{cand}}} K_h(x - x_i) q_i(y)}{\sum_{x_i \in \mathcal{X}_{\text{cand}}} K_h(x - x_i) \mathbb{Q}_X(\{x_i\})}.$$

This smoothing step allows us to evaluate conditional distributions at target covariate values $\bar{x}$ that do not appear exactly in the data.

Define the smoothed candidate class

$$\widehat{\mathcal{Q}}_{h,n}(\gamma; J) := \left\{ \widehat{\mathbb{Q}}_h : \widehat{\mathbb{Q}}_h = K_h *_x \mathbb{Q}, \quad \mathbb{Q} \in \mathcal{Q}_{\text{cand},n}(\gamma) \cap \mathcal{B}_J \right\}.$$

**Smoothed conditional ambiguity set.** We now define the conditional ambiguity set used for continuous covariates. For a target covariate value $\bar{x}$, let

$$\mathcal{K}_h(\delta, \gamma; J) := \left\{ \mathbb{P}(\cdot \mid \bar{x}) : \begin{array}{l} \exists \mathbb{Q} \in \mathcal{Q}_{\text{cand},n}(\gamma) \cap \mathcal{B}_J \text{ such that } \widehat{\mathbb{Q}}_h = K_h *_x \mathbb{Q}, \\ \mathbb{P}(\cdot \mid \bar{x}) \in B_{\text{LP},\delta}(\widehat{\mathbb{Q}}_h(\cdot \mid \bar{x})) \end{array} \right\}.$$

Thus, the likelihood constraint is imposed on the finite empirical support, while the conditional law at $\bar{x}$ is evaluated through the smoothed model.

The resulting robust contextual optimization problem is

$$\Phi_h(\bar{x}) := \inf_{z \in \mathcal{Z}} \sup_{\nu \in \mathcal{K}_h(\delta, \gamma; J)} \mathbb{E}_\nu[c(z, Y)].$$

This construction preserves the finite-dimensional structure of the empirical likelihood formulation while enabling stable conditional inference at continuous covariate values. A detailed treatment of these results would require substantial additional technical work and is therefore not included in this paper.

# E. Experimental Details

## E.1. Contextual Newsvendor Problem

In the newsvendor setting, a decision-maker must determine an order quantity $z$ to satisfy uncertain demand for a perishable product. The mismatch cost consists of an overage cost $c_o$ per unit (for unsold inventory) and an underage cost $c_u$ per unit (for unmet demand). This leads to the piecewise linear loss function:

$$c(z, y) = c_o(z - y)^+ + c_u(y - z)^+,$$

where $y$ denotes random demand and $(\cdot)^+ := \max\{\cdot, 0\}$.

In many applications, the distribution of demand $y$ is not fixed, but depends on observable covariates $x \in \mathbb{R}^{d_x}$—for instance, customer demographics, weather, regional trends, or economic indicators. Thus, the newsvendor problem becomes contextual, and the optimal order quantity $z$ must adapt to side information through the conditional distribution of $y$ given $x$.

This setting fits naturally into our framework and satisfies the convexity requirement in $z$ for each fixed $y$. We substitute the cost vector $c(z)$ into the dual program in Theorem 3.6, which yields a convex optimization problem. The formulation contains exponential cone constraints arising from the logarithmic terms; such problems can be handled by modern conic solvers. We solve all conic reformulations with MOSEK using default feasibility/optimality tolerances.

## E.2. Instance Generation

Set $c_o = 3, c_u = 6$, and we consider discrete covariates $X \in \{a, b, c\}^3$ and outcomes $Y \in [0, ..., 12]$. We generate $X$ uniformly at random over its support. In Experiment II, the observed covariates are therefore also uniformly sampled. For each context $x \in \mathcal{X}$, the conditional distribution of $Y$ is specified as a mixture of two truncated discrete normals on $\mathcal{Y}$. Let $\phi(y; \mu, \sigma)$ denote the normal density with mean $\mu$ and standard deviation $\sigma$, and define the truncated and renormalized probability mass function

$$g(y; \mu, \sigma) = \frac{\phi(y; \mu, \sigma)}{\sum_{y'} \phi(y'; \mu, \sigma)}$$

To induce heterogeneity across contexts, we assign each $x$ a base-3 index such that

$$\text{frac}(x) = \frac{\sum_{k=1}^{d_x} x_k \, 3^{k-1}}{3^{d_x} - 1} \in [0, 1],$$

which determines smoothly varying peak locations

$$\mu_{\text{L}}(x) = 1 + 5 \, \text{frac}(x), \qquad \mu_{\text{R}}(x) = 12 - 4 \, \text{frac}(x),$$

with fixed widths $\sigma_{\text{L}} = \sigma_{\text{R}} = 1.0$. The two components are given by

$$p_{\text{left}}(y \mid x) = g(y; \mu_{\text{L}}(x), 1.0), \qquad p_{\text{right}}(y \mid x) = g(y; \mu_{\text{R}}(x), 1.0).$$

The conditional law of $Y$ given $X = x$ is then

$$\mathbb{P}(Y \mid X = x) = \left(0.3 + 0.7 \tfrac{x_2 + x_3}{4}\right) p_{\text{left}}(\cdot \mid x) + \left(0.7 - 0.7 \tfrac{x_2 + x_3}{4}\right) p_{\text{right}}(\cdot \mid x),$$

where the weights depend on $(x_2, x_3)$ and further accentuate cross-context variation.

To simulate missing covariates, we generate a binary mask $M$ according to a logistic MAR mechanism driven by the observed coordinates $X_o$ and the realized outcome $Y$. We set the set of potentially-missing coordinates to `mar_indices` $= \{1, 2\}$.

For each observation i and coordinate $j \in$ `mar` $= \{1, 2\}$,

$$\mathbb{P}(M_{ij} = 1 \mid X_i, Y_i) = \frac{\texttt{missing strength}}{1 + \exp\left(-\{a \cdot \sum_{\ell \in \mathcal{O}} X_{i\ell} + b Y_i + c\}\right)},$$

where $\mathcal{O}$ is the observed-index set, $a, b, c$ are fixed logistic coefficients (default $a = 0.2, b = 0.05, c = 0$), and `missing strength` $\in (0, 1]$ (we use notation $\alpha$ in the main text) scales the overall intensity. The observed covariate matrix is

$$\tilde{X}_{ij} = \begin{cases} X_{ij}, & M_{ij} = 0, \\ \text{NaN}, & M_{ij} = 1. \end{cases}$$

Notice that setting a=b=0 reduces to MCAR at rate `missing strength`$/(1 + e^{-c})$; setting `missing strength` $= 0$ yields complete data. Hence each experiment ultimately yields the triplet $(\tilde{X}, Y, M)$, consisting of the partially observed covariates, the realized outcomes, and the missingness mask.

Given the constructed $\mathbb{P}(X)$ and $\mathbb{P}(Y \mid X)$, we generate $n$ i.i.d. samples by: 1. Drawing $X \sim \mathbb{P}(X)$, 2. Drawing $Y \sim \mathbb{P}(Y \mid X)$ using the precomputed PMFs. Across different experimental settings, the sample size $n$ and the missingness strength $\alpha$ are varied accordingly.

In **Experiment I-1** (Investigating the Impact of Radius Settings on Statistical Properties), the missingness strength is fixed at $\alpha = 1$, which corresponds to an overall missing proportion of approximately 0.8. The sample size varies across four levels, namely $n = 10|X| = 270, 50|X|, 250|X|$, and $1250|X|$.

In **Experiment I-2** (Investigating the Impact of Radius Settings on Statistical Properties), the sample size is fixed at $n = 270$, while the missingness strength $\alpha$ takes values in $\{0.1, 0.5, 1\}$, which approximately correspond to missing proportions of 0.2, 0.5, and 0.8, respectively.

In **Experiment II** (Comparison of Overall Method Performance under Varying Sample Sizes), the sample size ranges from 100 to 6000 on a logarithmic scale with bases $\{2, 4, 10\}$, while the missingness strength is fixed at $\alpha = 0.8$.

In **Experiment III**, we consider two complementary settings:

1. **Comparison of Method Performance under Varying Sample Sizes**, where the sample size ranges from 100 to 6000 on a logarithmic scale with bases $\{2, 4, 10\}$ with the missingness strength fixed at $\alpha = 0.8$;

2. **Comparison of Method Performance under Varying Missingness Strength**, where $\alpha$ varies from 0.1 to 1 with the sample size fixed at $n = 270$.

### E.3. Baselines

We evaluate the proposed **MAR–KL DRO** against several baselines:

- **CCA**: complete-case analysis, which discards all samples containing missing entries, followed by empirical risk minimization.

- **CFA** : complete-feature analysis, which retains samples—whether complete or incomplete—whose observed covariates coincide with the target context $\bar{x}$, followed by empirical risk minimization.

- **Impute+SAA**: missing covariates are first imputed via maximum-likelihood estimation under a MAR model (Little & Rubin, 2019), after which the completed data are treated as fully observed and the decision is obtained by sample average approximation (SAA) (Shapiro et al., 2021), i.e., by minimizing the empirical average cost over the imputed samples.

- **Impute+KL–DRO**: the same imputation step as above is performed, after which a standard KL-divergence-based distributionally robust optimization (Ben-Tal et al., 2013) is solved, controlling the worst-case distributional deviations around the imputed empirical distribution.

- **Impute+W1–DRO**: again using the same imputed data, ambiguity is modeled via a Wasserstein–1 ball around the empirical distribution (Mohajerin Esfahani & Kuhn, 2018).

Method names are preserved as *CCA*, *CFA*, *MAR–KL DRO*, *Impute+SAA*, *Impute+KL–DRO*, and *Impute+W1–DRO* in all figures.

### E.4. Ambiguity-radius tuning

Several baselines and our method involve an ambiguity radius parameter (denoted by $\varepsilon$ in the experiments). Within each replication and for each experimental configuration $(n, \alpha)$, we select the radius via a 25%/75% holdout procedure.

**Candidate grid.** We restrict the search to a representative log-scale grid

$$\mathcal{R} := \{\varepsilon_1, \ldots, \varepsilon_K\}, \qquad (\varepsilon_k)_{k=1}^{K} = \text{logspace}(-4, 0, 30),$$

i.e., 30 logarithmically spaced values between $10^{-4}$ and 1. This grid provides broad coverage over several orders of magnitude while remaining computationally feasible.

**Holdout tuning (deployable).** For each replication and each configuration $(n, \alpha)$, we randomly split the training set into a fitting subset (75%) and a validation subset (25%). For every candidate radius $\varepsilon \in \mathcal{R}$, we fit the corresponding decision rule using the 75% subset and compute the predicted cost on the 25% validation subset. We select

$$\hat{\varepsilon} \in \arg\min_{\varepsilon \in \mathcal{R}} \widehat{\text{Cost}}_{\text{val}}(\varepsilon).$$

In case of ties (within numerical tolerance), we break ties by choosing the *largest* radius among minimizers, i.e.,

$$\hat{\varepsilon} = \max\left\{\varepsilon \in \mathcal{R} : \widehat{\text{Cost}}_{\text{val}}(\varepsilon) \leq \min_{\varepsilon' \in \mathcal{R}} \widehat{\text{Cost}}_{\text{val}}(\varepsilon') + \tau\right\},$$

where $\tau > 0$ is a small tolerance (we use $\tau = [\,10^{-8} \ \ 10^{-6}\,]$). This conservative tie-breaking rule favors better reliability when validation costs are indistinguishable.

**Oracle radius (diagnostic only).** To decouple modeling effects from tuning noise, we also report results under an oracle-selected radius. Specifically, within each replication and configuration $(n, \alpha)$, the oracle radius is defined as

$$\varepsilon^\star \in \arg\min_{\varepsilon \in \mathcal{R}} \text{Cost}_{\text{test}}(\varepsilon),$$

i.e., the candidate radius in $\mathcal{R}$ that minimizes the out-of-sample (test) cost. We emphasize that this oracle choice uses test information and is therefore used *only* for diagnostic purposes; all deployable results rely on the holdout tuning above.

### E.5. Definition of reliability

For each method and replication, let $\widehat{\text{Cost}}_{\text{val}}$ be the validation-based estimated cost (under the tuning procedure), and let $\text{Cost}_{\text{test}}$ be the realized cost on the test set. Reliability is defined as

$$\text{Rel} := \frac{1}{R} \sum_{r=1}^{R} \mathbf{1}\{\text{Cost}_{\text{test}}^{(r)} > \widehat{\text{Cost}}_{\text{val}}^{(r)}\},$$

where $R$ is the number of replications.

### E.6. Additional details for Experiment II

The overall comparison across covariates evaluates the quantity $\mathbb{E}_{\bar{X}}\left[\mathbb{E}\left[c(z, Y) \mid X = \bar{X}\right]\right]$. In our experimental setting, the covariates are sampled uniformly over their support. Consequently, we estimate this expectation by taking the uniform average of the out-of-sample conditional costs $\mathbb{E}[c(z, Y) \mid X = \bar{x}]$ over all $\bar{x} \in \mathcal{X}$.

For each training sample size $n$ and each method, we run $R = 100$ independent replications. In replication $r$, let $\text{Cost}_{\text{test}}^{(r)}(\text{method})$ denote the out-of-sample test cost of the method, and let $\text{Cost}_{\text{test}}^{(r)}(\text{MAR–KL})$ denote the corresponding cost of MAR–KL DRO. We report the *relative cost increase* (in percent) defined by $\Delta^{(r)} := 100\left(\frac{\text{Cost}_{\text{test}}^{(r)}(\text{method})}{\text{Cost}_{\text{test}}^{(r)}(\text{MAR–KL})} - 1\right)$.

We report mean, $[q_{20}, q_{80}]$, where mean $:= \frac{1}{R}\sum_{r=1}^{R}\Delta^{(r)}, q_{20} :=$ 20th percentile of $\{\Delta^{(r)}\}_{r=1}^{R}, q_{80} :=$ 80th percentile of $\{\Delta^{(r)}\}_{r=1}^{R}$. Postive values indicate that the method achieves a larger test cost than MAR–KL DRO in that replication.

*Table 6.* Experiment II (Overall comparison across covariates): relative cost increase (%) w.r.t. MAR–KL DRO. Entries are reported as mean$[q_{20}, q_{80}]$ over $R = 100$ replications.

| $n$ | CCA | CFA | Impute+KL–DRO | Impute+SAA | Impute+W1–DRO |
|---|---|---|---|---|---|
| 100 | $25\,[0, 57]$ | $19\,[-12, 43]$ | $17\,[0, 43]$ | $16\,[0, 43]$ | $16\,[-1, 44]$ |
| 166 | $18\,[0, 38]$ | $13\,[-8, 33]$ | $12\,[0, 29]$ | $12\,[0, 18]$ | $8\,[-1, 10]$ |
| 278 | $22\,[-3, 54]$ | $11\,[-11, 26]$ | $10\,[-2, 14]$ | $12\,[-3, 31]$ | $14\,[0, 29]$ |
| 464 | $16\,[-2, 41]$ | $7\,[-9, 22]$ | $5\,[-2, 8]$ | $7\,[-1, 9]$ | $6\,[-4, 10]$ |
| 774 | $12\,[-1, 33]$ | $7\,[-4, 14]$ | $6\,[-2, 8]$ | $4\,[-5, 9]$ | $6\,[-4, 14]$ |
| 1291 | $15\,[-1, 39]$ | $5\,[-3, 16]$ | $3\,[-1, 4]$ | $6\,[-7, 7]$ | $1\,[-6, 6]$ |
| 2154 | $12\,[-1, 21]$ | $18\,[0, 19]$ | $1\,[-2, 2]$ | $2\,[-3, 2]$ | $2\,[-3, 2]$ |
| 3593 | $9\,[0, 13]$ | $10\,[0, 18]$ | $0\,[0, 1]$ | $0\,[-1, 0]$ | $-1\,[-1, 0]$ |
| 5994 | $4\,[0, 6]$ | $9\,[0, 16]$ | $0\,[0, 0]$ | $0\,[0, 0]$ | $0\,[0, 0]$ |

We also provide a granular breakdown of the experimental results. To avoid visual clutter in the main text, we present highlight plots for key baselines.

Specifically, we focus on the comparison between `MAR-KL DRO` (Ours) and the competitive baselines (e.g., `Impute+KL-DRO`). As observed in the detailed cost analysis, our method consistently exhibits a tighter cost distribution around the optimum, whereas baselines often suffer from significant outliers when the sample size is limited ($n < 500$).

For each training sample size $n$ and each method, we also report the reliability. Let $B_r \in \{0, 1\}$ denote the reliability indicator in replication $r$, and let $\hat{p} = \frac{1}{R}\sum_{r=1}^{R} B_r$ be the empirical reliability over $R = 100$ independent replications. We report $\hat{p}$ together with a $95\%$ confidence interval constructed using the Wilson score method (Wilson, 1927) for binomial proportions. The Wilson interval does not rely on normal approximations and provides accurate coverage even for moderate sample sizes and probabilities close to $0$ or $1$. For a fixed training sample size $n$ and a fixed method, each replication is generated by rerunning the entire experimental pipeline with an independent random seed. In particular, the training data, the missingness pattern, the tuning split, and the out-of-sample evaluation set are all resampled independently across replications. As a result, the resulting reliability indicators across replications can be viewed as independent and identically distributed Bernoulli trials, which justifies the use of binomial proportion confidence intervals.

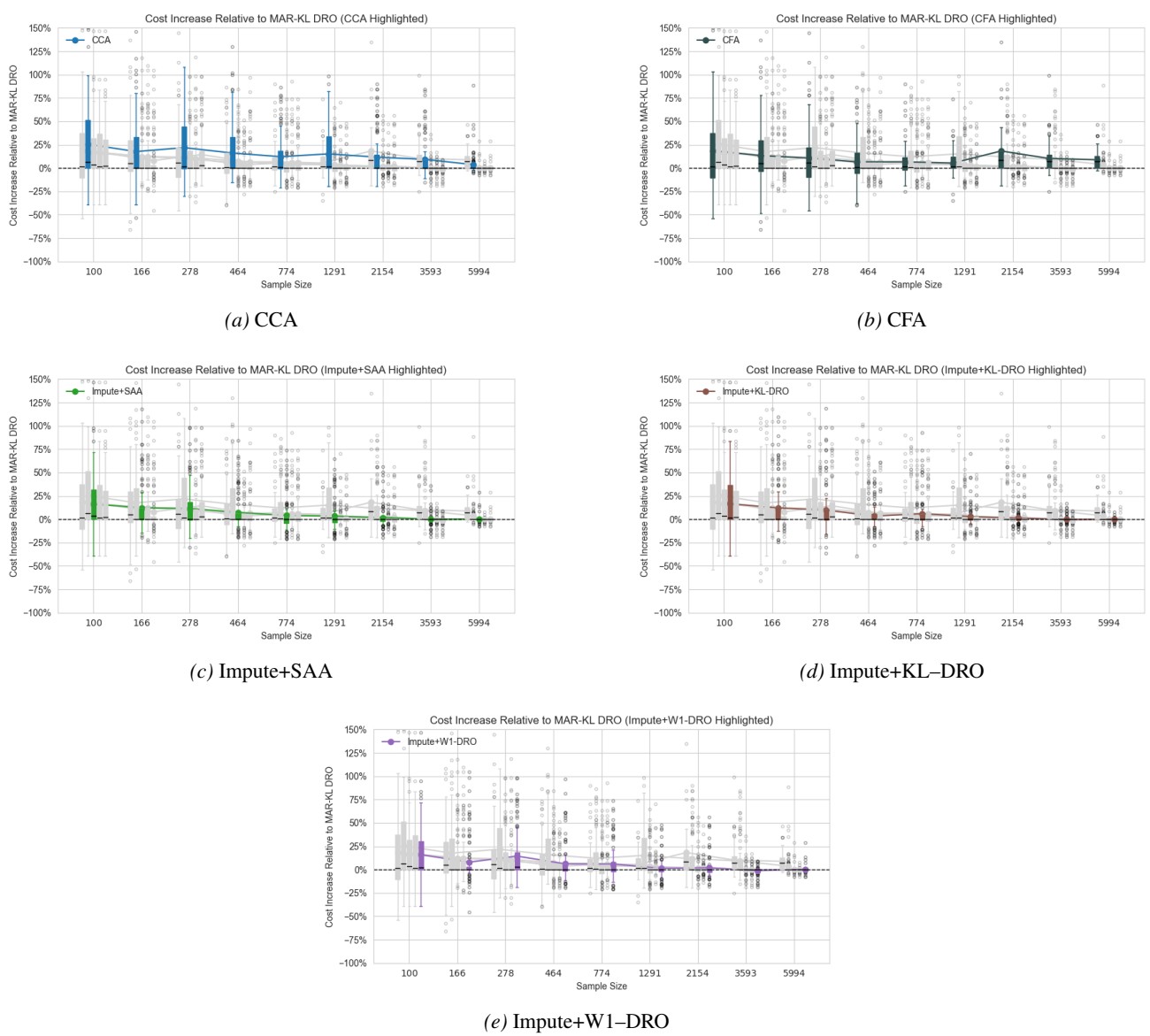

*(a)* CCA

*(b)* CFA

*(c)* Impute+SAA

*(d)* Impute+KL–DRO

*(e)* Impute+W1–DRO

*Figure 7.* Detailed breakdown of Figure 4.

*Table 7.* Reliability with 95% Wilson confidence intervals. Entries report the empirical reliability $\hat{p}$ together with the corresponding Wilson score interval (in percent) over $R = 100$ independent replications.

| $n$ | MAR–KL DRO | CFA | CCA | Impute+SAA | Impute+KL–DRO | Impute+W1–DRO |
|---|---|---|---|---|---|---|
| 100 | 63.0 [53.2, 71.8] | 32.0 [23.7, 41.7] | 1.0 [0.2, 5.4] | 13.0 [7.8, 21.0] | 16.0 [10.1, 24.4] | 16.0 [10.1, 24.4] |
| 166 | 68.0 [58.3, 76.3] | 29.0 [21.0, 38.5] | 6.0 [2.8, 12.5] | 18.0 [11.7, 26.7] | 21.0 [14.2, 30.0] | 18.0 [11.7, 26.7] |
| 278 | 74.0 [64.6, 81.6] | 38.0 [29.1, 47.8] | 12.0 [7.0, 19.8] | 22.0 [15.0, 31.1] | 43.0 [33.7, 52.8] | 27.0 [19.3, 36.4] |
| 464 | 75.0 [65.7, 82.5] | 35.0 [26.4, 44.7] | 21.0 [14.2, 30.0] | 21.0 [14.2, 30.0] | 43.0 [33.7, 52.8] | 29.0 [21.0, 38.5] |
| 774 | 80.0 [71.1, 86.7] | 39.0 [30.0, 48.8] | 19.0 [12.5, 27.8] | 28.0 [20.1, 37.5] | 52.0 [42.3, 61.5] | 34.0 [25.5, 43.7] |
| 1291 | 86.0 [77.9, 91.5] | 42.0 [32.8, 51.8] | 28.0 [20.1, 37.5] | 38.0 [29.1, 47.8] | 61.0 [51.2, 70.0] | 51.0 [41.3, 60.6] |
| 2154 | 86.0 [77.9, 91.5] | 41.0 [31.9, 50.8] | 36.0 [27.3, 45.8] | 37.0 [28.2, 46.8] | 65.0 [55.3, 73.6] | 54.0 [44.3, 63.4] |
| 3593 | 87.0 [79.0, 92.2] | 40.0 [30.9, 49.8] | 48.0 [38.5, 57.7] | 51.0 [41.3, 60.6] | 68.0 [58.3, 76.3] | 56.0 [46.2, 65.3] |
| 5994 | 89.0 [81.4, 93.7] | 37.0 [28.2, 46.8] | 65.0 [55.3, 73.6] | 55.0 [45.2, 64.4] | 66.0 [56.3, 74.5] | 64.0 [54.2, 72.7] |

## E.7. Additional details for Experiment III

We provide additional sensitivity analyses that complement the main experimental findings. In addition to the main text, we provide two results: (i) performance under oracle-selected (optimal) ambiguity radii, including both out-of-sample cost and reliability; and (ii) reliability under holdout-selected radii. All results are reported over $R = 100$ independent replications.

**Varying sample size.** Same as the main text setting, we fix the missingness strength at $\alpha = 0.8$ and vary the training sample size $n$ on a logarithmic grid between $10^2$ and $10^4$. For each $n$, we evaluate performance under oracle-selected (optimal) ambiguity radii, which are chosen retrospectively to minimize the out-of-sample cost over a predefined candidate set. The top panel of figure 8 reports the mean out-of-sample cost with $\pm 1$ standard deviation bands under oracle-selected ambiguity radii. As the sample size increases, all methods benefit from reduced variance, while MAR–KL DRO consistently achieves the lowest cost in the entire range of $n$.

The bottom panels report reliability under the oracle-selected (left) and holdout-selected (right) radii. While tuning noise under the holdout procedure leads to moderate variability for all methods, the reliability of MAR-KL DRO improves steadily as the sample size increases. Moreover, in both tuning schemes, MAR-KL DRO exhibits consistently high and stable reliability across sample sizes.

**Varying missingness strength.** We fix the training sample size at $n = 270$ and vary the missingness strength $\alpha \in [0.1, 1.0]$. The top panel of figure 9 reports the mean out-of-sample cost with $\pm 1$ standard deviation bands under oracle-selected radii, exhibiting a pattern consistent with the previous experiment (hold-out tuning scheme), with MAR–KL DRO showing a more stable performance advantage across the range of $\alpha$.

The bottom panels report reliability under oracle-selected (left) and holdout-selected (right) radii. The reliability of MAR–KL DRO remains high and stable, with only a mild degradation as $\alpha$ increases. Across both tuning schemes, MAR–KL DRO exhibits robust reliability throughout the full range of missingness strengths.

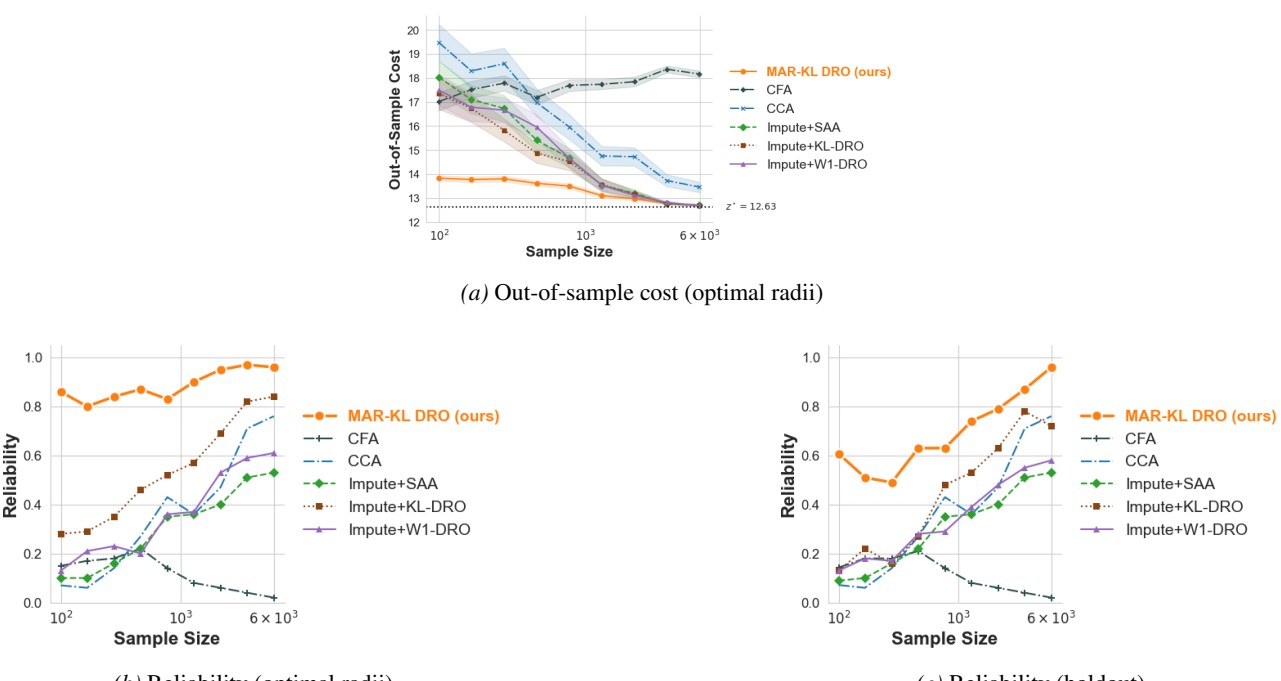

*(a)* Out-of-sample cost (optimal radii)

*(b)* Reliability (optimal radii)                *(c)* Reliability (holdout)

*Figure 8.* Sensitivity to sample size under fixed missingness strength $\alpha = 0.8$. Top: mean out-of-sample cost with $\pm 1$ standard deviation bands under oracle-selected radii. Bottom: reliability under oracle-selected (left) and holdout-selected (right) radii.

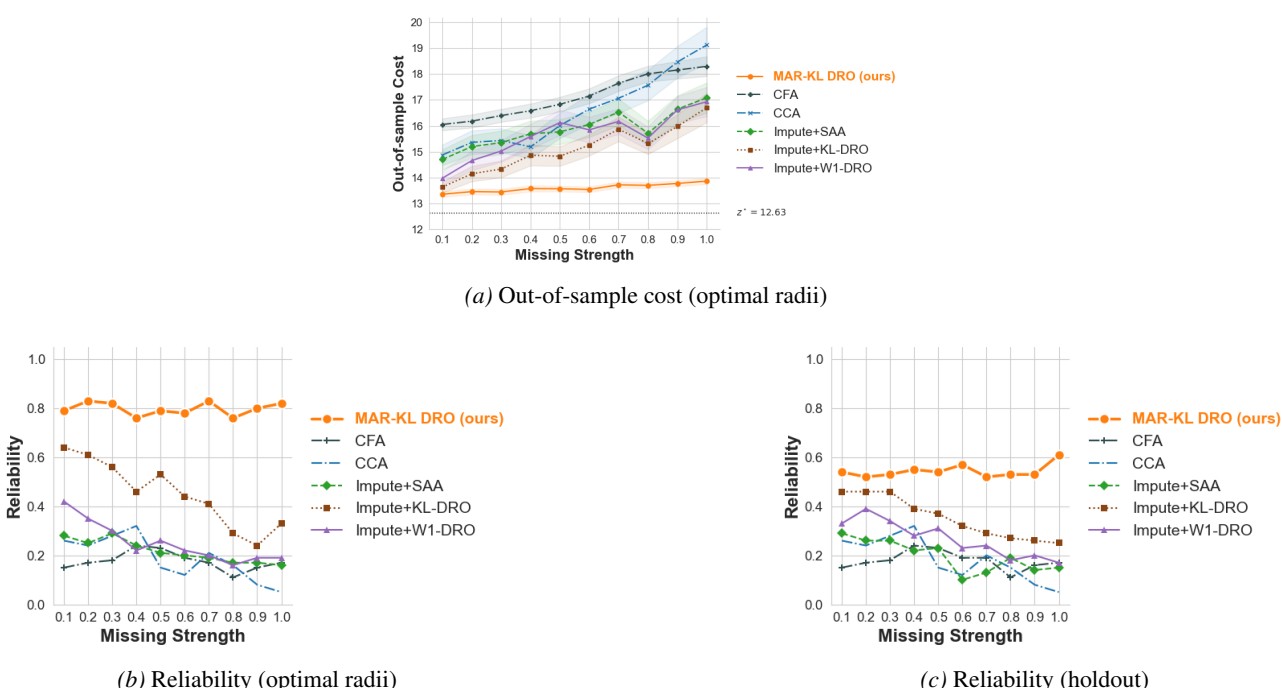

*(a)* Out-of-sample cost (optimal radii)

*(b)* Reliability (optimal radii)

*(c)* Reliability (holdout)

*Figure 9.* Sensitivity to missingness strength under fixed sample size $n = 270$. Top: mean out-of-sample cost with $\pm 1$ standard deviation bands under oracle-selected radii. Bottom: reliability under oracle-selected (left) and holdout-selected (right) radii.

