# OpenReview forum: "Robust Contextual Optimization with Missing Covariates"
_ICML.cc/2026/Conference — ICML 2026 spotlight_

### Official Review · Reviewer_A6RL · 2026-03-08

**Soundness:** 4
**Presentation:** 4
**Significance:** 4
**Originality:** 4
**Overall Recommendation:** 6
**Confidence:** 4

**Summary:**

This paper studies contextual stochatic optimization (CSO) when the cotextual information is partially missing, which is common in practice and makes standard CSO approaches not available to use.
The authors propose a distributionally robust optimization (DRO) that does not impute missing covariates. Instead, they build a latent joint  distributional ambiguity set over historical data under the so-called missing-at-random (MAR) mechanism. They then provide tractable convex reformulation and (non)-asymptotic statistical guarantees of their framework under different data regimes, e.g. discrete covariates/outcomes and continuous outcomes, etc. Empirically they test the performance on contextual newsvendor problems and demonstrate their advantages over other methods, e.g. CCA, CFA, impute-then-optimize pipelines.

The contributions are listed as follows:
(1) They utilize the data messing mechanism and propose a DRO framework to solve the problem, avoiding the impute-then-optimize pipelines and directly integrating estimation with optimization; (2) They provide (non)-asymptotic statistical guarantees and tractable reformulations for various data geometries. (4) They empirically demonstrate their advantages to baseline approaches.

**Compliance With Llm Reviewing Policy:**

Affirmed.

**Final Justification:**

Based on the responses which have addressed my concerns, I maintain strong accept (6) and recommend it as an oral paper.

**Key Questions For Authors:**

See the weakness part.

**Limitations:**

yes

**Strengths And Weaknesses:**

Strength:

1. The problem itself is important in contextual stochastic optimization. The authors successfully identify the problem and the suitable missing data mechanism (MAR), which helps the theoretical guarantees provided later.

2. The results are comprehensive. Both computational and statistical theoretical guarantees are provided, across discrete and continuous data regimes. Empirical findings validate the theoretical insights.

3. The technical analysis and writing is solid and clear. Though I did not check the proof details to the full extent, I find the analysis is non trivial, at least not an immediate translation from standard optimization/statistics textbook or some previous seminal papers. Many special structures are explored to generate the final results. The mechanisms and assumptions are described properly, instead of merely listing them without giving any explanations. The paper itself is indeed educational.

Weakness and questions:

1. The KL/PKL divergence based DRO seems bisect the data regimes. Is it possible to use a uniform metric that achieves the best of both worlds, e.g. Wasserstein metric?

2. The authors mentioned MNAR is beyond the scope of this work. I am curious why MNAR is essentially technically much harder than MAR (though I know they are indeed harder from the description). For example, what kind of non-identifiability issues would occur? Can you bring any examples? If I want to directly translate the analysis to MNAR settings, what parts will break?

3. It could be great is the authors tailor the length of the paper a bit since is not reasonable to expect reviewers to be able to cover this much material in the limited time frame of the review process in ICML. The current version is more appropriate as a journal version.

---

> ### Author Rebuttal · Authors · 2026-03-30
>
> We sincerely appreciate the time you dedicated to reviewing our paper, and we are grateful for the valuable insights you provided!  Please see below the response to the remaining questions :
>
> ---
>
> > **Q1**:  The KL/PKL divergence based DRO seems bisect the data regimes. Is it possible to use a uniform metric that achieves the best of both worlds, e.g. Wasserstein metric?
>
> **Re**: Many thanks for this insightful question! We did consider working directly with the Wasserstein distance to generate a unified ambiguity set for discrete and continuous outcomes Y. However, this approach appears to also introduce several practical and conceptual difficulties.
>
> First, from a theoretical perspective, the Wasserstein distance depends much more explicitly on the metric structure of the sample space. In many missing-covariate settings, however, the covariates are typically categorical, for which an information-theoretic divergence seems a more natural structural choice.
>
> Second, from a computational perspective, it is generally difficult to express the $H$ operator in terms of the Wasserstein distance. One possible route would be to consider causal transport, but our preliminary investigations suggested that the resulting optimization formulation is intractable to solve.
>
> Third, we also considered a hybrid KL--Wasserstein formulation, although we did not pursue this in the current paper. As compared with LP-type distances, convergence under the Wasserstein distance typically requires a combination of weak convergence and moment control, and the corresponding reliability results generally hold only under an additional condition of finite $p$-moments.
>
> We hence ended up with the current choice of KL/PKL divergences for discrete and continuous Y, respectively. Nevertheless, the KL divergence can be seen as a special case of PKL with $\delta = 0$.
>
> ---
>
> > **Q2**:  The authors mentioned MNAR is beyond the scope of this work. I am curious why MNAR is essentially technically much harder than MAR (though I know they are indeed harder from the description). For example, what kind of non-identifiability issues would occur? Can you bring any examples? If I want to directly translate the analysis to MNAR settings, what parts will break?
>
> **Re**:  From a theoretical perspective, handling MNAR generally requires additional assumptions or external information, such as a shadow variable or an instrumental variable, to restore identifiability. Without such ingredients, the problem is fundamentally non-identifiable: different joint distributions of $(X,Y)$, together with different missingness mechanisms, can induce exactly the same observable distribution. In that sense, the difficulty is structural.
>
> For this reason, a extension of our analysis to the MNAR setting would necessarily depend on the particular additional assumptions used to recover identifiability, since those assumptions effectively define a new admissible class $\mathcal{Q}$ of missingness mechanisms. With this change, several parts of the current analysis no longer go through in the same way. In particular, the convenient decoupling between $P(X,Y)$ and $q$ fails, which introduces substantial complications such as non-convexity.
>
> Our preliminary study suggested that, for certain special classes of MNAR problems, one can still obtain convex and computationally tractable formulations. However, developing this systematically would require substantial technical discussions, which we believe is beyond the scope of the present work.

---

> > ### Author Rebuttal · Reviewer_A6RL · 2026-03-31
> >
> > We thank the author for addressing our questions. We will keep our score.

---

> > > ### Author Response · Authors · 2026-04-04
> > >
> > > Thanks again for your positive evaluation and insightful suggestions!

---

### Official Review · Reviewer_zQe9 · 2026-03-12

**Soundness:** 3
**Presentation:** 3
**Significance:** 3
**Originality:** 3
**Overall Recommendation:** 4
**Confidence:** 3

**Summary:**

This paper proposes a distributionally robust optimization (DRO) framework for contextual stochastic optimization (CSO) when covariates are partially missing under a Missing at Random (MAR) mechanism. The authors construct a "latent ambiguity set" that encodes compatibility between candidate joint distributions and the observed (incomplete) data without requiring imputation. They derive tractable convex reformulations for discrete and continuous settings, establish finite-sample out-of-sample performance guarantees and asymptotic consistency, and evaluate on a contextual newsvendor problem.

**Compliance With Llm Reviewing Policy:**

Affirmed.

**Final Justification:**

Concern addressed.

**Key Questions For Authors:**

- Can you report wall-clock solve times for MAR-KL DRO versus the imputation baselines as a function of n and |X|×|Y|? The theoretical scalability discussion is insufficient without empirical timing.
- What happens when the missingness mechanism deviates from MAR? Even a small MNAR perturbation (e.g., missingness depends weakly on $x_m$) would be informative. Does performance degrade gracefully or catastrophically?

**Limitations:**

yes

**Strengths And Weaknesses:**

**Strengths**
- Principled integration of missingness into the optimization pipeline. The core idea, building ambiguity sets directly from incomplete observations while encoding the MAR structure, is well-motivated and technically sound. The equivalence result in Proposition 3.2 eliminates the missingness mechanism as an explicit optimization variable, which is a genuine contribution.
- Complete theoretical development for the discrete case. The chain from the abstract formulation (Robust-CSO) through the lifted convex reformulation (Proposition 3.5), the dual (Theorem 3.6), finite-sample guarantees (Theorem 3.7), and asymptotic consistency (Theorem 3.8) is logically coherent.
- Empirical reliability metric is informative. Reporting reliability alongside cost is a good practice and directly relates to the theoretical guarantees. Figure 2(b) and Table 5 show that MAR-KL DRO achieves meaningfully higher reliability than all baselines across sample sizes.

**Weeknesses**
- Extremely narrow experimental evaluation. One synthetic problem, one cost structure (piecewise linear newsvendor), discrete covariates with |X| = 27, outcomes in {0,...,12}. The total state space is small enough that the method's advantages may stem from exact enumeration that wouldn't scale. No computational time comparisons are reported.
- Scalability concerns are unaddressed. The paper acknowledges (Section 3.1.1) that the bottleneck scales with |X| × |Y|, and mentions separation methods, but provides no computational experiments. For the tested instance |X| × |Y| = 27 × 13 = 351, which is trivial. What happens at |X| = $10^4$ or with continuous covariates?
- The missingness mechanism in experiments is exactly MAR by construction. The logistic model in E.2 is precisely MAR, which perfectly matches the method's assumptions. This makes the experimental comparison favorable by design.

---

> ### Author Rebuttal · Authors · 2026-03-31
>
> Thank you for the thorough review and overall recognition of our work. We are happy to address the following points.
>
> ---
>
> > **Q1/W1/W2**:  Extensive evaluation and scalability concerns of the reformulation
>
> **Re** : As those questions all pertain to the experiments and computational aspects of the paper, we respond to them together in a sequence.
>
> - Computational burden. Following the idea in Appendix D.1, we first provide an experiment that isolates the effect of the sample size n. In this experiment, we fix the support size at 1250 and vary only n on a logarithmic scale. The results below show that the runtime remains fairly stable as n increases, indicating that the data sample size is not the primary computational bottleneck.
>
> | n               |  100 |  231 |  533 | 1232 | 2848 | 4328 | 10000 |
> | --------------- | ---: | ---: | ---: | ---: | ---: | ---: | ----: |
> | mean time (sec) | 2.67 | 2.59 | 2.65 | 2,73 | 2.81 | 2.82 |  3.09 |
>
> This result is also consistent with the complexity discussion in the paper. As a benchmark, we report the computational time of the impute-then-optimize methods:
>
> | n             |  100 |  231 |  533 | 1232 | 2848 | 4328 | 10000 |
> | ------------- | ---: | ---: | ---: | ---: | ---: | ---: | ----: |
> | Impute+SAA    | 0.08 | 0.12 | 0.16 | 0.28 | 0.60 | 1.08 |  2.37 |
> | Impute+KL-DRO | 0.05 | 0.10 | 0.16 | 0.33 | 0.65 | 1.14 |  2.70 |
> | Impute+W1-DRO | 0.06 | 0.11 | 0.17 | 0.41 | 0.95 | 2.16 |  7.19 |
>
> As the results demonstrate, these benchmark methods scale linearly/superlinearly with the sample size. In particular, the Impute+W1-DRO method becomes considerably more time-consuming to solve when the sample size is large.
>
> Next, we conduct experiments on scalability with respect to the support size, with or without applying the separation approach.
>
> - Effect of support size. We fix the sample size at 1000 and vary only the support size, without implementing the separation approach.
>
> | Support size  | $10^2$ | $10^3$ | $10^4$ |
> | :-----------: | :----: | :----: | :----: |
> | Mean time (s) |  0.28  |  2.95  | 29.79  |
>
> The results show that the computational time scales essentially linearly with respect to $|X||Y|$.
>
> - Acceleration via separation. After incorporating the separation idea, the computational times admit a substantial speedup.
>
> | Support size  | $10^2$ | $10^3$ | $10^4$ | $10^5$ | $10^6$ |
> | :-----------: | :----: | :----: | :----: | :----: | :----: |
> | Mean time (s) |  0.21  |  0.45  |  3.67  | 17.27  | 151.23 |
>
> One can observe that the computational time scales essentially sublinearly with respect to $|X||Y|$. Notably, the runtime for the 100,000 support-case is now even shorter than that of the previous approach for the 10,000-support case.
>
> - Furthermore, we test our approach beyond the piecewise linear cost structure. Specifically, we replace the newsvendor objective function with a quadratic cost and the outcome Y is continuous. The computational time of this extended model setting (without separation) is reported below. From this table, we observe that the computational performance does not change significantly.
>
> | Support size  | $10^2$ | $10^3$ | $10^4$ |
> | :-----------: | :----: | :----: | :----: |
> | Mean time (s) |  0.23  |  2.83  | 28.20  |
>
> - Finally, for the case of continuous covariates, we kindly refer the reviewer to Appendix D.4, where we discuss an extension framework.
>
> We hope these results clarify your questions.
>
> ---
>
> > **W3/Q2**:  What happens when the missingness mechanism deviates from MAR? Even a small MNAR perturbation (e.g., missingness depends weakly on $x_m$) would be informative. The missingness mechanism in experiments is exactly MAR by construction ... This makes the experimental comparison favorable by design.
>
> **Re** : This is an excellent question! We have conducted additional experiments for an MNAR setting that mildly departs from MAR, obtained by introducing an additional dependence on $X_m$ compared with the MAR mask-generation mechanism. We find that, although the performance degrades slightly, our method still outperforms other benchmark approaches, e.g., the impute-then-optimize pipelines.  The corresponding results are reported in the [figure](https://anonymous.4open.science/r/ICML26-anonymous/output.png).
>
> Moreover, the type of slight departure from MAR described here can in fact be captured by a marginal sensitivity model. In particular, the odds ratio under MNAR is assumed to be bounded by a sensitivity parameter $\Gamma \ge 1$ and its reciprocal $1/\Gamma$ relative to the empirical odds ratio under MAR. Under this framework, our formulation admits a natural extension by redesigning the missing mechanism $\mathcal{Q}$ and enjoys tractable convex reformulation. When one writes out this formulation explicitly, it becomes clear that, for $\Gamma$ is not too far from 1, the MAR-based model still enjoys strong theoretical reliability as well as potentially meaningful out-of-sample performance guarantees.

---

> > ### Author Rebuttal · Reviewer_zQe9 · 2026-04-02
> >
> > Thanks for the authors' response. The detailed analysis of the computation burden and the MNAR experiment addressed my concern. I keep my positive score for this paper.

---

> > > ### Author Response · Authors · 2026-04-04
> > >
> > > Thanks again for your positive evaluation and constructive suggestions!

---

### Official Review · Reviewer_2JP1 · 2026-03-13

**Soundness:** 4
**Presentation:** 3
**Significance:** 4
**Originality:** 3
**Overall Recommendation:** 5
**Confidence:** 4

**Summary:**

This paper proposes a distributionally robust optimization (DRO) framework for contextual optimization with missing covariates that does not require missing-value imputation. The authors propose an ambiguity set that incorporates the Missing At Random (MAR) mechanism and reformulates it into a tractable convex optimization problem. The authors provide some theoretical properties, including finite-sample guarantees and asymptotic optimality. Numerical experiments demonstrate that the proposed method outperforms conventional impute-then-optimize methods in reliability and out-of-sample performance.

**Compliance With Llm Reviewing Policy:**

Affirmed.

**Final Justification:**

Since the authors carefully addressed all of my concerns, I maintain my positive evaluation.

**Key Questions For Authors:**

1. The authors extend the method to continuous outcomes by assuming the existence of a finite partition satisfying Assumption 3.10.
While I understand this as a technical assumption, it is not obvious how to construct such a partition in practice.
How large should we set the number of partitions $J$?
Also, if the partition violates Assumption 3.10, I am interested in the extent of the negative impact on the performance of the proposed method.
2. Could the authors explain  the technical difficulties of handling missing values in a DRO model using the Wasserstein distance?
If we use the KL divergence, partitioning seems inevitable when dealing with continuous values.  Wouldn’t the Wasserstein distance allow for a more natural treatment of continuous missing values?
I would like to hear the authors’ insights on whether using the Wasserstein distance could potentially relax or entirely remove structural assumptions on the missing mechanism, such as Assumption 3.10.

**Limitations:**

yes

**Strengths And Weaknesses:**

Strengths

- Addressing missing covariates in contextual optimization has broad practical significance for both the machine learning and operations research communities.
- The KL-divergence-based formulation is reasonable. The theoretical contributions are solid, with clear investigations of the model's statistical properties.
- Extending the method to handle continuous covariates, rather than limiting it to discrete ones, increases its practical value.
- The technical sections are written clearly and are easy to follow.

Weaknesses

- Because the method relies on the KL divergence, handling continuous variables requires heuristic modifications.
- The scalability of the proposed method is unclear, particularly for high-dimensional random variables or a large number of bins $J$ for continuous values. The size of the random variable vector is $|\mathcal{X}| \times |\mathcal{Y}|$, which is a critical parameter determining the method’s applicability. This size should be explicitly discussed in the main text when describing the experimental setup, rather than leaving it in the appendix.

---

> ### Author Rebuttal · Authors · 2026-03-30
>
> Thank the reviewer for the positive assessment and for offering constructive suggestions. We appreciate the opportunity to further discuss the following questions :
>
> ---
>
> > **Q1**:  The authors extend the method to continuous outcomes by assuming the existence of a finite partition satisfying Assumption 3.10... I am interested in the extent of the negative impact on the performance of the proposed method.
>
> **Re** : Thank you for this thoughtful question!  This assumption was introduced mainly for the theoretical analysis of a given calibration rule. Departures from this assumption primarily affect the asymptotic convergence properties of the fixed calibration rule. In the finite-sample regime, the partition affects only the closed-form calibration rule that we provided. In practical applications, however, the radii can be selected through a standard cross validation, so it is not necessary to explicitly estimate such a partition therein.
>
> We therefore recommend constructing an approximate partition, either using domain knowledge of response homogeneity groups/cells or using a data-driven procedure (e.g., clustering) that enforces minimum cell sizes. Then, one can conduct local hyperparameter search and cross-validation to select suitable radii.
>
> ---
>
> > **W1(Q2)**: - Because the method relies on the KL divergence, handling continuous variables requires heuristic modifications... I would like to hear the authors’ insights on whether using the Wasserstein distance could potentially relax or entirely remove structural assumptions on the missing mechanism, such as Assumption 3.10.
>
> **Re** : In general, Assumption 2.4 is difficult to remove, and positivity of the propensity score is a standard condition for identifiability. Regarding Assumption 3.10, we believe that it is possible to relax it with an additional modification: specifically, the distance should no longer be defined with respect to the full observed-data distribution, but rather with respect to a projected component corresponding to the $M=0$ subsample. The trade-off, however, is a loss of efficiency and information, although this would relax the regularity conditions of missingness.
>
> In addition, we did consider working directly with the Wasserstein distance. Nonetheless, this introduces several practical and conceptual difficulties.
>
> - From a computational standpoint, it is generally difficult to express the $H$-operator in terms of the Wasserstein distance. Our preliminary investigations suggested that the resulting optimization problem is typically intractable in the nonparametric setting.
>
> - The Wasserstein distance depends much more explicitly on the metric structure of the sample space. In many missing-covariate settings, however, the covariates are often categorical, for which an information-theoretic divergence may provide a more natural structural choice.
>
> - We also considered a hybrid KL--Wasserstein formulation, although we did not pursue it in the current paper. Relative to the present setup, convergence under the Wasserstein distance typically requires both weak convergence and moment control, and the corresponding guarantees generally hold only under an additional condition of finite \(p\)-moments.
>
> We hope this helps clarify our modeling choice.
>
> ---
>
> > **W2**:  The scalability of the proposed method is unclear. This support size should be explicitly discussed in the main text when describing the experimental setup, rather than leaving it in the appendix.
>
> **Re** : Thank you for raising this point. In the current manuscript, tractability and computational issues are discussed in the main text after Theorems 3.6 and 3.14, with additional discussion provided in Appendices D.1 and D.3.  In the revised manuscript, we will use the additional space to further expand this discussion in the main text, e.g., computational performance of separation-based methods.
>
> Specifically, we have done additional experiments on the scalability with respect to the support size, with or without applying the separation approach.
>
> - Effect of support size. We fixed the sample size at 1000 and varied only the support size, without implementing the separation idea. The results suggest that the computational time scales approximately linearly with support size.
>
> | Support size  | $10^2$ | $10^3$ | $10^4$ |
> | :-------: | :----: | :----: | :----: |
> | Mean time (s) |  0.28  |  2.95  | 29.79  |
>
> - Acceleration via separation. After incorporating the separation approach, the computational time admits a substantial speedup.
>
> | Support size  | $10^2$ | $10^3$ | $10^4$ | $10^5$ | $10^6$ |
> | :-------: | :----: | :----: | :----: | :----: | :----: |
> | Mean time (s) |  0.21  |  0.45  |  3.67  | 17.27  | 151.23 |
>
> The results suggest that the computational time scales sublinearly with respect to support size. Notably, the runtime for the 100,000 support-case is now even shorter than that of the previous approach for the 10,000-support case.

---

> > ### Author Rebuttal · Reviewer_2JP1 · 2026-04-03
> >
> > Thank you for the detailed response. Since my initial concerns have been resolved through the authors' clarifications, I will keep my positive evaluation.
> >
> > The additional experiments on scalability with respect to the support size are interesting. I look forward to seeing the expanded discussions in the revised manuscript.

---

> > > ### Author Response · Authors · 2026-04-04
> > >
> > > Thanks again for your positive evaluation and constructive suggestions!

---

### Official Review · Reviewer_KuJU · 2026-03-13

**Soundness:** 3
**Presentation:** 2
**Significance:** 3
**Originality:** 3
**Overall Recommendation:** 4
**Confidence:** 1

**Summary:**

This paper studies contextual stochastic optimization with partially missing covariates. The key idea is to define a latent ambiguity set over complete-data laws that are consistent with the observed incomplete-data distribution under an admissible MAR missingness mechanism, and then solve a distributionally robust conditional optimization problem over this set.

**Compliance With Llm Reviewing Policy:**

Affirmed.

**Final Justification:**

Core concerns addressed. Hope authors incorporate necessary discussions in the revised version.

**Key Questions For Authors:**

Q1: The latent ambiguity set is the conceptual core of the paper. Could the authors provide a more intuitive explanation of Definition 2.6, especially for readers outside DRO or measure-theoretic statistics?
Q2: The paper notes that the computational bottleneck is governed by the latent support size. Can the authors provide stronger empirical evidence on this?

**Limitations:**

see above.

**Strengths And Weaknesses:**

Strengths:
S1: The problem is well motivated. The paper addresses a clear and practically important gap: contextual optimization methods usually rely on fully observed covariates, whereas missing covariates are common in modern decision-making pipelines.
S2: The method section is substantial. For the discrete case, the paper derives an equivalent transformed ambiguity set and a convex reformulation; for the continuous-outcome case, it introduces the PKL divergence and develops a tractable surrogate characterization.

Weaknesses:
W1: The main novelty is meaningful but somewhat incremental relative to the broader DRO literature. There is already prior work on integrated DRO with incomplete data, so the novelty here is best understood as bringing that perspective into MAR-aware contextual conditional optimization, rather than introducing an entirely new paradigm.
W2: The presentation is mathematically dense. While the notation is mostly standard, some symbols such as the product measure notation in Assumption 2.2 may be too terse for a broader ML audience, and the intuition behind the latent ambiguity set could be explained more simply.
W3: Tractability is established, but the computational burden is not negligible, which needs more discussion.

---

> ### Author Rebuttal · Authors · 2026-03-30
>
> Thank you for your effort devoted to reviewing our paper and overall recognition of our work. We appreciate the opportunity to further discuss the following questions.
>
> ---
>
> > **W1**: The main novelty is meaningful but somewhat incremental relative to the broader DRO literature. There is already prior work on integrated DRO with incomplete data, so the novelty here is best understood as bringing that perspective into MAR-aware contextual conditional optimization, rather than introducing an entirely new paradigm.
>
> **Re**: Prior literature and practice in the broader community address incomplete data through a *preprocess-data-then-optimize* paradigm. These approaches treat identification, inference, and optimization as separate steps, and therefore depend heavily on the specific imputation or estimation procedure used. As demonstrated in our work, they may raise concerns about reliability and out-of-sample performance, even when combined with conventional DRO methods. Our work does *not* follow this paradigm.
>
> Instead, we propose an integrated way to incorporate the missingness mechanism into the design of the ambiguity set and the ensuing DRO model.   **We believe this new, integrated approach could extend to a broader class of missingness mechanisms and a wider range of problems**.  One direct example is presented in Appendix D.2, where we extend the study to have missingness in both covariates and outcome. In this case, the problem can provide solutions to a non-contextual optimization problem by setting $|X|=1$ and always observable. From this perspective, our framework offers a new lens not only for contextual optimization, but also for the broader ML/OR community.
>
> ---
>
> > **W2(Q1)**: Intuition behind notation could be explained more simply. While the notation is mostly standard, some symbols such as the product measure notation in Assumption 2.2 may be too terse for a broader ML audience, and the intuition behind the latent ambiguity set could be explained more simply. Could the authors provide a more intuitive explanation of Definition 2.6, especially for readers outside DRO or measure-theoretic statistics?
>
> **Re**: Thank you for the suggestion. While the mathematical formalism is needed for a precise treatment of the problem, we will add informal interpretations for some technical notations to the revised version. To address your Q1 specifically, for definition 2.6, the intuition is that the ambiguity set contains all distributions over (X,Y) that could plausibly have generated the observed empirical distribution, under the admissible missingness mechanism.
>
> ---
>
> > **W3(Q2)**: Tractability is established, but the computational burden is not negligible, which needs more discussion. Can the authors provide stronger empirical evidence on "computational bottleneck is governed by the latent support size" ?
>
> **Re**: Thank you for raising this point! In the current manuscript, tractability and computational issues are discussed in the main text after Theorems 3.6 and 3.14, with additional discussions provided in Appendices D.1 and D.3.  In the revised manuscript, we will use the additional space to further expand this discussion in the main text, e.g., computation performance of separation-based methods.
>
> For your question Q2 specifically, we report an additional result that isolates the effect of the sample size n. In the experiment, we fixed the support size at $1250$ and varied only n on a logarithmic scale. By implementing the idea of Appendix D.1, the results below show that the runtime remains fairly stable as n increases, indicating that scales well with the data sample size. The result is consistent with the discussion in the paper.
>
> | \(n\)         | 100  | 231  | 533  | 1232 | 2848 | 4328 | 10000 |
> |:-------:|:----:|:----:|:----:|:----:|:----:|:----:|:-----:|
> | Mean time (s) | 2.67 | 2.59 | 2.65 | 2.73 | 2.81 | 2.82 | 3.09  |
>
> Moreover, we have done additional experiments on the scalability with respect to the support size, with or without applying the separation approach.
>
> (1) Effect of support size. We fixed the sample size at 1000 and varied only the support size, without implementing the separation idea. The results suggest that the computational time scales approximately linearly with support size.
>
> | Support size  | $10^2$ | $10^3$ | $10^4$ |
> | :-------: | :----: | :----: | :----: |
> | Mean time (s) |  0.28  |  2.95  | 29.79  |
>
>  (2) Acceleration via separation. After incorporating the separation approach, the computational time admits a substantial speedup.  The results suggest that the computational time scales sublinearly with respect to support size. Notably, the runtime for the 100,000 support-case is now even shorter than that of the previous approach for the 10,000-support case.
>
> | Support size  | $10^2$ | $10^3$ | $10^4$ | $10^5$ | $10^6$ |
> | :-------: | :----: | :----: | :----: | :----: | :----: |
> | Mean time (s) |  0.21  |  0.45  |  3.67  | 17.27  | 151.23 |

---

> > ### Author Rebuttal · Reviewer_KuJU · 2026-04-04
> >
> > Thank you for your response!

---

> > > ### Author Response · Authors · 2026-04-04
> > >
> > > Thanks again for your positive evaluation and constructive suggestions!

---

### Decision · Program_Chairs · 2026-04-30

**Decision:**

Accept (spotlight)

**Comment:**

The paper addresses a critical gap in modern decision-making pipelines where missing data is common. It provides a principled integration of missingness into optimization by building ambiguity sets from incomplete observations and leveraging the Missing at Random (MAR) structure. The paper offers a complete theoretical development for discrete cases, and extends its reach by introducing a tractable surrogate characterization for continuous cases. Reviewers also highlighted the novel technical insights and solid theoretical foundation. The paper should clearly be accepted.